# Self-Verification Provably Prevents Model Collapse in Recursive Synthetic Training

**Shi Fu**[1]    **Yingjie Wang**[1*]   **Yuzhu Chen**[2]    **Li Shen**[3]    **Dacheng Tao**[1*]

[1]College of Computing and Data Science, Nanyang Technological University, Singapore,
[2]University of Science and Technology of China, Hefei, China,
[3]Shenzhen Campus of Sun Yat-sen University, Shenzhen, China
`shi011@e.ntu.edu.sg`, `yingjiewang@upc.edu.cn`,
`cyzkrau@mail.ustc.edu.cn`,`mathshenli@gmail.com`,`dacheng.tao@gmail.com`

## Abstract

Large generative models are increasingly trained on synthetic data from earlier generations, raising concerns about *model collapse*, a progressive performance decline consistently observed in empirical studies. However, theoretical understanding of recursive training dynamics and their failure modes remains limited. In this work, we theoretically show that recursive training inherently leads to exponential error growth unless mitigated by sufficient real data. Addressing the growing scarcity of real data, we introduce a self-verification mechanism enabling models to filter their outputs based on internal confidence scores without external validation. Through rigorous analysis, we derive finite-sample error bounds demonstrating that self-verification alone can prevent collapse, even in fully synthetic training regimes. Our theoretical framework extends to large language models (LLMs), characterizing the conditions under which recursive training can maintain stability without performance degradation.

## 1   Introduction

High-quality training data is a critical foundation for the remarkable success of large generative models, such as LLMs [Fu et al., 2024a]. However, it is increasingly acknowledged that the pool of publicly available, real-world data is nearing exhaustion [Villalobos et al., 2022]. As a result, the training of modern LLMs increasingly relies on synthetic data produced by previous generations of the models [Briesch et al., 2023, Martínez et al., 2023, Xing et al., 2025]. Moreover, even unintentionally, models today are often trained on synthetic data, as many existing datasets are already polluted with synthetic content [Schuhmann et al., 2022]. This paradigm, known as recursive synthetic data training, has raised significant concerns regarding the risk of *model collapse*, a phenomenon characterized by a drastic and often irreversible deterioration in model performance across generations [Shumailov et al., 2024, Alemohammad et al., 2024a, Bertrand et al., 2024, Dohmatob et al., 2024b].

Numerous empirical studies have highlighted the potential risks associated with model collapse in recursive training loops, including substantial reductions in output diversity [Guo et al., 2023, Zhu et al., 2024], sharply increasing error rates [Dohmatob et al., 2024b,c], and the amplification of biases [Wyllie et al., 2024]. To mitigate these risks, researchers have proposed various approaches, such as incorporating sufficient real data [Alemohammad et al., 2024a], augmenting synthetic datasets [Gerstgrasser et al., 2024], and introducing external verification mechanisms for synthesized data [Feng et al., 2025, Firdoussi et al., 2025]. Nevertheless, despite these empirical advances, there remains a notable gap in rigorous theoretical understanding of both the observed phenomena and the underlying dynamics of recursive synthetic training loops.

---

*Corresponding authors

39th Conference on Neural Information Processing Systems (NeurIPS 2025).

Initial theoretical explorations have started to address these gaps by analyzing population risk dynamics under specific assumptions [Bertrand et al., 2024, Dohmatob et al., 2024a, Seddik et al., 2024]. For instance, Bertrand et al. [2024] derived upper bounds on parameter deviations for likelihood-based models when real data is incorporated, establishing convergence under strict statistical and optimization assumptions. Further work by Fu et al. [2024b, 2025] relaxed these assumptions on error bounds through more fine-grained analyses. Fu et al. [2024b] provided bounds on the Total Variation distance for simplified diffusion models, linking distributional divergence to dataset size and the proportion of real data. Complementing this, Fu et al. [2025] introduced the concept of recursive stability to establish the first generalization error bounds. They demonstrated for transformer architectures that a constant proportion of real data is sufficient to prevent model collapse.

More recently, theoretical studies focusing on verification strategies have gained particular attention [Feng et al., 2025, Firdoussi et al., 2025]. These works investigated Gaussian mixture models with linear classifiers in non-recursive settings, modeling external verification of synthesized data as Bernoulli random variables. In particular, Feng et al. [2025] demonstrated, in the infinite-sample limit, that linear verifiers can significantly improve model robustness when training solely on synthetic data, while Firdoussi et al. [2025] extended these findings to scenarios involving a mixture of real and synthetic samples.

However, existing theoretical analyses suffer from critical limitations [Feng et al., 2025, Firdoussi et al., 2025]. Current frameworks lack rigorous external verification and rely on simplified probabilistic assumptions for data quality. These analyses remain narrowly focused on Gaussian mixture models, limiting their applicability to broader generative models, particularly transformer-based LLMs used in practice. Previous theoretical studies are also restricted to non-recursive training scenarios, whereas model collapse emerges specifically within recursive settings [Schaeffer et al., 2025]. Additionally, analyses typically rely on asymptotic regimes with infinite-sample assumptions, failing to capture realistic finite-sample conditions relevant to actual deployment. This paper addresses these gaps with the following contributions:

1. **Unified Framework for Self-Verification in Recursive Training:** We propose a general theoretical framework for recursive training with self-verification, where models filter their own outputs based on internally estimated confidence scores. This approach removes the need for external verification and enables model-internal quality control applicable to a broad class of generative architectures.

2. **Finite-Sample Error Bounds Across Training Regimes:** We provide rigorous finite-sample error bounds across three recursive training regimes. First, we show that naive recursive training leads to exponential error accumulation across generations. We then demonstrate that incorporating sufficient real data mitigates this degradation. Most notably, we prove that self-verification alone suffices to prevent model collapse, even in fully synthetic settings, by ensuring provable convergence of error.

3. **Theoretical Guarantees for Transformer-Based LLMs:** We extend our analysis to transformer-based LLMs, establishing convergence guarantees for self-verification through recursive coverage coefficient control and PAC-Bayesian analysis. Our results provide the first theoretical justification for stable, fully synthetic recursive training in high-capacity transformer architectures.

## 2 Related Work

High-quality training data is essential to the success of large generative models. However, the vast supply of human-annotated or naturally occurring data on the internet is nearing exhaustion [Villalobos et al., 2022]. As a result, LLMs are increasingly trained on data generated by earlier versions of themselves, forming recursive training loops, also known as self-consuming loops [Martínez et al., 2023, Bohacek and Farid, 2023, Shumailov et al., 2024, Tao et al., 2024, Schaeffer et al., 2025]. Recent high-profile research has highlighted the potential for dramatic degradation in model performance, a phenomenon known as *model collapse*, which represents a critical challenge for the future of AI development and deployment [Gibney, 2024, Alemohammad et al., 2024a, Dohmatob et al., 2024b].

Empirical studies have documented a number of alarming phenomena associated with model collapse in recursive training loops, including a substantial reduction in output diversity [Guo et al., 2023, Shumailov et al., 2024, Briesch et al., 2023, Zhu et al., 2024], rapidly increasing error rates [Fu et al., 2024b], and bias amplification [Wyllie et al., 2024]. To mitigate these effects, prior work has proposed several strategies, such as incorporating real data into training [Alemohammad et al., 2024a,

Kanabar and Gastpar, 2025], enlarging synthetic datasets [Dohmatob et al., 2024a, Gerstgrasser et al., 2024, Kazdan et al., 2024], or guiding the data generation process using reward or control signals [Gillman et al., 2024, Alemohammad et al., 2024b, Feng et al., 2025, Firdoussi et al., 2025].

Despite the wealth of empirical investigations, theoretical understanding of recursive synthetic training loops remains relatively limited [Bertrand et al., 2024, Dohmatob et al., 2024c, Fu et al., 2024b, 2025, Seddik et al., 2024, Dey and Donoho, 2024]. A line of recent work has begun to explore this direction by analyzing population risk dynamics under specific modeling assumptions, such as linear contexts [Dohmatob et al., 2024a], Gaussian models [Shumailov et al., 2024, Alemohammad et al., 2024a, Suresh et al., 2024, Jain et al., 2024], and asymptotic regimes [Marchi et al., 2024]. Furthermore, Bertrand et al. [2024] derived upper bounds on parameter drift in likelihood-based models under strong statistical assumptions, while Fu et al. [2024b, 2025] relaxed these constraints and provided generalization bounds for simplified diffusion and attention-based architectures.

Most closely related to our work are Feng et al. [2025] and Firdoussi et al. [2025], which introduce verification strategies into training loops to prevent model collapse. These studies investigated Gaussian mixture models with linear classifiers in non-recursive settings under the infinite sample size assumption. They modeled external verification of synthesized data as Bernoulli random variables and examined the impact on model performance. Specifically, Feng et al. [2025] demonstrated that linear verifiers can provide feedback that improves the robustness of models trained on fully synthesized data, while Firdoussi et al. [2025] extended these findings to mixed data scenarios.

Our work further advances this line of research by introducing the self-verification mechanism, extending the theoretical framework to general architectures like transformers, analyzing error accumulation in recursive settings, and deriving practical finite sample bounds.

## 3  Preliminary

In this section, we formalize the recursive training framework, where each model is trained on data generated by its predecessor. We then outline the self-verification mechanism, which enables models to evaluate and filter their own outputs to ensure stable training.

**Recursive Training Loops**. Let $\{\mathcal{G}_t\}_{t=0}^T$ be a sequence of models trained recursively. The initial model $\mathcal{G}_0$ is trained on a real dataset $S_0 = \{(x_{0,j}, y_{0,j}^*)\}_{j=1}^n \subseteq \mathcal{X} \times \mathcal{Y}$, where $x_{0,j}$ are inputs sampled from the data distribution and $y_{0,j}^*$ are ground-truth labels. For each generation $t = 1, \ldots, T$, model $\mathcal{G}_t$ is trained on a synthetic dataset $S_t = \{(x_{t,j}, \hat{y}_{t,j})\}_{j=1}^n$, where $x_{t,j} \sim \mathcal{X}$ and $\hat{y}_{t,j} \sim \mathcal{G}_{t-1}(x_{t,j})$ are labels predicted by the previous model. The process continues recursively until generation $T$.

Next, we introduce a structured framework that integrates self-verification into each training generation, consisting of two key components: (1) a confidence-guided filtering mechanism for self-verification, and (2) a recursive training procedure leveraging verified data. Together, these components enable robust self-evolution through selective data curation.

**Self-Verification via Confidence-Guided Filtering**. In the absence of external supervision, we introduce a self-verification mechanism that allows the model to evaluate the quality of its own outputs. The verification score is computed using a fixed evaluator $\mathcal{G}_0$, trained solely on the initial real dataset, and reflects the alignment of a generated output with the base model's inductive biases. Formally, the score for output $y$ given input $x$ is defined as:

$$s_{\text{verify}}(y \mid x) := \log \mathcal{G}_0(y \mid x).$$

This design offers two main benefits: it anchors verification to a stable reference, mitigating distributional drift across generations, and acts as an implicit regularizer that discourages semantic deviation from the base distribution.

Given an input $x_{t,j}$, we sample $N$ candidate outputs from the previous model $\mathcal{G}_{t-1}$, denoted $\{\hat{y}_{t,j,k}\}_{k=1}^N \sim \mathcal{G}_{t-1}(\cdot \mid x_{t,j})$. We then apply a verification-based filtering step that retains only candidates whose scores fall within a margin $\gamma > 0$ of the highest score in the sampled set:

$$\boldsymbol{y}_{t,j,\gamma}^+ = \left\{ \hat{y}_{t,j,k} \;\middle|\; s_{\text{verify}}(\hat{y}_{t,j,k} \mid x_{t,j}) \geq \max_{1 \leq k' \leq N} s_{\text{verify}}(\hat{y}_{t,j,k'} \mid x_{t,j}) - \gamma \right\},$$

A verification-weighted distribution over $\boldsymbol{y}_{t,j,\gamma}^{+}$ is then defined via a softmax over the scores:

$$\mathcal{G}_{t,N}^{*}(\hat{y} \mid x_{t,j}) = \frac{\exp\left(s_{\text{verify}}(\hat{y} \mid x_{t,j})\right)}{\sum_{\hat{y}' \in \boldsymbol{y}_{t,j,\gamma}^{+}} \exp\left(s_{\text{verify}}(\hat{y}' \mid x_{t,j})\right)}, \text{ where } \hat{y} \in \boldsymbol{y}_{t,j,\gamma}^{+}.$$

Then, a verified output $\hat{y}_{t,j}$ is sampled from this distribution. This mechanism allows the model to act as both generator and verifier, enforcing quality control in recursive training.

**Recursive Training with Verified Data**. The verified outputs from confidence-guided filtering are used to construct the synthetic training set for the next generation. For each input $x_{t,j}$, we sample a verified target $\hat{y}_{t,j} \sim \mathcal{G}_{t,N}^{*}(\cdot \mid x_{t,j})$, forming the dataset $S_t = \{(x_{t,j}, \hat{y}_{t,j})\}_{j=1}^{n}$. The next model $\mathcal{G}_t$ is trained by maximizing the log-likelihood over $S_t$:

$$\mathcal{G}_t = \arg\max_{\mathcal{G} \in \mathcal{H}} \sum_{j=1}^{n} \log \mathcal{G}(\hat{y}_{t,j} \mid x_{t,j}),$$

where $\mathcal{H}$ is the model hypothesis class. This process facilitates iterative self-improvement by leveraging verified high-quality outputs for training.

# 4 Main Results: Understanding and Preventing Model Collapse

In this section, we present our theoretical analysis of model collapse in recursive synthetic training to understand when and how this phenomenon can be prevented. We first demonstrate that, without intervention, error accumulates exponentially across generations, leading to inevitable model collapse. We then demonstrate that introducing real data mitigates this collapse. Finally, we prove that a self-verification mechanism alone can prevent collapse, even under fully synthetic training.

## 4.1 Assumptions and Definitions

We begin by stating the core assumptions and definitions that underpin our analysis.

**Assumption 1** (Confidence-Calibrated Agreement). Let $\mathcal{G}_0$ denote the base model trained on a ground-truth real dataset $S_0 = \{(x_j, y_j^*)\}_{j=1}^{n}$. For any input $x \in \mathcal{X}$, define the verified high-confidence prediction as: $\boldsymbol{y}_{\gamma}^{+} = \{y \mid s_{\text{verify}}(y \mid x) \geq \max_{y' \in \mathcal{Y}} s_{\text{verify}}(y' \mid x) - \gamma\}$, where $s_{\text{verify}}$ is chosen as $\log \mathcal{G}_0$. Let $y_t^{+}(x) = \arg\max_{y \in \mathcal{Y}} \mathcal{G}_t(y \mid x)$ denote the prediction of $\mathcal{G}_t$. Then, we assume the existence of a constant $\tau \in (0,1)$ such that:

$$\mathbb{P}_{x \sim \mathcal{X}}\left[y_t^{+}(x) = y^*(x) \mid \mathcal{G}_t(\boldsymbol{y}_{\gamma}^{+}(x) \mid x) \geq 1 - \tau\right] \geq 1 - \epsilon(\gamma),$$

where $\epsilon(\gamma)$ is a function that controls the residual error, $\tau$ is a fixed constant representing the confidence threshold, and $y^*(x)$ denotes the ground-truth label.

As the filtering becomes more selective ($\gamma \to 0$), we expect $\epsilon(\gamma) \to 0$ if the base model $\mathcal{G}_0$, trained on a real dataset, is well-calibrated. This expectation is reasonable, as extensive prior work has explored confidence calibration techniques that enable models to achieve well-calibrated predictions [Mehrtash et al., 2020, Wang et al., 2021, Zhu et al., 2022, Liu et al., 2025]. Furthermore, similar assumptions have been extensively employed in the literature on self-learning paradigms [Huang et al., 2025a,b].

**Definition 1** (Minimum Confidence). Let $y_t^{+}(x) = \arg\max_{y \in \mathcal{Y}} \mathcal{G}_t(y \mid x)$ denote the model's top prediction at generation $t$. We define the minimum confidence level of these predictions as: $C_0 = \min_{x \sim \mathcal{X}, \, t \in [0,T]} \mathcal{G}_t\left(y_t^{+}(x) \mid x\right)$.

The quantity $C_0$ denotes the minimum confidence assigned by any model in the sequence to its top prediction over the input distribution. Ensuring $C_0 > 0$ guarantees that a non-trivial level of certainty is maintained throughout the recursive training process. This assumption is standard in prior works [Zhang et al., 2023, Huang et al., 2025a].

**Definition 2** (Test Error). We define the test error of the final model $\mathcal{G}_T$ after $T$ recursive generations as: $\text{Err}(\mathcal{G}_T) = \mathbb{P}_{x \sim \mathcal{X}}\left[y_T^{+}(x) \neq y^*(x)\right]$, where $y_T^{+}(x) = \arg\max_{y \in \mathcal{Y}} \mathcal{G}_T(y \mid x)$ is the model's prediction, and $y^*(x)$ is the ground-truth label.

This metric provides a clear measure of model performance and serves as the foundation for analyzing error propagation in recursive training.

## 4.2 Model Collapse in Naive Recursive Training

We begin by analyzing the failure mode of naive recursive training, where models are trained solely on synthetic data generated by their predecessors without any verification. In this setting, performance inevitably degrades over generations, leading to eventual collapse. Our first theorem shows that prediction errors propagate and amplify exponentially across recursive steps.

**Theorem 1** (Error Propagation in Naive Recursive Training). *Let $\{\mathcal{G}_t\}_{t=0}^T$ denote a sequence of models trained via naive recursive training, where each $\mathcal{G}_t$ is trained solely on synthetic data generated by its predecessor $\mathcal{G}_{t-1}$, without any verification. Assume that for all $t \in [1, T]$, the Total Variation (TV) distance between successive generations satisfies $D_{\mathrm{TV}}\left(\mathcal{G}_t(\cdot \mid x), \mathcal{G}_{t-1}(\cdot \mid x)\right) \asymp D_{\mathrm{TV}}$, where $D_{\mathrm{TV}}$ denotes the characteristic magnitude of distributional shift across generations. Define the coverage coefficient as $C_\gamma := \mathbb{E}_{x \sim \mathcal{X}}\left[1/\mathcal{G}_0\left(\boldsymbol{y}_\gamma^+(x) \mid x\right)\right]$. Suppose the final model $\mathcal{G}_T$ satisfies Assumption 1. Then the test error after $T$ recursive steps satisfies:*

$$\mathrm{Err}(\mathcal{G}_T) \lesssim \epsilon(\gamma) + \frac{1}{\tau} \cdot D_{\mathrm{TV}} + \left(C_\gamma + D_{\mathrm{TV}} \log \frac{1}{D_{\mathrm{TV}}}\right) \cdot \left(\frac{1}{2C_0} + \frac{1}{2} \log \frac{1}{C_0}\right)^T. \tag{1}$$

**Remark 1. Exponential Error Amplification in Naive Recursive Training.** This theorem underscores the core limitation of naive recursive training: the prediction error grows exponentially with the number of generations $T$ when the condition $\frac{1}{2C_0} + \frac{1}{2}\log\frac{1}{C_0} > 1$ is met. Here, $C_0$, defined in Definition 1, denotes the minimum confidence assigned to the model's top prediction across all generations. In practice, $C_0$ tends to be small, making the condition readily satisfied. Notably, when $C_0 \leq 0.5$, the error term in Equation 1 exhibits exponential growth as $T$ increases. This growth stems from the fact that each model not only inherits information from its predecessor but also compounds its biases and errors, leading to eventual model collapse.

This result aligns with prior theoretical works [Bertrand et al., 2024, Gillman et al., 2024], which shows that recursive training without control leads to exponential error amplification. However, these analyses directly assume upper bounds on optimization and statistical errors from finite sampling, and typically posit unrealistic iterative retraining where each model inherits its predecessor's parameters and optimizer state [Schaeffer et al., 2025]. In contrast, we analyze training dynamics for specific generative models, including Transformer-based LLMs (Appendix I, Theorem 7), under a more practical setting where each model is trained from scratch on synthetic data.

The phenomenon of model collapse observed here is also consistent with empirical observations in recursive training setups, where the absence of quality control mechanisms results in severe performance degradation over successive generations [Shumailov et al., 2024, Alemohammad et al., 2024a]. This theoretical result formalizes these observations, providing a rigorous explanation for the inevitable instability in naive recursive training.

## 4.3 Mitigating Collapse with Real Data

A common approach to prevent model collapse is to inject real data during training. While this strategy has been empirically validated in previous work [Alemohammad et al., 2024a], we provide a theoretical characterization of how real data injection affects error propagation.

**Theorem 2** (Error Propagation with Real Data). *Let $\{\mathcal{G}_t\}_{t=0}^T$ be a sequence of models trained via recursive training that incorporates a proportion $\alpha \in [0, 1]$ of real data $\mathcal{D}_{real} = \{(x, y^*)\}$ at each generation. The remaining $1 - \alpha$ proportion of the training data comes from synthetic data generated by the previous model $\mathcal{G}_{t-1}$. Suppose that the TV distance satisfies $D_{\mathrm{TV}}\left(\mathcal{G}_t(\cdot \mid x), \mathcal{G}_{t-1}(\cdot \mid x)\right) \asymp D_{\mathrm{TV}}$ for all $t$, and that the final model $\mathcal{G}_T$ satisfies Assumption 1. Assume $\alpha > 1 - \frac{1}{2(1/C_0 - 1)}$, Then as $T \to +\infty$, the final error after $T$ recursive steps satisfies:*

$$\mathrm{Err}(\mathcal{G}_T) \lesssim \epsilon(\gamma) + \frac{1}{\tau} \cdot D_{\mathrm{TV}} + \frac{1 - \alpha}{C_0} D_{\mathrm{TV}} \log \frac{1}{D_{\mathrm{TV}}}.$$

**Remark 2. Error Stabilization through Real Data Injection.** This theorem demonstrates the stabilizing effect of incorporating real data into recursive training loops. Specifically, when the proportion of real data $\alpha$ satisfies the condition $\alpha > 1 - \frac{1}{2(1/C_0 - 1)}$, the exponential error growth observed in naive recursive training (Theorem 1) is mitigated. The resulting error bound becomes stable as $T \to +\infty$, with the dominant terms depending on $\epsilon(\gamma)$, the total variation distance $D_{\mathrm{TV}}$, and the proportion of synthetic data $(1 - \alpha)$.

Notably, as discussed in Assumption 1, $\epsilon(\gamma)$ can be made arbitrarily small through confidence calibration techniques [Liu et al., 2025]. Meanwhile, the TV distance $D_{\mathrm{TV}}$ can diminish under certain theoretical conditions. Prior studies have shown that $D_{\mathrm{TV}}(n)$ converges at a rate of $\mathcal{O}(n^{-1/4})$ in diffusion models [Fu et al., 2024b], with similar rates established in GANs [Liang, 2021]. These results suggest that as the number of training samples $n$ increases, the distributional shift across generations diminishes, further stabilizing the overall error bound.

The key insight from this result is that introducing even a modest fraction of real data, $\alpha > 1 - \frac{1}{2(1/C_0 - 1)}$, fundamentally changes the error dynamics and ensures that the exponential amplification factor from Theorem 1 is suppressed. From a practical perspective, this theorem underscores the importance of incorporating real data in recursive training loops to ensure the long-term stability of model performance. Even a small fraction of real data acts as a corrective mechanism, counteracting the biases, distributional shifts, and errors introduced by synthetic data. This result also aligns with prior empirical and theoretical findings [Briesch et al., 2023, Bertrand et al., 2024, Fu et al., 2024b, 2025, Seddik et al., 2024] that emphasize the critical role of real data in mitigating error propagation.

**Remark 3. Limitations of Real Data Dependency and Motivation for Self-Verification.** While this result confirms the effectiveness of incorporating real data in preventing model collapse, it also highlights a significant limitation: the approach relies on access to an ongoing stream of high-quality real data. In many practical scenarios, such data may be scarce, expensive to obtain, or entirely unavailable [Villalobos et al., 2022, Alemohammad et al., 2024a]. Furthermore, in publicly available datasets, it may be challenging to accurately distinguish between real and synthetic data [Sadasivan et al., 2023], which can undermine the stability of the training process.

This limitation motivates the need for alternative strategies like self-verification. By internally validating and correcting the quality of synthetic data, a self-verification mechanism can prevent error propagation across generations, offering a more resource-efficient and generalizable solution to model collapse without depending on external data.

### 4.4 Mitigating Collapse through Self-Verification

Our next main contribution is to demonstrate that stable recursive training is possible without real data or external supervision, by leveraging a self-verification mechanism. This builds on the insight that a model's verification ability often exceeds its generation capability [Weng et al., 2023, Huang et al., 2025a], allowing it to effectively filter its own outputs. We then present our main theoretical result, showing that self-verification can prevent model collapse:

**Theorem 3** (Error Bound with Self-Verification). *Let* $\{\mathcal{G}_t\}_{t=0}^{T}$ *be a sequence of models obtained via recursive training with the self-verification mechanism described in Section 3. Assume that for all* $t \in [1, T]$*, the TV distance satisfies* $D_{\mathrm{TV}}(\mathcal{G}_t(\cdot \mid x), \mathcal{G}_{t-1}(\cdot \mid x)) \asymp D_{\mathrm{TV}}$*. Suppose the final model* $\mathcal{G}_T$ *also satisfies Assumption 1. Then, after* $T$ *recursive generations, the final model* $\mathcal{G}_T$ *satisfies:*

$$\mathrm{Err}(\mathcal{G}_T) \lesssim \epsilon(\gamma) + \frac{1}{\tau} D_{\mathrm{TV}} + \left( \frac{1}{NC_0} \log \frac{1}{C_0} \right)^T \frac{C_\gamma}{N} \log \frac{1}{\tau} + \left( \frac{1}{N} + \frac{1}{NC_0} D_{\mathrm{TV}} \log \frac{1}{D_{\mathrm{TV}}} \right) \log \frac{1}{\tau}.$$

*Furthermore, if* $N > \frac{1}{C_0} \log \left( \frac{1}{C_0} \right)$*, the exponential term vanishes as* $T \to \infty$*, yielding:*

$$\mathrm{Err}(\mathcal{G}_T) \lesssim \epsilon(\gamma) + \frac{1}{\tau} D_{\mathrm{TV}} + \frac{1}{N} \left( 1 + \frac{1}{C_0} D_{\mathrm{TV}} \log \frac{1}{D_{\mathrm{TV}}} \right) \log \frac{1}{\tau}. \tag{2}$$

**Remark 4. The Role of Self-Verification in Error Control**. Theorem 3 demonstrates how self-verification prevents the exponential error growth shown in Theorem 1 during recursive training loops. Specifically, the method enforces quality control using a verification score $s_{\mathrm{verify}}(y \mid x) = \log \mathcal{G}_0(y \mid x)$ which quantifies how well a generated output aligns with the distribution of the initial model. This reference model, trained on real data, remains fixed across generations and serves as a stable anchor for self-verification. By selecting outputs within a margin $\gamma$ of the maximum score, the mechanism creates a filtered subset of high-quality candidates for training. This anchoring to the initial model $\mathcal{G}_0$ prevents distributional drift from compounding, ensuring each recursive generation remains tethered to a consistent inductive prior. As generations progress, outputs deviating from the original data manifold are excluded, effectively suppressing error amplification even without external supervision.

Specifically, Theorem 3 establishes that the exponential decay term vanishes asymptotically when $N > \log(1/C_0)/C_0$, where $N$, as described in Section 3, represents the number of data samples

used by the self-verification mechanism to select a subset of high-quality samples for training. This condition highlights the importance of both the size of the data pool and the quality of the selected samples in stabilizing the recursive training process. When this condition is met, the error bound converges to an asymptotic form, ensuring that the error remains bounded even as $T \to \infty$.

This result is particularly significant because it demonstrates that, through proper quality control mechanisms performed entirely within the model itself, recursive training can achieve sustained improvement without relying on real data or external verification. By leveraging self-verification to prioritize reliable outputs and suppress low-quality generations, the training process becomes robust to error amplification, enabling scalable and stable model development in resource-constrained or fully synthetic environments.

**Remark 5. Comparison with previous works.** The study of model collapse in recursive training loops was initiated by theoretical insights from Shumailov et al. [2024], Alemohammad et al. [2024a], which analyzed simplified Gaussian models under fully synthetic data settings. Most closely related to our work is the emerging line of research that introduces verification strategies into training loops to prevent model collapse [Feng et al., 2025, Firdoussi et al., 2025]. Specifically, Feng et al. [2025] and Firdoussi et al. [2025] investigated Gaussian mixture models with linear classifiers in non-recursive settings. Their analysis examined how external verification of synthesized data influences model performance as training dataset size approaches infinity. Feng et al. [2025] primarily demonstrated that appropriate feedback mechanisms can significantly improve the robustness of models trained on fully synthetic data, while Firdoussi et al. [2025] extended these findings to mixed data scenarios.

Our work makes significant theoretical advancements over Feng et al. [2025] and Firdoussi et al. [2025] in several key aspects:

1. **Self-Verification vs. External Verification:** While prior work often relies on external verifiers to guide data quality [Feng et al., 2025, Firdoussi et al., 2025], we propose a self-verification framework in which models assess their own outputs. This intrinsic approach removes the reliance on external quality signals, enables autonomous self-improvement. For advanced or superhuman models, internal verification capabilities may surpass what weak external mechanisms can offer [Wang et al., 2025].

2. **A General and Practical Theoretical Framework:** Our work considers a more general and practical theoretical framework compared to Feng et al. [2025] and Firdoussi et al. [2025], which focused on Gaussian mixture models and linear classifiers. Importantly, we do not impose any assumptions about data distribution, making our findings applicable to a wide range of real-world scenarios. Moreover, our framework addresses more realistic language model settings, and in Section 5, we extend our theoretical results to Transformer-based LLMs, demonstrating their applicability to modern and widely-used architectures.

3. **Recursive Training vs. Non-Recursive Settings:** While Feng et al. [2025] and Firdoussi et al. [2025] investigated non-recursive training settings, our work tackles the more complex and challenging recursive training setup, which is the primary scenario in which model collapse occurs [Shumailov et al., 2024, Alemohammad et al., 2024a, Schaeffer et al., 2025]. Recursive training involves compounding errors and distributional shifts across multiple generations, making it significantly harder to analyze theoretically. By addressing this setting, our work provides deeper insights into the dynamics of recursive training loops and offers solutions that are more meaningful and impactful for preventing model collapse in practical scenarios.

4. **Finite-Sample Guarantees vs. Asymptotic Analyses:** Prior work [Feng et al., 2025, Firdoussi et al., 2025] primarily established asymptotic results under infinite-sample assumptions, limiting their applicability to real-world training regimes. In contrast, we derive finite-sample generalization bounds that remain valid under practical recursive training scenarios.

In summary, we advance the field by introducing self-verification, developing a general framework applicable to transformer-based LLMs, analyzing recursive training dynamics, and deriving finite-sample guarantees. These contributions mark a significant step toward understanding and mitigating model collapse in generative models.

**Remark 6. Proof Sketch of Theorem 3**. Theorem 3 provides a recursive generalization bound for self-verifying models under synthetic training, and shows how error amplification can be effectively controlled through confidence-guided filtering. The key innovation lies in leveraging *recursive coverage coefficient control* to quantify the reliability of verified data and to suppress error propagation across recursive generations.

**Step 1: Verification-Induced Quality Control and Error Decomposition.** We leverage the verification score $s_{\text{verify}}(y \mid x) = \log \mathcal{G}_0(y \mid x)$ to filter model outputs and restrict supervision to high-quality generations. Let $\boldsymbol{y}_\gamma^+(x)$ denote the verified candidate set, and define the confident support region:

$$\mathcal{D}_1^+ := \left\{ x \in \mathcal{X} \,\middle|\, \mathcal{G}_0(\boldsymbol{y}_\gamma^+(x) \mid x) \geq \log(2/\tau)/N \right\}.$$

Using standard decomposition, the failure probability of the first-generation model can be split as:

$$\mathbb{P}_{x \sim \mathcal{X}} \left[ \mathcal{G}_1(\boldsymbol{y}_\gamma^+(x) \mid x) \leq 1 - \tau \right] \leq \mathbb{P}_{x \sim \mathcal{X}} \left[ \mathcal{G}_1(\boldsymbol{y}_\gamma^+(x) \mid x) \leq 1 - \tau, \, x \in \mathcal{D}_1^+ \right] + \mathbb{P}_{x \sim \mathcal{X}} \left[ x \notin \mathcal{D}_1^+ \right].$$

The first term is controlled via a Monte Carlo coverage argument and a TV distance bound between $\mathcal{G}_1$ and the empirical verification distribution $\mathcal{G}_{1,N}^*$, while the second is bounded using the expected inverse coverage. This yields:

$$\mathbb{P}_{x \sim \mathcal{X}} \left[ \mathcal{G}_1(\boldsymbol{y}_\gamma^+(x) \mid x) \leq 1 - \tau \right] \leq \frac{4}{\tau} \cdot D_{\text{TV}}(\mathcal{G}_1, \mathcal{G}_{1,N}^*) + \frac{C_\gamma}{N} \log \frac{2}{\tau},$$

with coverage coefficient $C_\gamma := \mathbb{E}_x \left[ 1/\mathcal{G}_0(\boldsymbol{y}_\gamma^+(x) \mid x) \right]$. This completes the base case.

**Step 2: Recursive Coverage Propagation and Error Control.** To extend the analysis across generations, we study how coverage evolves recursively. Specifically, the coverage coefficient at generation $t$ satisfies the recurrence:

$$C_{t+1} \lesssim \beta + \alpha C_t, \quad \text{with} \quad \alpha = \frac{1}{NC_0} \log \left( \frac{1}{C_0} \right), \quad \beta = \frac{1}{N} + \frac{D_{\text{TV}}}{NC_0} \log \left( \frac{1}{D_{\text{TV}}} \right),$$

where $C_0$ is the minimum confidence. Solving this recurrence gives:

$$C_t \lesssim \left( \frac{1}{NC_0} \log \left( \frac{1}{C_0} \right) \right)^t \cdot \frac{C_\gamma}{N} + \frac{1 + \frac{A}{C_0} \log \left( \frac{1}{A} \right)}{N - \frac{1}{C_0} \log \left( \frac{1}{C_0} \right)}.$$

Substituting this into the error bound completes the proof.

**Key Insight: Recursive Coverage Coefficient Control**. Unlike classical generalization bounds, which depend solely on data size and model capacity, recursive synthetic training introduces a new challenge: maintaining high-quality supervision under model-generated data. This requires bounding the evolution of coverage across generations: $C_{\gamma,t} := \mathbb{E}_x \left[ 1/\mathcal{G}_t(\boldsymbol{y}_\gamma^+(x) \mid x) \right]$, which quantifies the expected inverse confidence over the verified candidate region. Intuitively, a smaller $C_{\gamma,t}$ implies that the model assigns higher probability mass to its own verified predictions, indicating better self-consistency and internal alignment.

## 5 Application to Transformer-based LLMs

This section instantiates our theoretical framework on Transformer-based LLMs, which form the foundation of modern commercial systems such as GPT-4 [Achiam et al., 2023], Claude [Anthropic, 2024], LLaMA [Touvron et al., 2023], and Gemini [Team et al., 2023]. Our formalization considers token sequences $S_t = (x_1, \ldots, x_t)$ within a structured space $\mathcal{S} \subseteq \mathcal{X}^{\leq \ell}$, where $\mathcal{X}$ represents the token vocabulary and $\ell$ denotes maximum sequence length. The family of distributions induced by the model is denoted as $\{\mathcal{G}_\theta^{\text{LLM}} \mid \theta \in \Theta_{\text{LLM}}\}$, where $\theta$ represents the model parameters. Then, the initial model $\mathcal{G}_{\theta_0}^{\text{LLM}}$ is trained on a corpus of real token sequences $\{(x_1^{(j)}, \ldots, x_\ell^{(j)})\}_{j=1}^m$ using maximum likelihood estimation:

$$\theta_0 = \arg \max_{\theta \in \Theta_{\text{LLM}}} \frac{1}{n} \sum_{j=1}^m \sum_{t=1}^\ell \log \mathcal{G}_\theta^{\text{LLM}}(x_{t+1}^{(j)} \mid S_t^{(j)}),$$

where $S_t^{(j)} = (x_1^{(j)}, \ldots, x_t^{(j)}) \in \mathcal{S}$ represents each prefix sequence, and $n = m\ell$ is the total sample count. This initial model then participates in recursive training as described in Section 3, using the verification score function defined as $s_{\text{verify}}(x \mid S) := \log \mathcal{G}_{\theta_0}^{\text{LLM}}(x \mid S)$. Then, We adopt the standard Transformer architecture [Vaswani et al., 2017, Brown, 2020, Zhang et al., 2025, Hu et al., 2024], consisting of $D$ layers with multi-head self-attention (MHA) and feed-forward network (FFN) modules. For input matrix $X^{(0)} \in \mathbb{R}^{L \times d}$, computation in layer $t \in [D]$ follows:

$$Y^{(t)} = \Pi_{\text{norm}} \left[ \text{MHA}(X^{(t-1)}, W^{(t)}) + \gamma_1^{(t)} X^{(t-1)} \right], \quad X^{(t)} = \Pi_{\text{norm}} \left[ \text{FFN}(Y^{(t)}, A^{(t)}) + \gamma_2^{(t)} Y^{(t)} \right],$$

where $\gamma_1^{(t)}, \gamma_2^{(t)} \in [-1, 1]$ are residual weights, and $\Pi_{\text{norm}}$ normalizes rows to the unit $\ell_2$-ball. The FFN module is defined as $\text{FFN}(Y^{(t)}, A^{(t)}) = \text{ReLU}(Y^{(t)} A_1^{(t)}) A_2^{(t)}$, with $A_1^{(t)} \in \mathbb{R}^{d \times d_F}$ and $A_2^{(t)} \in \mathbb{R}^{d_F \times d}$. Token prediction logits are produced via $Y^{(D+1)} = \text{softmax}\left(I_L^\top X^{(D)} A^{(D+1)}/(L\beta)\right)$, where $I_L \in \mathbb{R}^L$ is the all-ones vector, $A^{(D+1)} \in \mathbb{R}^{d \times d_y}$ is the output projection, $\beta \in (0, 1]$ controls temperature, and $d_y$ is the output dimension.

For theoretical tractability, we assume uniformly bounded model parameters, as detailed in Appendix H. This standard assumption in Transformer analysis aligns with prior work [Li et al., 2023, Hu et al., 2024, Zhang et al., 2025, Fu et al., 2025]. In Appendix I, Theorem 7 demonstrates how naive recursive LLM training leads to exponential error growth, while Theorem 8 in Appendix J shows how incorporating real data prevents this collapse. We now present our key contribution: error bounds for LLMs with self-verification that maintain training stability without external real data.

**Theorem 4** (Error Bound for Transformer-based LLMs with Self-Verification). *Let $\mathcal{G}_{\theta_t}^{LLM}(x \mid S)$ denote the conditional generation distribution of a Transformer-based LLM at the $t$-th recursive generation, where $S \in \mathcal{S} \subseteq \mathcal{X}^{\leq \ell}$ is a token prefix sequence and $x \in \mathcal{X}$ is the next token. Then, after $T$ recursive updates, the final prediction error is bounded with probability at least $1 - \delta$:*

$$\text{Err}(\mathcal{G}_{\theta_T}^{LLM}) \lesssim \epsilon(\gamma) + \left(\frac{1}{\tau} + \frac{1}{NC_0} \log \frac{1}{\tau}\right) \frac{\log(n\tilde{B})\tilde{D}}{\sqrt{n}} \log \frac{1}{\delta} + \left(\frac{1}{N} + \left(\frac{1}{NC_0} \log \frac{1}{C_0}\right)^T \frac{C_\gamma}{N}\right) \log \frac{1}{\tau}.$$

*Furthermore, if $N > \frac{1}{C_0} \log\left(\frac{1}{C_0}\right)$, the exponential term vanishes as $T \to \infty$, yielding:*

$$\text{Err}(\mathcal{G}_{\theta_T}^{LLM}) \lesssim \epsilon(\gamma) + \left(\frac{1}{\tau} + \frac{1}{NC_0} \log \frac{1}{\tau}\right) \frac{\log(n\tilde{B})\tilde{D}}{\sqrt{n}} \log \frac{1}{\delta} + \frac{1}{N} \log \frac{1}{\tau},$$

*where $n = m\ell$ is the total number of training samples, and $\tilde{D}, \tilde{B}$ reflect the model capacity and prior radius respectively. A complete version of the theorem is presented as Theorem 5 in Appendix H.*

**Remark 7. Stabilizing Recursive Training of LLMs via Verification Mechanisms**. This theorem establishes that self-verification mechanisms, when incorporated into Transformer-based LLMs, effectively eliminate the exponential error accumulation typically observed in recursive synthetic training loops. Through rigorous analysis of the error bound, we identify three principal components governing model performance: (i) the confidence-calibrated agreement error $\epsilon(\gamma)$, which can be minimized when the verification slack parameter $\gamma$ is sufficiently small, leveraging established confidence calibration methods [Liu et al., 2025]; (ii) the statistical generalization gap, characterized by $\log(n\tilde{B})\tilde{D}/\sqrt{n}$, where $\tilde{B}$ and $\tilde{D}$ represent the prior radius and effective dimensionality of the model capacity—parameters that directly scale with the architectural complexity of Transformer networks; and (iii) a recursive decay term determined by verification sampling intensity.

Notably, when the verification sample size exceeds the critical threshold $N > \log(1/C_0)/C_0$, the decay term asymptotically vanishes, enabling stable convergence of the training process regardless of recursion depth. This result not only provides theoretical guarantees for the stability of recursive training language models but also offers practical guidance for calibrating verification intensity based on architectural complexity and confidence characteristics of specific Transformer implementations.

# 6 Conslusion

This paper establishes a theoretical foundation for understanding and mitigating model collapse in recursive synthetic training. We demonstrate that naive recursive training leads to exponential error growth, whereas incorporating a sufficient fraction of real data can stabilize performance. To address scenarios where real data is scarce or unavailable, we propose a self-verification mechanism that enables models to filter their own outputs based on internal confidence scores. Our analysis provides finite-sample error bounds showing that self-verification alone is sufficient to prevent collapse. We further extend these guarantees to Transformer-based LLMs, offering the first formal justification for stable, fully synthetic training in high-capacity generative models. Overall, our framework paves the way for principled and scalable self-evolution in future generations of LLMs without reliance on external supervision.

## Acknowledgement

This project is supported by the National Research Foundation, Singapore, under its NRF Professorship Award No. NRF-P2024-001.

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

# Appendix

## A  Overview

In this supplementary material, we provide additional details and complete proofs to support the theoretical developments presented in the main paper. The appendix is organized as follows:

- **Appendix B. Related Work: Self-Verification in LLMs.**
  We situate our work within the emerging literature on self-verification for LLMs, highlighting key differences between our theoretically grounded, training-time framework and prior prompt-based or post hoc empirical approaches.

- **Appendix C. Limitations.**
  We discuss a primary limitation of our framework—its reliance on a single verification strategy based on internal confidence scores—and outline possible extensions using richer or more adaptive verification methods.

- **Appendix D. Broader Impacts.**
  We outline the potential positive societal implications of our work, such as reducing dependence on human-labeled data through self-verification, and note that our theoretical analysis poses no immediate negative societal risks.

- **Appendix E. Proof of Theorem 3: Error Bound with Self-Verification.**
  We present a detailed finite-sample analysis of recursive training with self-verification. The proof introduces recursive coverage coefficients to quantify verification quality and demonstrates how confidence-guided filtering prevents error amplification across generations.

- **Appendix F. Proof of Theorem 1: Error Amplification in Naive Recursive Training.**
  We formalize the failure mode of naive recursive training without any verification and prove that, under mild conditions, prediction error grows exponentially with the number of generations.

- **Appendix G. Proof of Theorem 2: Stabilization via Real Data Injection.**
  We analyze how introducing a fixed proportion of real data at each generation mitigates exponential error growth, and derive sufficient conditions under which training remains stable.

- **Appendix H. Proof of Theorem 4: Theoretical Guarantees for Transformer-Based LLMs with Self-Verification.**
  We extend our theoretical framework to Transformer-based language models, providing generalization bounds under recursive training with self-verification, accounting for model capacity and confidence calibration.

- **Appendix I. Exponential Error Growth in Naive Recursive Training of Transformer-Based LLMs.**
  We instantiate Theorem 1 in the Transformer setting and analyze how parameter confidence bounds affect the rate of error amplification during unverified recursive training.

- **Appendix J. Mitigating Collapse via Real Data in Recursive Training of Transformer-Based LLMs.**
  We apply Theorem 2 to the Transformer case, showing that even limited real data injection suffices to stabilize recursive training and suppress error accumulation in practical scenarios.

- **Appendix K. Additional Experimental Analysis and Future Work.**
  We provide further empirical validation for our assumptions, present experimental results demonstrating how self-verification prevents model collapse, and discuss potential directions for future work.

This appendix provides the full mathematical foundations of our results and demonstrates their applicability to modern high-capacity models used in practice.

## B  Related Work: Self-Verification in LLMs

Recent research has explored self-verification as a means of improving LLM reasoning performance during inference [Weng et al., 2023, Stechly et al., 2024, Ma et al., 2025, Lightman et al., 2023, Xu et al., 2024, Kamoi et al., 2024, Tao et al., 2024]. Most existing approaches adopt a prompt-based paradigm, where LLMs are instructed at inference time to critique or revise their own outputs [Huang et al., 2023, Tyen et al., 2023, Zhang et al., 2024a], or are trained post hoc to mimic such behavior via curated demonstrations or reinforcement learning [Saunders et al., 2022, Jiang et al., 2024, Kumar et al., 2024, Zhang et al., 2024b, Ma et al., 2025]. While effective in certain settings, these methods largely focus on empirical improvements, and rely on externally crafted prompts or behavior cloning pipelines.

In contrast to these empirical efforts, our work develops a formal theoretical framework for self-verification in recursive training settings. Rather than prompting models to verify at inference time, we integrate self-verification directly into the training process via internal confidence-based filtering. This allows us to establish provable error bounds, showing that self-verification alone can prevent collapse under fully synthetic training. Furthermore, we extend these guarantees to Transformer-based LLMs, providing the first formal justification of stable self-evolution in high-capacity models without external supervision. Our contribution bridges a gap in the literature by grounding the concept of self-verification in statistical learning theory, rather than relying on prompt design or behavioral imitation.

## C  Limitations

A primary limitation of our framework lies in its reliance on a single verification strategy—confidence-based filtering using a fixed base model. While effective in our theoretical setting, this approach may be insufficient to capture semantic correctness in more complex tasks. Future work should explore richer and more adaptive verification methods, such as ensemble-based voting or task-specific evaluators, to enhance robustness.

## D  Broader Impacts

This work provides a theoretical foundation for stable and data-efficient training of language models through self-verification, potentially reducing the reliance on large-scale human-labeled datasets and enabling broader access to capable models. As a theoretical contribution, it does not pose any immediate negative societal impacts.

## E  Proof of Theorem 3: Error Bound with Self-Verification

In this section, we present the proof of Theorem 3, which provides a finite-sample error bound for recursive training with self-verification. Our analysis draws inspiration from the confidence framework introduced in Huang et al. [2025a], but extends it to a more general recursive setting that captures the dynamics of synthetic supervision over multiple generations.

The key insight lies in controlling the evolution of the *recursive coverage coefficient*, which measures how well the model concentrates probability mass on its own verified predictions across generations. While classical generalization bounds typically depend only on data size and model capacity, recursive training introduces a new challenge: maintaining reliable supervision when the training data is generated by the model itself. To formalize this, we define the recursive coverage coefficient as

$$C_{\gamma,t} := \mathbb{E}_{x \sim \mathcal{X}} \left[ \frac{1}{\mathcal{G}_t(\boldsymbol{y}_\gamma^+(x) \mid x)} \right],$$

which captures the expected inverse confidence over the high-quality, self-verified region. A smaller value of $C_{\gamma,t}$ implies that the model assigns higher likelihood to its verified outputs, indicating better internal consistency and semantic alignment.

By bounding the growth of this coefficient over generations, we are able to characterize how self-verification mitigates error amplification and ensures stability, even under fully synthetic training.

The proof proceeds by decomposing failure events and recursively controlling the probability mass over verified subsets.

*Proof of Theorem 3.* We begin by recalling that the initial model $\mathcal{G}_0$ is trained via maximum likelihood estimation on a ground-truth dataset $S_0 = \{(x_j, y_j^*)\}_{j=1}^n$, that is,

$$\mathcal{G}_0 = \arg\max_{\mathcal{G} \in \mathcal{H}} \sum_{j=1}^n \log \mathcal{G}(y_j^* \mid x_j),$$

where $\mathcal{H}$ denotes the model hypothesis class. The resulting model $\mathcal{G}_0$ encodes a set of inductive biases aligned with human-annotated supervision, and serves as a fixed and calibrated reference model in the recursive training pipeline.

**Defining Confident Support Set $\mathcal{D}_1^+$**

We define the subset of inputs with sufficient sample coverage as

$$\mathcal{D}_1^+ := \left\{ x \in \mathcal{X} \,\middle|\, \mathcal{G}_0\left(\boldsymbol{y}_\gamma^+(x) \mid x\right) \geq \frac{\log \frac{2}{\tau}}{N} \right\},$$

where the high-quality candidate set $\boldsymbol{y}_\gamma^+(x)$ is defined based on the internal self-verification score function:

$$\boldsymbol{y}_\gamma^+ = \left\{ y \,\middle|\, s_{\text{verify}}(y \mid x) \geq \max_{y' \in \mathcal{Y}} s_{\text{verify}}(y' \mid x) - \gamma \right\},$$

with self-verification score function defined as $s_{\text{verify}}(y \mid x) := \log \mathcal{G}_0(y \mid x)$. This gives:

$$\begin{aligned}
\boldsymbol{y}_\gamma^+ &= \left\{ y \,\middle|\, \log \mathcal{G}_0(y \mid x) \geq \max_{y' \in \mathcal{Y}} \log \mathcal{G}_0(y' \mid x) - \gamma \right\} \\
&= \left\{ y \,\middle|\, \mathcal{G}_0(y \mid x) \geq \max_{y' \in \mathcal{Y}} \gamma' \mathcal{G}_0(y' \mid x) \right\},
\end{aligned} \tag{3}$$

where $\gamma' = e^{-\gamma}$.

**Decomposing the Probability of Low Confidence Generation**

We aim to bound:

$$\begin{aligned}
&\mathbb{P}_{x \sim \mathcal{X}} \left[ \mathcal{G}_1(\boldsymbol{y}_\gamma^+(x) \mid x) \leq 1 - \tau \right] \\
&\leq \mathbb{P}_{x \sim \mathcal{X}} \left[ \mathcal{G}_1(\boldsymbol{y}_\gamma^+(x) \mid x) \leq 1 - \tau, x \in \mathcal{D}_1^+ \right] + \mathbb{P}_{x \sim \mathcal{X}} \left[ x \notin \mathcal{D}_1^+ \right].
\end{aligned} \tag{4}$$

**Bounding the First Term: Using Monte Carlo Estimate**

For $x \in \mathcal{D}_1^+$, we aim to show

$$\mathcal{G}_{1,N}^*(\boldsymbol{y}_\gamma^+(x) \mid x) \geq 1 - \tau/2.$$

Indeed, observe that $y \sim \mathcal{G}_{1,N}^*(\cdot \mid x) \notin \boldsymbol{y}_\gamma^+(x)$ if and only if $y_{1,1}, \ldots, y_{1,N} \sim \mathcal{G}_0(\cdot)$ have $y_{1,j} \notin \boldsymbol{y}_\gamma^+(x)$ for all $j$, since $x \in \mathcal{D}_1^+$ which happens with probability

$$(1 - \mathcal{G}_0(\boldsymbol{y}_\gamma^+(x) \mid x))^N \leq (1 - \frac{\log \frac{2}{\tau}}{N})^N. \tag{5}$$

Then we show that:

$$(1 - \frac{\log \frac{2}{\tau}}{N})^N \leq \frac{\tau}{2},$$

To do so, we analyze its asymptotic behavior, and verify that it satisfies the above inequality. This is now a typical exponential approximation problem, where the base $\left(1 - \frac{1}{N}\right)^N$ plays a key role. We notice that:

$$\lim_{N \to \infty} \left(1 - \frac{1}{N}\right)^N = \frac{1}{e},$$

Thus, we get

$$\left(1 - \frac{\log \frac{2}{\tau}}{N}\right)^N = \left[\left(1 - \frac{\log \frac{2}{\tau}}{N}\right)^{\frac{N}{\log \frac{2}{\tau}}}\right]^{\log \frac{2}{\tau}} \le \left(\frac{1}{e}\right)^{\log \frac{2}{\tau}} = \frac{\tau}{2}. \tag{6}$$

Then, we obtain

$$(1 - \mathcal{G}_0(\boldsymbol{y}_\gamma^+(x) \mid x))^N \le \frac{\tau}{2},$$

Thus, we get

$$\mathcal{G}_{1,N}^*(\boldsymbol{y}_\gamma^+(x) \mid x) \ge 1 - \tau/2.$$

**Bounding Discrepancy via Total Variation Distance**

The total variation (TV) distance between two probability measures $P$ and $Q$ is defined as

$$D_{\mathrm{TV}}(P, Q) = \frac{1}{2} \sum_y |P(y) - Q(y)|.$$

Importantly, the TV distance provides a direct measure of the pointwise probability differences between $P$ and $Q$. In particular, for any measurable event $A$, the following inequality holds:

$$|P(A) - Q(A)| \le 2D_{\mathrm{TV}}(P, Q).$$

Specializing to singleton events $\{y\}$, it follows that

$$\left|\mathcal{G}_1(y \mid x) - \mathcal{G}_{1,N}^*(y \mid x)\right| \le 2D_{\mathrm{TV}}\left(\mathcal{G}_1(\cdot \mid x), \mathcal{G}_{1,N}^*(\cdot \mid x)\right).$$

Equivalently, we can lower bound the TV distance by

$$D_{\mathrm{TV}}\left(\mathcal{G}_1(\cdot \mid x), \mathcal{G}_{1,N}^*(\cdot \mid x)\right) \ge \frac{1}{2}\left|\mathcal{G}_1(\boldsymbol{y}_\gamma^+(x) \mid x) - \mathcal{G}_{1,N}^*(\boldsymbol{y}_\gamma^+(x) \mid x)\right|.$$

From the definition of the set $\mathcal{D}_1^+$, we know that for any $x \in \mathcal{D}_1^+$,

$$\mathcal{G}_{1,N}^*(\boldsymbol{y}_\gamma^+(x) \mid x) \ge 1 - \frac{\tau}{2}.$$

Suppose that $\mathcal{G}_1(\boldsymbol{y}_\gamma^+(x) \mid x) \le 1 - \tau$. Then it follows that

$$\left|\mathcal{G}_1(\boldsymbol{y}_\gamma^+ \mid x) - \mathcal{G}_{1,N}^*(\boldsymbol{y}_\gamma^+ \mid x)\right| \ge \tau - \frac{\tau}{2} = \frac{\tau}{2}.$$

Substituting back into the previous inequality, we conclude that for all $x \in \mathcal{D}_1^+$,

$$D_{\mathrm{TV}}\left(\mathcal{G}_1(\cdot \mid x), \mathcal{G}_{1,N}^*(\cdot \mid x)\right) \ge \frac{\tau}{4} \cdot \mathbb{I}\left\{\mathcal{G}_1(\boldsymbol{y}_\gamma^+(x) \mid x) \le 1 - \tau\right\}.$$

Thus, we get:

$$\mathbb{P}_{x \sim \mathcal{X}}\left[\mathcal{G}_1(\boldsymbol{y}_\gamma^+(x) \mid x) \le 1 - \tau, x \in \mathcal{D}_1^+\right] \le \frac{4}{\tau} D_{\mathrm{TV}}\left(\mathcal{G}_1(\cdot \mid x), \mathcal{G}_{1,N}^*(\cdot \mid x)\right) \tag{7}$$

**Bounding the Second Term: Using Markov's Inequality**

Let $C_\gamma := \mathbb{E}_{x \sim \mathcal{X}}\left[\frac{1}{\mathcal{G}_0\left(\boldsymbol{y}_\gamma^+(x) \mid x\right)}\right]$. Then:

$$\begin{aligned}
\mathbb{P}_{x \sim \mathcal{X}}[x \notin \mathcal{D}_1^+] &= \mathbb{P}_{x \sim \mathcal{X}}\left[\mathcal{G}_0\left(\boldsymbol{y}_\gamma^+(x) \mid x\right) < \frac{\log \frac{2}{\tau}}{N}\right] \\
&= \mathbb{P}_{x \sim \mathcal{X}}\left[\frac{\log \frac{2}{\tau}}{N\mathcal{G}_0\left(\boldsymbol{y}_\gamma^+(x) \mid x\right)} > 1\right] \\
&\le \frac{\log \frac{2}{\tau}}{N} \mathbb{E}_{x \sim \mathcal{X}}\left[\frac{1}{\mathcal{G}_0\left(\boldsymbol{y}_\gamma^+(x) \mid x\right)}\right] \\
&\le \frac{C_\gamma}{N} \log \frac{2}{\tau}. \tag{8}
\end{aligned}$$

**Final Bound for $\mathcal{G}_1$**

Substituting equalities 7 and 8 into 4, we obtain

$$\mathbb{P}_{x\sim\mathcal{X}}\left[\mathcal{G}_1(\boldsymbol{y}_\gamma^+(x)\mid x)\leq 1-\tau\right]\leq\frac{4}{\tau}D_{\mathrm{TV}}\left(\mathcal{G}_1(\cdot\mid x),\mathcal{G}_{1,N}^*(\cdot\mid x)\right)+\frac{C_\gamma}{N}\log\frac{2}{\tau}. \tag{9}$$

**Recursive Bound for the Second Generation Model $\mathcal{G}_2$**

Building upon the prior analysis, we now investigate how a self-verification mechanism can be systematically employed to refine model training across generations. Specifically, we focus on filtering the outputs generated by $\mathcal{G}_1$ through confidence-aware criteria, and subsequently utilizing only the verified outputs to train the next-generation model $\mathcal{G}_2$. This recursive training framework ensures that each successive model benefits from high-quality and semantically coherent supervision.

**Definition of Verified Input Set for $\mathcal{G}_2$**

We define the subset of inputs with sufficient verified coverage as:

$$\mathcal{D}_2^+:=\left\{x\in\mathcal{X}\;\middle|\;\mathcal{G}_1\left(\boldsymbol{y}_\gamma^+(x)\mid x\right)\geq\frac{\log\frac{2}{\tau}}{N}\right\},$$

where the high-quality candidate set $\boldsymbol{y}_\gamma^+(x)$ is defined based on the internal verification score function:

$$\boldsymbol{y}_\gamma^+=\left\{y\;\middle|\;s_{\mathrm{verify}}(y\mid x)\geq\max_{y'\in\mathcal{Y}}s_{\mathrm{verify}}(y'\mid x)-\gamma\right\}.$$

**Bounding the Low-Confidence Region for $\mathcal{G}_2$**

Following the same logic as in Eq. (7), we derive the following bound on the probability that $\mathcal{G}_2$ assigns low confidence to the verified candidate region:

$$\mathbb{P}_{x\sim\mathcal{X}}\left[\mathcal{G}_2(\boldsymbol{y}_\gamma^+(x)\mid x)\leq 1-\tau,x\in\mathcal{D}_2^+\right]\leq\frac{4}{\tau}D_{\mathrm{TV}}\left(\mathcal{G}_2(\cdot\mid x),\mathcal{G}_{2,N}^*(\cdot\mid x)\right) \tag{10}$$

**Bounding the Failure Rate of Verified Samples**

Similarly, we follow the derivation of Eq. (8) to obtain an upper bound on the probability that an input falls outside the region $\mathcal{D}_2^+$:

$$\begin{aligned}\mathbb{P}_{x\sim\mathcal{X}}[x\notin\mathcal{D}_2^+]&=\mathbb{P}_{x\sim\mathcal{X}}\left[\mathcal{G}_1\left(\boldsymbol{y}_\gamma^+(x)\mid x\right)<\frac{\log\frac{2}{\tau}}{N}\right]\\&=\mathbb{P}_{x\sim\mathcal{X}}\left[\frac{\log\frac{2}{\tau}}{N\mathcal{G}_1\left(\boldsymbol{y}_\gamma^+(x)\mid x\right)}>1\right]\\&\leq\frac{\log\frac{2}{\tau}}{N}\mathbb{E}_{x\sim\mathcal{X}}\left[\frac{1}{\mathcal{G}_1\left(\boldsymbol{y}_\gamma^+(x)\mid x\right)}\right]\end{aligned} \tag{11}$$

The quantity $\mathbb{E}_{x\sim\mathcal{X}}\left[\frac{1}{\mathcal{G}_1\left(\boldsymbol{y}_\gamma^+(x)\mid x\right)}\right]$ captures the expected inverse confidence in the high-quality region across all inputs, and plays a central role in the recursion.

To proceed, we introduce the variable:

$$Z(x)=\frac{1}{\mathcal{G}_1\left(\boldsymbol{y}_\gamma^+(x)\mid x\right)}.$$

Its expectation is computed via the tail-integral identity for nonnegative random variables:

$$\mathbb{E}_{x\sim\mathcal{X}}[Z(x)]=\int_0^\infty\mathbb{P}(Z(x)\geq t)\,\mathrm{d}t. \tag{12}$$

Since the posterior satisfies $C_0 \leq \mathcal{G}_1 \left( \boldsymbol{y}_\gamma^+(x) \mid x \right) \leq 1$, it follows that

$$1 \leq Z(x) = \frac{1}{\mathcal{G}_1 \left( \boldsymbol{y}_\gamma^+(x) \mid x \right)} \leq \frac{1}{C_0}.$$

Hence,

$$\mathbb{P}\big(Z(x) \geq t\big) = \begin{cases} 1, & 0 \leq t < 1, \\ 0, & t > \frac{1}{C_0}. \end{cases}$$

Thus the integral in (12) reduces to

$$\mathbb{E}\big[Z(x)\big] = \int_0^1 1 \, dt + \int_1^{\frac{1}{C_0}} \mathbb{P}\big(Z(x) \geq t\big) \, dt$$

$$= 1 + \int_1^{\frac{1}{C_0}} \mathbb{P}\big(Z(x) \geq t\big) \, dt.$$

Recall that Eq (9), for $t \geq 1$, then we can deduce that:

$$\mathbb{P}_{x \sim \mathcal{X}}\left[Z(x) \geq t\right] \leq \frac{4t}{t-1} D_{\mathrm{TV}}\left(\mathcal{G}_1(\cdot \mid x), \mathcal{G}_{1,N}^*(\cdot \mid x)\right) + \frac{C_\gamma}{N} \log \frac{2t}{t-1}$$

The rest of the proof proceeds by upper bounding this integral through a sharp relaxation using $B(t)$, characterizing tail probabilities of inverse confidence values, and leading to a recursive generalization bound.

**Explicit Upper Bound on Recursive Confidence**

We define:

$$B(t) = \frac{4t}{t-1} D_{\mathrm{TV}}\left(\mathcal{G}_1(\cdot \mid x), \mathcal{G}_{1,N}^*(\cdot \mid x)\right) + \frac{C_\gamma}{N} \log \frac{2t}{t-1}.$$

Then we Upper bound for the tail probability via $\min\{1, B(t)\}$. However, probabilities cannot exceed 1. Hence a tighter statement is

$$\mathbb{P}(Z(x) \geq t) \leq \min\{1, B(t)\}.$$

and therefore

$$\mathbb{E}\big[Z(x)\big] = 1 + \int_1^{\frac{1}{C_0}} \mathbb{P}\big(Z(x) \geq t\big) dt \leq 1 + \int_1^{\frac{1}{C_0}} \min\{1, B(t)\} \, dt.$$

Since $B(t)$ is monotonically decreasing for $t \in (1^+, +\infty)$, we focus on the non-trivial case where $B\left(\frac{1}{C_0}\right) < 1$. In this setting, there exists a *unique* point $t_0 \in \left(1, \frac{1}{C_0}\right)$ such that

$$B(t_0) = 1 \quad \text{and} \quad B(t) \begin{cases} \geq 1, & 1 \leq t \leq t_0, \\ \leq 1, & t_0 \leq t \leq \frac{1}{C_0}. \end{cases}$$

Consequently,

$$\min\{1, B(t)\} = \begin{cases} 1, & t \in [1, t_0], \\ B(t), & t \in [t_0, 1/C_0]. \end{cases}$$

Hence,

$$\int_1^{\frac{1}{C_0}} \min\{1, B(t)\} \, dt = \int_1^{t_0} 1 \, dt + \int_{t_0}^{\frac{1}{C_0}} B(t) \, dt = (t_0 - 1) + \int_{t_0}^{\frac{1}{C_0}} B(t) \, dt. \quad (13)$$

Then, we compute $\int B(t) \, dt$ explicitly. Let us denote

$$B(t) = A \frac{t}{t-1} + C \log\left(\frac{2t}{t-1}\right),$$

where
$$A = 4D_{\text{TV}}\left(\mathcal{G}_1(\cdot \mid x), \mathcal{G}_{1,N}^*(\cdot \mid x)\right), \quad C = \frac{C_\gamma}{N}.$$

Then we decompose and integrate each part:

$$\int \frac{t}{t-1}\, dt = \int \left(1 + \frac{1}{t-1}\right) dt = t + \ln|t-1| + \text{ constant},$$

$$\int \log\left(\frac{2t}{t-1}\right) dt = \int \log(2t)\, dt - \int \log(t-1)\, dt$$
$$= \left[t\, \log(2t) - t\right] - \left[(t-1)\, \log(t-1) - (t-1)\right] + \text{ constant}$$
$$= t\, \log(2t) - (t-1)\, \log(t-1) + \text{ constant}.$$

Therefore,

$$\int B(t)\, dt = A\left[t + \ln|t-1|\right] + C\left[t\, \log(2t) - (t-1)\, \log(t-1)\right] + \text{ constant}.$$

We shall denote a particular antiderivative by $F(t)$. Then

$$\int_{t_0}^{\frac{1}{C_0}} B(t)\, dt = F\left(\frac{1}{C_0}\right) - F(t_0),$$

where $t_0$ is the unique solution to $B(t_0) = 1$. Then we conclude the final upper bound. Combining (13) with the preceding results, we see that whenever $B\left(\frac{1}{C_0}\right) < 1$, we obtain

$$\int_{1}^{\frac{1}{C_0}} \min\{1,\, B(t)\}\, dt = (t_0 - 1) + \left[F\left(\frac{1}{C_0}\right) - F(t_0)\right],$$

and thus

$$\mathbb{E}\left[Z(x)\right] = 1 + \int_{1}^{\frac{1}{C_0}} P\left(Z(x) \geq t\right) dt \leq 1 + (t_0 - 1) + \left[F\left(\frac{1}{C_0}\right) - F(t_0)\right].$$

Recalling that $Z(x) = 1/\mathcal{G}_1\left(\boldsymbol{y}_\gamma^+(x) \mid x\right)$, we finally arrive at

$$\mathbb{E}_{x \sim \mathcal{X}}\left[\frac{1}{\mathcal{G}_1\left(\boldsymbol{y}_\gamma^+(x) \mid x\right)}\right] \leq 1 + (t_0 - 1) + F\left(\frac{1}{C_0}\right) - F(t_0),$$

which provides an explicit upper bound on the coverage coefficient in the regime $B\left(\frac{1}{C_0}\right) < 1$. Here, $t_0 \in (1, \frac{1}{C_0})$ is the unique point where $B(t_0) = 1$, and $F(\cdot)$ is any antiderivative of $B(\cdot)$.

Then we further derive the explicit coverage coefficient upper bound. First evaluate $F\left(\frac{1}{C_0}\right) - F(t_0)$. From the explicit antiderivative, ignoring any additive constant (which cancels in the difference), we get

$$F\left(\frac{1}{C_0}\right) - F(t_0) = A\left[\left(\frac{1}{C_0} - t_0\right) + \ln\left(\frac{\frac{1}{C_0} - 1}{t_0 - 1}\right)\right]$$
$$+ C\left[\left(\frac{1}{C_0}\log\left(\frac{2}{C_0}\right) - t_0 \log(2t_0)\right) - \left(\left(\frac{1}{C_0} - 1\right)\log\left(\frac{1}{C_0} - 1\right) - (t_0 - 1)\log(t_0 - 1)\right)\right].$$

Thus,

$$(t_0 - 1) + F\left(\frac{1}{C_0}\right) - F(t_0) = (t_0 - 1) + A\left[\frac{1}{C_0} - t_0 + \ln\left(\frac{\frac{1}{C_0} - 1}{t_0 - 1}\right)\right] + C\left[\ldots\right],$$

where the bracket $[\ldots]$ in the $C$-term is the obvious difference from above.

Then we incorporate the condition $B(t_0) = 1$. Recall

$$B(t_0) = A\frac{t_0}{t_0 - 1} + C\log\left(\frac{2t_0}{t_0 - 1}\right) = 1.$$

This equation confines $t_0$ to the interval $(1, \frac{1}{C_0})$. Although $t_0$ generally cannot be expressed in an elementary closed form, we can appropriately scale the equation to obtain an approximate range for $t_0$.

Moreover, elementary inequalities for $\log t$ imply

$$\frac{t-1}{t} \;\leq\; \log t \;\leq\; t-1 \quad \text{for all } t > 0,$$

and setting $t = \frac{2t_0}{t_0 - 1}$ yields

$$\frac{1}{2} + \frac{1}{2t_0} \;\leq\; \log\Big(\frac{2t_0}{t_0 - 1}\Big) \;\leq\; 1 + \frac{2}{t_0 - 1}. \tag{14}$$

Hence, from $B(t_0) = 1$ we deduce two bracketing inequalities:

$$A \frac{t_0}{t_0 - 1} + C\Big(1 + \frac{2}{t_0 - 1}\Big) \;\geq\; 1, \tag{15}$$

$$A \frac{t_0}{t_0 - 1} + C\Big(\frac{1}{2} + \frac{1}{2t_0}\Big) \;\leq\; 1. \tag{16}$$

Next, we consider the non-trivial case that aligns more closely with practical scenarios, $A$ and $C$ are small enough that $A + C < 1$. Under that assumption, (15) and (16) is valid and yields an *upper* and *lower* bound on $t_0$. Thus

$$\frac{1 + \sqrt{1 - 2C + C^2 + 2AC}}{2 - (2A + C)} \leq t_0 \leq \frac{1 + C}{1 - (A + C)}.$$

Thus,

$$\begin{aligned}
t_0 - 1 &\geq \frac{1 + \sqrt{1 - 2C + C^2 + 2AC}}{2 - (2A + C)} - 1 \\
&\geq \frac{\sqrt{1 - 2C + C^2 + 2AC} - (1 - C) + 2A}{2 - (2A + C)} \\
&\geq \frac{2A}{2 - (2A + C)}.
\end{aligned} \tag{17}$$

Our goal is to convert these into simpler *numerical* bounds on $t_0$, which can then be substituted into the final coverage coefficient bound

$$C_{\gamma,1} \;=\; \mathbb{E}_{x\sim\mathcal{X}}\Big[\frac{1}{\mathcal{G}_1\big(\boldsymbol{y}_\gamma^+(x)\mid x\big)}\Big] \leq 1 + \big(t_0 - 1\big) + \big[F\big(\tfrac{1}{C_0}\big) - F(t_0)\big].$$

Thus, we get

$$\begin{aligned}
C_{\gamma,1} \;=\; &\mathbb{E}_{x\sim\mathcal{X}}\Big[\frac{1}{\mathcal{G}_1\big(\boldsymbol{y}_\gamma^+(x)\mid x\big)}\Big] \leq t_0 + \big[F\big(\tfrac{1}{C_0}\big) - F(t_0)\big]. \\
\leq\; &t_0 + A\Big[\Big(\frac{1}{C_0} - t_0\Big) + \ln\Big(\frac{\frac{1}{C_0} - 1}{t_0 - 1}\Big)\Big] \tag{18} \\
&+ C\Big[\Big(\frac{1}{C_0}\log(\tfrac{2}{C_0}) - t_0\log(2t_0)\Big) - \Big((\tfrac{1}{C_0} - 1)\log(\tfrac{1}{C_0} - 1) - (t_0 - 1)\log(t_0 - 1)\Big)\Big]. \\
\lesssim\; &t_0 + A\frac{1}{C_0}\log(\frac{1}{t_0 - 1}) + C\frac{1}{C_0}\log(\frac{1}{C_0}) \\
\lesssim\; &\frac{1 + C}{1 - (A + C)} + A\frac{1}{C_0}\log(\frac{2 - (2A + C)}{2A}) + C\frac{1}{C_0}\log(\frac{1}{C_0}) \\
\lesssim\; &1 + C + A\frac{1}{C_0}\log(\frac{1}{A}) + C\frac{1}{C_0}\log(\frac{1}{C_0}) \\
\lesssim\; &1 + A\frac{1}{C_0}\log(\frac{1}{A}) + C\frac{1}{C_0}\log(\frac{1}{C_0}). \tag{19}
\end{aligned}$$

**Final Bound for $\mathcal{G}_2$**

Combining the bounds from (10) and (11), we obtain:

$$\mathbb{P}_{x \sim \mathcal{X}} \left[ \mathcal{G}_2(\boldsymbol{y}_\gamma^+(x) \mid x) \leq 1 - \tau \right]$$

$$\leq \frac{4}{\tau} D_{\mathrm{TV}} \left( \mathcal{G}_2(\cdot \mid x), \mathcal{G}_{2,N}^*(\cdot \mid x) \right) + \frac{\log \frac{2}{\tau}}{N} \mathbb{E}_{x \sim \mathcal{X}} \left[ \frac{1}{\mathcal{G}_1 \left( \boldsymbol{y}_\gamma^+(x) \mid x \right)} \right]$$

$$\leq \frac{4}{\tau} D_{\mathrm{TV}} \left( \mathcal{G}_2(\cdot \mid x), \mathcal{G}_{2,N}^*(\cdot \mid x) \right) + \frac{C_{\gamma,1}}{N} \log \frac{2}{\tau}. \tag{20}$$

where $C_{\gamma,1}$ is the bound derived above. This completes the recursive error bound analysis for the second generation model.

**Recursive Generalization Bounds for Arbitrary Generation $t$**

We now extend our recursive analysis to arbitrary generation steps beyond $t = 2$. To simplify the discussion and maintain consistency, we make the following assumption:

**Assumption: Uniform Total Variation Distance**

Without loss of generality, we assume that the total variation distance across all generations $1 \leq t \leq T$ remains approximately the same, that is,

$$D_{\mathrm{TV}} \left( \mathcal{G}_t(\cdot \mid x), \mathcal{G}_{t,N}^*(\cdot \mid x) \right) \asymp D_{\mathrm{TV}}.$$

This allows us to write for the second generation:

$$A_2 = D_{\mathrm{TV}} \left( \mathcal{G}_2(\cdot \mid x), \mathcal{G}_{2,N}^*(\cdot \mid x) \right) \asymp D_{\mathrm{TV}},$$

$$C_2 = \frac{C_{\gamma,1}}{N} \lesssim \frac{1}{N} + \frac{1}{N} A \frac{1}{C_0} \log \left( \frac{1}{A} \right) + \frac{C}{N} \cdot \frac{1}{C_0} \log \left( \frac{1}{C_0} \right). \tag{21}$$

Continuing this recursion, we arrive at:

$$C_{\gamma,2} = \mathbb{E}_{x \sim \mathcal{X}} \left[ \frac{1}{\mathcal{G}_2 \left( \boldsymbol{y}_\gamma^+(x) \mid x \right)} \right] \lesssim 1 + A_2 \frac{1}{C_0} \log \left( \frac{1}{A_2} \right) + C_2 \frac{1}{C_0} \log \left( \frac{1}{C_0} \right) \tag{22}$$

leading to:

$$A_3 \asymp A_2, \tag{23}$$

$$C_3 = \frac{C_{\gamma,2}}{N} \lesssim \frac{1}{N} + \frac{1}{N} A_2 \frac{1}{C_0} \log \left( \frac{1}{A_2} \right) + \frac{C_2}{N} \cdot \frac{1}{C_0} \log \left( \frac{1}{C_0} \right). \tag{24}$$

This recurrence generalizes to all future generations:

$$A_t \asymp A, \tag{25}$$

$$C_t \lesssim \frac{1}{N} + \frac{1}{N} A \frac{1}{C_0} \log \left( \frac{1}{A} \right) + \frac{C_t}{N} \cdot \frac{1}{C_0} \log \left( \frac{1}{C_0} \right). \tag{26}$$

**Solving the Recurrence Relation**

Given the following definitions:

$$\alpha = \frac{1}{N C_0} \log \left( \frac{1}{C_0} \right), \quad \beta = \frac{1}{N} + \frac{A}{N C_0} \log \left( \frac{1}{A} \right),$$

we have the recurrence relation:

$$C_t \lesssim \beta + \alpha C_t,$$

which can be solved as:

$$C_t \lesssim (\alpha)^t C + \frac{\beta}{1 - \alpha} \left( 1 - (\alpha)^t \right).$$

Thus, we have:

$$C_t \lesssim \left(\frac{1}{NC_0} \log\left(\frac{1}{C_0}\right)\right)^t C + \frac{\frac{1}{N} + \frac{A}{NC_0} \log\left(\frac{1}{A}\right)}{1 - \frac{1}{NC_0} \log\left(\frac{1}{C_0}\right)} \left[1 - \left(\frac{1}{NC_0} \log\left(\frac{1}{C_0}\right)\right)^i\right].$$

Now recall $C = \frac{C_\gamma}{N}$. Under the assumption $N > \frac{1}{C_0} \log\left(\frac{1}{C_0}\right)$, we conclude:

$$C_t \leq \left(\frac{1}{NC_0} \log\left(\frac{1}{C_0}\right)\right)^t \frac{C_\gamma}{N} + \frac{1 + \frac{A}{C_0} \log\left(\frac{1}{A}\right)}{N - \frac{1}{C_0} \log\left(\frac{1}{C_0}\right)}. \tag{27}$$

Substituting this into the bound on $C_{\gamma,t}$, we get:

$$C_{\gamma,t} \lesssim 1 + A_t \frac{1}{C_0} \log\left(\frac{1}{A_t}\right) + C_t \frac{1}{C_0} \log\left(\frac{1}{C_0}\right)$$

$$\lesssim 1 + \frac{1}{C_0} D_{\text{TV}} \log\left(\frac{1}{D_{\text{TV}}}\right) \tag{28}$$

$$+ \left(\left(\frac{1}{NC_0} \log\left(\frac{1}{C_0}\right)\right)^t \frac{C_\gamma}{N} + \frac{1 + \frac{A}{C_0} \log\left(\frac{1}{A}\right)}{N - \frac{1}{C_0} \log\left(\frac{1}{C_0}\right)}\right) \cdot \frac{1}{C_0} \log\left(\frac{1}{C_0}\right). \tag{29}$$

**Final Error Bound After $T$ Recursive Generations**

Thus, after $T$ generations of the recursive training loop, we obtain

$$\mathbb{P}_{x \sim \mathcal{X}} \left[\mathcal{G}_T\left(\mathbf{y}_\gamma^+(x) \mid x\right) \leq 1 - \tau\right] \lesssim \frac{1}{\tau} \cdot D_{\text{TV}} + \frac{C_{\gamma,t}}{N} \log\left(\frac{1}{\tau}\right)$$

$$\lesssim \frac{1}{\tau} \cdot D_{\text{TV}} + \frac{1}{N} \log\left(\frac{1}{\tau}\right) + \frac{1}{NC_0} \log\left(\frac{1}{\tau}\right) D_{\text{TV}} \log\frac{1}{D_{\text{TV}}}$$

$$+ \left(\left(\frac{1}{NC_0} \log\left(\frac{1}{C_0}\right)\right)^t \frac{C_\gamma}{N} + \frac{1 + \frac{A}{C_0} \log\left(\frac{1}{A}\right)}{N - \frac{1}{C_0} \log\left(\frac{1}{C_0}\right)}\right) \frac{1}{C_0} \log\left(\frac{1}{C_0}\right) \frac{1}{N} \log\left(\frac{1}{\tau}\right)$$

$$\lesssim \frac{1}{\tau} \cdot D_{\text{TV}} + \frac{1}{N} \log\left(\frac{1}{\tau}\right) + \frac{1}{NC_0} \log\left(\frac{1}{\tau}\right) D_{\text{TV}} \log\frac{1}{D_{\text{TV}}}$$

$$+ \left(\left(\frac{1}{NC_0} \log\left(\frac{1}{C_0}\right)\right)^t \frac{C_\gamma}{N} + \frac{1}{C_0} \log\left(\frac{1}{C_0}\right) \frac{1}{N^2}\right) \log\left(\frac{1}{\tau}\right)$$

$$\lesssim \frac{1}{\tau} \cdot D_{\text{TV}} + \left(\frac{1}{N} + \frac{1}{NC_0} D_{\text{TV}} \log\frac{1}{D_{\text{TV}}} + \frac{1}{N^2} \frac{1}{C_0} \log\left(\frac{1}{C_0}\right)\right) \log\left(\frac{1}{\tau}\right)$$

$$+ \left(\frac{1}{NC_0} \log\left(\frac{1}{C_0}\right)\right)^t \frac{C_\gamma}{N} \log\left(\frac{1}{\tau}\right). \tag{30}$$

Furthermore, we obtain:

$$\mathbb{P}_{x \sim \mathcal{X}} \left[\mathcal{G}_T\left(\mathbf{y}_\gamma^+(x) \mid x\right) \geq 1 - \tau\right]$$

$$\gtrsim 1 - \frac{1}{\tau} \cdot D_{\text{TV}} - \left(\frac{1}{N} + \frac{1}{NC_0} D_{\text{TV}} \log\frac{1}{D_{\text{TV}}} + \frac{1}{N^2} \frac{1}{C_0} \log\left(\frac{1}{C_0}\right)\right) \log\left(\frac{1}{\tau}\right)$$

$$- \left(\frac{1}{NC_0} \log\left(\frac{1}{C_0}\right)\right)^t \frac{C_\gamma}{N} \log\left(\frac{1}{\tau}\right). \tag{31}$$

**Final Accuracy and Error Decomposition**

Define $y_T^+(x) = \arg\max_y \mathcal{G}_T(y \mid x)$. Then:

$$\text{Acc}(\mathcal{G}_T)$$
$$= \mathbb{P}_{x \sim \mathcal{X}}[y_T^+(x) = y^*(x)]$$
$$= \mathbb{P}_{x \sim \mathcal{X}}\left[y_T^+(x) = y^*(x) \mid \mathcal{G}_T(\boldsymbol{y}_\gamma^+(x) \mid x) \geq 1 - \tau\right] \cdot \mathbb{P}_{x \sim \mathcal{X}}\left[\mathcal{G}_T\left(\boldsymbol{y}_\gamma^+(x) \mid x\right) \geq 1 - \tau\right]$$
$$\gtrsim 1 - \epsilon(\gamma) - \frac{1 - \epsilon(\gamma)}{\tau} \cdot D_{\text{TV}} - \left(\frac{1}{NC_0}\log(\frac{1}{C_0})\right)^t \frac{C_\gamma(1 - \epsilon(\gamma))}{N}\log(\frac{1}{\tau})$$
$$- (1 - \epsilon(\gamma))\left(\frac{1}{N} + \frac{1}{NC_0}D_{\text{TV}}\log\frac{1}{D_{\text{TV}}} + \frac{1}{N^2}\frac{1}{C_0}\log(\frac{1}{C_0})\right)\log(\frac{1}{\tau}). \tag{32}$$

Furthermore, we get

$$\text{Err}(\mathcal{G}_T)$$
$$= \mathbb{P}_{x \sim \mathcal{X}}[y_T^+(x) \neq y^*(x)]$$
$$\lesssim \epsilon(\gamma) + \frac{1 - \epsilon(\gamma)}{\tau} \cdot D_{\text{TV}} + \left(\frac{1}{NC_0}\log(\frac{1}{C_0})\right)^t \frac{C_\gamma(1 - \epsilon(\gamma))}{N}\log(\frac{1}{\tau})$$
$$+ (1 - \epsilon(\gamma))\left(\frac{1}{N} + \frac{1}{NC_0}D_{\text{TV}}\log\frac{1}{D_{\text{TV}}} + \frac{1}{N^2}\frac{1}{C_0}\log(\frac{1}{C_0})\right)\log(\frac{1}{\tau})$$
$$\lesssim \frac{1}{\tau} \cdot D_{\text{TV}} + \left(\frac{1}{N} + \frac{1}{NC_0}D_{\text{TV}}\log\frac{1}{D_{\text{TV}}} + \frac{1}{N^2}\frac{1}{C_0}\log(\frac{1}{C_0})\right)\log(\frac{1}{\tau})$$
$$+ \left(\frac{1}{NC_0}\log(\frac{1}{C_0})\right)^t \frac{C_\gamma}{N}\log(\frac{1}{\tau}) + \epsilon(\gamma). \tag{33}$$

The proof is completed. $\qquad\square$

# F   Proof of Theorem 1: Error Amplification in Naive Recursive Training

In this section, we prove Theorem 1, which characterizes the failure mode of naive recursive training—where each model is trained solely on synthetic data generated by its predecessor, without any verification or quality control. The key result shows that prediction errors amplify exponentially across generations, ultimately leading to model collapse.

This phenomenon stems from the accumulation of distributional shifts at each generation, which progressively degrade supervision quality. Our analysis formalizes this effect by bounding the total error in terms of the cumulative divergence between successive models. Even when per-step errors are small, their unregulated propagation drives the model away from the original data distribution.

This negative result underscores the need for intervention—whether via real data, external feedback, or self-verification—to stabilize recursive training. We now present the detailed proof.

*Proof of Theorem 1.* We establish our results by carefully analyzing the propagation of errors in a recursive training paradigm without verification mechanisms. Our analytical approach parallels the methodology used in Theorem 3, with critical distinctions reflecting the absence of quality control measures.

**Foundational Definitions and Decomposition**

We begin by defining the subset of inputs with sufficient confidence under the base model:

$$\mathcal{D}_1^+ := \left\{x \in \mathcal{X} \;\middle|\; \mathcal{G}_0\left(\boldsymbol{y}_\gamma^+(x) \mid x\right) \geq 1 - \frac{\tau}{2}\right\},$$

where the high-quality candidate set $\boldsymbol{y}_\gamma^+(x)$ is defined via the verification score function:

$$\boldsymbol{y}_\gamma^+ = \left\{y \;\middle|\; s_{\text{verify}}(y \mid x) \geq \max_{y' \in \mathcal{Y}} s_{\text{verify}}(y' \mid x) - \gamma\right\},$$

with verification score function defined as $s_{\text{verify}}(y \mid x) := \log \mathcal{G}_0(y \mid x)$.

To establish our primary results, we decompose the probability of low-confidence generation:

$$\mathbb{P}_{x \sim \mathcal{X}} \left[ \mathcal{G}_1(y_\gamma^+(x) \mid x) \leq 1 - \tau \right]$$
$$\leq \mathbb{P}_{x \sim \mathcal{X}} \left[ \mathcal{G}_1(y_\gamma^+(x) \mid x) \leq 1 - \tau, x \in \mathcal{D}^+ \right] + \mathbb{P}_{x \sim \mathcal{X}} \left[ x \notin \mathcal{D}_1^+ \right]. \tag{34}$$

**Analysis of Model Divergence via Total Variation Distance**

To quantify the disagreement between consecutive model generations, we leverage the total variation distance. For any input $x$ and output $y$:

$$|\mathcal{G}_1(y \mid x) - \mathcal{G}_0(y \mid x)| \leq 2 D_{\text{TV}} \left( \mathcal{G}_1(\cdot \mid x), \mathcal{G}_0(\cdot \mid x) \right).$$

This inequality can be reformulated to establish a lower bound on the TV distance:

$$D_{\text{TV}} \left( \mathcal{G}_1(\cdot \mid x), \mathcal{G}_0(\cdot \mid x) \right) \geq \frac{1}{2} \left| \mathcal{G}_1(\boldsymbol{y}_\gamma^+(x) \mid x) - \mathcal{G}_0(\boldsymbol{y}_\gamma^+(x) \mid x) \right|.$$

For inputs in the high-confidence region $\mathcal{D}_1^+$, we have:

$$\mathcal{G}_0 \left( \boldsymbol{y}_\gamma^+(x) \mid x \right) \geq 1 - \frac{\tau}{2}.$$

When $\mathcal{G}_1$ assigns insufficient confidence to the optimal region, i.e., $\mathcal{G}_1(\boldsymbol{y}_\gamma^+(x) \mid x) \leq 1 - \tau$, the discrepancy between models is bounded below:

$$\left| \mathcal{G}_1(\boldsymbol{y}_\gamma^+(x) \mid x) - \mathcal{G}_0 \left( \boldsymbol{y}_\gamma^+(x) \mid x \right) \right| \geq \tau - \frac{\tau}{2} = \frac{\tau}{2}.$$

This allows us to express the probability of low confidence for inputs in $\mathcal{D}_1^+$ in terms of the TV distance:

$$\mathbb{P}_{x \sim \mathcal{X}} \left[ \mathcal{G}_1(\boldsymbol{y}_\gamma^+(x) \mid x) \leq 1 - \tau, x \in \mathcal{D}_1^+ \right] \leq \frac{4}{\tau} D_{\text{TV}} \left( \mathcal{G}_1(\cdot \mid x), \mathcal{G}_0(\cdot \mid x) \right). \tag{35}$$

**Characterizing the Complement Set via Inverse Confidence**

Let $C_\gamma := \mathbb{E}_{x \sim \mathcal{X}} \left[ \frac{1}{\mathcal{G}_0(\boldsymbol{y}_\gamma^+(x) \mid x)} \right]$ denote the expected inverse confidence under the base model. Applying Markov's inequality, we derive:

$$\mathbb{P}_{x \sim \mathcal{X}}[x \notin \mathcal{D}_1^+] = \mathbb{P}_{x \sim \mathcal{X}} \left[ \mathcal{G}_0 \left( \boldsymbol{y}_\gamma^+(x) \mid x \right) < 1 - \frac{\tau}{2} \right]$$
$$= \mathbb{P}_{x \sim \mathcal{X}} \left[ \frac{2 - \tau}{2 \mathcal{G}_0 \left( \boldsymbol{y}_\gamma^+(x) \mid x \right)} > 1 \right]$$
$$\leq \left( 1 - \frac{\tau}{2} \right) \mathbb{E}_{x \sim \mathcal{X}} \left[ \frac{1}{\mathcal{G}_0 \left( \boldsymbol{y}_\gamma^+(x) \mid x \right)} \right]$$
$$\leq \left( 1 - \frac{\tau}{2} \right) C_\gamma. \tag{36}$$

Integrating equations (35) and (36) into (34), we establish the comprehensive bound for the first generation:

$$\mathbb{P}_{x \sim \mathcal{X}} \left[ \mathcal{G}_1(\boldsymbol{y}_\gamma^+(x) \mid x) \leq 1 - \tau \right] \leq \frac{4}{\tau} D_{\text{TV}} \left( \mathcal{G}_1(\cdot \mid x), \mathcal{G}_0(\cdot \mid x) \right) + \left( 1 - \frac{\tau}{2} \right) C_\gamma. \tag{37}$$

**Recursive Analysis for the Second Generation Model $\mathcal{G}_2$**

Following an analogous approach, we define the high-confidence region for the second generation:

$$\mathcal{D}_2^+ := \left\{ x \in \mathcal{X} \mid \mathcal{G}_1 \left( \boldsymbol{y}_\gamma^+(x) \mid x \right) \geq 1 - \frac{\tau}{2} \right\},$$

By symmetric reasoning, we obtain:

$$\mathbb{P}_{x\sim\mathcal{X}}\left[\mathcal{G}_2(\boldsymbol{y}_\gamma^+(x)\mid x)\leq 1-\tau, x\in\mathcal{D}_2^+\right]\leq\frac{4}{\tau}D_{\text{TV}}\left(\mathcal{G}_2(\cdot\mid x),\mathcal{G}_1(\cdot\mid x)\right). \tag{38}$$

For the complement set:

$$\begin{aligned}
\mathbb{P}_{x\sim\mathcal{X}}[x\notin\mathcal{D}_2^+]&=\mathbb{P}_{x\sim\mathcal{X}}\left[\mathcal{G}_1\left(\boldsymbol{y}_\gamma^+(x)\mid x\right)<1-\frac{\tau}{2}\right]\\
&\leq\left(1-\frac{\tau}{2}\right)\mathbb{E}_{x\sim\mathcal{X}}\left[\frac{1}{\mathcal{G}_1\left(\boldsymbol{y}_\gamma^+(x)\mid x\right)}\right].
\end{aligned} \tag{39}$$

**Advanced Analysis of Expected Inverse Confidence**

The expected inverse confidence $\mathbb{E}_{x\sim\mathcal{X}}\left[\frac{1}{\mathcal{G}_1(\boldsymbol{y}_\gamma^+(x)\mid x)}\right]$ represents a critical quantity for understanding error propagation across generations. We analyze this quantity by introducing the random variable:

$$Z(x)=\frac{1}{\mathcal{G}_1\left(\boldsymbol{y}_\gamma^+(x)\mid x\right)}.$$

Utilizing the tail-integral identity for non-negative random variables:

$$\begin{aligned}
\mathbb{E}\big[Z(x)\big]&=\int_0^\infty\mathbb{P}\big(Z(x)\geq t\big)\,\mathrm{d}t\\
&=\int_0^1 1\,\mathrm{d}t+\int_1^{\frac{1}{C_0}}\mathbb{P}\big(Z(x)\geq t\big)\,\mathrm{d}t\\
&=1+\int_1^{\frac{1}{C_0}}\mathbb{P}\big(Z(x)\geq t\big)\,\mathrm{d}t.
\end{aligned}$$

From equation (37), for $t\geq 1$, we derive:

$$\mathbb{P}_{x\sim\mathcal{X}}\left[Z(x)\geq t\right]\leq\frac{4t}{t-1}D_{\text{TV}}\left(\mathcal{G}_1(\cdot\mid x),\mathcal{G}_0(\cdot\mid x)\right)+\left(\frac{1}{2}+\frac{1}{2t}\right)C_\gamma.$$

**Threshold-based Analysis of Inverse Confidence Distribution**

We define the bounding function:

$$B(t)=\frac{4t}{t-1}D_{\text{TV}}\left(\mathcal{G}_1(\cdot\mid x),\mathcal{G}_0(\cdot\mid x)\right)+\left(\frac{1}{2}+\frac{1}{2t}\right)C_\gamma.$$

Since $B(t)$ is monotonically decreasing for $t>1$, we focus on the non-trivial case where $B\left(\frac{1}{C_0}\right)<1$, then there exists a unique threshold $t_0\in\left(1,\frac{1}{C_0}\right)$ such that:

$$B(t_0)=1\quad\text{and}\quad B(t)\begin{cases}\geq 1,&1\leq t\leq t_0,\\\leq 1,&t_0\leq t\leq\frac{1}{C_0}.\end{cases}$$

This threshold partitioning enables us to refine our expectation bound:

$$\mathbb{E}\big[Z(x)\big]\leq 1+(t_0-1)+\int_{t_0}^{\frac{1}{C_0}}B(t)\,\mathrm{d}t.$$

**Explicit Evaluation of the Integral Term**

For analytical tractability, we represent $B(t)$ in the form:

$$B(t) = A\frac{t}{t-1} + C\frac{1}{t} + \frac{C_\gamma}{2},$$

where:

$$A = 4D_{\text{TV}}\left(\mathcal{G}_1(\cdot \mid x), \mathcal{G}_0(\cdot \mid x)\right), \quad C = \frac{C_\gamma}{2}.$$

This decomposition yields:

$$\mathbb{E}\big[Z(x)\big] \le t_0 + \frac{C_\gamma}{2}\left(\frac{1}{C_0} - t_0\right) + \int_{t_0}^{\frac{1}{C_0}}\left(A\frac{t}{t-1} + C\frac{1}{t}\right)\,\mathrm{d}t.$$

Through careful integration and algebraic manipulation:

$$\int_{t_0}^{\frac{1}{C_0}} A\cdot\frac{t}{t-1} + C\cdot\frac{1}{t}\,\mathrm{d}t = A\left(\frac{1}{C_0} - t_0 + \log\left|\frac{\frac{1}{C_0}-1}{t_0-1}\right|\right) + C\log\left(\frac{1}{C_0 t_0}\right)$$

$$\lesssim A\left(\frac{1}{C_0} + \log\frac{1}{t_0-1}\right) + C\log\left(\frac{1}{C_0}\right).$$

**Determining the Threshold Value $t_0$**

The threshold $t_0$ is characterized by the equation $B(t_0) = 1$. Under the reasonable assumption that $C_\gamma + A < 2$, we can express $t_0$ explicitly:

$$t_0 = \frac{\sqrt{(C_\gamma - 1)^2 + 2AC_\gamma} + 1}{2 - (C_\gamma + 2A)}. \tag{40}$$

This allows us to derive the lower bound:

$$t_0 - 1 \ge \frac{2A}{2 - (2A + C_\gamma)}. \tag{41}$$

**Analytical Expression for Expected Inverse Confidence**

After substituting the threshold expressions and simplifying, we arrive at:

$$\mathbb{E}_{x\sim\mathcal{X}}\left[\frac{1}{\mathcal{G}_1\left(\boldsymbol{y}_\gamma^+(x) \mid x\right)}\right] \le \frac{\frac{1}{2}C_\gamma + A}{C_0} + A\log\frac{1}{A} + \frac{C_\gamma}{2}\log\frac{1}{C_0}. \tag{42}$$

**Consolidated Bound for $\mathcal{G}_2$**

Denoting $C_{\gamma,1} = \mathbb{E}_{x\sim\mathcal{X}}\left[\frac{1}{\mathcal{G}_1\left(\boldsymbol{y}_\gamma^+(x)\mid x\right)}\right]$ and combining equations (38) and (39), we establish:

$$\mathbb{P}_{x\sim\mathcal{X}}\left[\mathcal{G}_2(\boldsymbol{y}_\gamma^+(x) \mid x) \le 1 - \tau\right] \le \frac{4}{\tau}D_{\text{TV}}\left(\mathcal{G}_2(\cdot \mid x), \mathcal{G}_1(\cdot \mid x)\right) + \left(1 - \frac{\tau}{2}\right)C_{\gamma,1}. \tag{43}$$

**Generalization to Arbitrary Generations**

To extend our analysis beyond the second generation, we assume uniform distributional shift across generations:

$$D_{\text{TV}}\left(\mathcal{G}_t(\cdot \mid x), \mathcal{G}_{t-1}(\cdot \mid x)\right) \asymp D_{\text{TV}},$$

This assumption enables us to derive the recursive relation:

$$C_{\gamma,t} \le \frac{C_{\gamma,t-1} + A}{C_0} + A\log\frac{1}{A} + C_{\gamma,t-1}\log\frac{1}{C_0}$$

$$\le \left(\frac{1}{C_0} + \log\frac{1}{C_0}\right)\frac{C_{\gamma,t-1}}{2} + A\log\frac{1}{A}. \tag{44}$$

**Solution to the Recursive Relation**

We reframe this as a standard first-order linear recurrence relation:

$$C_{\gamma,t} \leq \alpha C_{\gamma,t-1} + \beta,$$

where $\alpha := \frac{1}{2}\left(\frac{1}{C_0} + \log \frac{1}{C_0}\right) > 0$ and $\beta := A \log \frac{1}{A}$. Crucially, $\alpha > 1$, indicating exponential growth.

Unrolling this recurrence yields:

$$\begin{aligned}
C_{\gamma,t} &\leq \alpha C_{\gamma,t-1} + \beta \\
&\leq \alpha\left(\alpha C_{\gamma,t-2} + \beta\right) + \beta = \alpha^2 C_{\gamma,t-2} + \alpha\beta + \beta \\
&\leq \alpha^3 C_{\gamma,t-3} + \alpha^2\beta + \alpha\beta + \beta \\
&\vdots \\
&\leq \alpha^t C_\gamma + \beta \sum_{k=0}^{t-1} \alpha^k.
\end{aligned}$$

Applying the formula for finite geometric series:

$$\sum_{k=0}^{t-1} \alpha^k = \frac{1-\alpha^t}{1-\alpha}.$$

This gives:

$$C_{\gamma,t} \leq \alpha^t C_\gamma + \frac{\beta}{1-\alpha}(1-\alpha^t).$$

With simplification and substitution of our specific parameters:

$$C_{\gamma,t} \lesssim \left(C_\gamma + A \log \frac{1}{A}\right)\left(\frac{1}{2C_0} + \frac{1}{2}\log \frac{1}{C_0}\right)^t. \tag{45}$$

**Asymptotic Error Analysis After $T$ Generations**

Extending our analysis to the terminal generation $T$, we establish:

$$\mathbb{P}_{x \sim \mathcal{X}}\left[\mathcal{G}_T\left(\boldsymbol{y}_\gamma^+(x) \mid x\right) \leq 1-\tau\right] \lesssim \frac{1}{\tau} \cdot D_{\text{TV}} + \left(C_\gamma + D_{\text{TV}} \log \frac{1}{D_{\text{TV}}}\right)\left(\frac{1}{2C_0} + \frac{1}{2}\log \frac{1}{C_0}\right)^T. \tag{46}$$

Correspondingly, for the complementary event:

$$\mathbb{P}_{x \sim \mathcal{X}}\left[\mathcal{G}_T\left(\boldsymbol{y}_\gamma^+(x) \mid x\right) \geq 1-\tau\right] \gtrsim 1 - \frac{1}{\tau} \cdot D_{\text{TV}} - \left(C_\gamma + D_{\text{TV}} \log \frac{1}{D_{\text{TV}}}\right)\left(\frac{1}{2C_0} + \frac{1}{2}\log \frac{1}{C_0}\right)^T. \tag{47}$$

**Quantification of Model Accuracy and Error**

For the final model $\mathcal{G}_T$, we define $y_T^+(x) = \arg\max_y \mathcal{G}_T(y \mid x)$ as its preferred prediction. The accuracy is then:

$$\begin{aligned}
\text{Acc}(\mathcal{G}_T) &= \mathbb{P}_{x \sim \mathcal{X}}[y_T^+(x) = y^*(x)] \\
&= \mathbb{P}_{x \sim \mathcal{X}}\left[y_t^+(x) = y^*(x) \mid \mathcal{G}_t(\boldsymbol{y}_\gamma^+(x) \mid x) \geq 1-\tau\right] \cdot \mathbb{P}_{x \sim \mathcal{X}}\left[\mathcal{G}_T\left(\boldsymbol{y}_\gamma^+(x) \mid x\right) \geq 1-\tau\right] \\
&\gtrsim (1-\epsilon(\gamma)) \cdot \left(1 - \frac{1}{\tau} \cdot D_{\text{TV}} - \left(C_\gamma + D_{\text{TV}} \log \frac{1}{D_{\text{TV}}}\right)\left(\frac{1}{2C_0} + \frac{1}{2}\log \frac{1}{C_0}\right)^T\right) \\
&\gtrsim 1 - \epsilon(\gamma) - \frac{1-\epsilon(\gamma)}{\tau} \cdot D_{\text{TV}} - (1-\epsilon(\gamma))\left(C_\gamma + D_{\text{TV}} \log \frac{1}{D_{\text{TV}}}\right)\left(\frac{1}{2C_0} + \frac{1}{2}\log \frac{1}{C_0}\right)^T. 
\end{aligned} \tag{48}$$

The corresponding error rate is:

$$\mathrm{Err}(\mathcal{G}_T) = \mathbb{P}_{x \sim \mathcal{X}}[y_T^+(x) \neq y^*(x)]$$

$$\lesssim \epsilon(\gamma) + \frac{1}{\tau} \cdot D_{\mathrm{TV}} + (C_\gamma + D_{\mathrm{TV}} \log \frac{1}{D_{\mathrm{TV}}}) \left( \frac{1}{2C_0} + \frac{1}{2} \log \frac{1}{C_0} \right)^T . \tag{49}$$

The critical observation is that if $\frac{1}{2C_0} + \frac{1}{2} \log \frac{1}{C_0} > 1$, the error term grows exponentially with $T$, inexorably leading to model collapse as $T \to \infty$. This establishes the theorem. $\qquad\square$

# G Proof of Theorem 2: Stabilization via Real Data Injection

In this section, we prove Theorem 2, which establishes that incorporating a sufficient proportion of real data at each generation can prevent model collapse in recursive training. Unlike naive training that relies exclusively on synthetic data, this hybrid strategy anchors the training distribution to the ground-truth data manifold, thereby mitigating the accumulation of distributional shift across generations.

The proof shows that even modest amounts of real data are sufficient to dominate the recursive error dynamics and ensure bounded generalization error over time. This result provides a theoretical justification for widely adopted empirical practices that mix real and synthetic data to maintain model quality. We now present the formal derivation.

*Proof of Theorem 2.* We establish our results by analyzing the error propagation in a recursive training paradigm that incorporates a proportion $\alpha$ of real data at each generation. The remaining $1 - \alpha$ proportion of data is generated synthetically by the previous model.

**Foundational Definitions and Decomposition**

Let the training dataset for generation $t$ be a mixture of real data and synthetic data:

$$\mathcal{D}_t = \alpha \cdot \mathcal{D}_{\mathrm{real}} + (1 - \alpha) \cdot \mathcal{D}_{\mathrm{synthetic}},$$

where $\mathcal{D}_{\mathrm{real}} = \{(x, y^*) \mid x \sim \mathcal{X}, y^* = \text{ground-truth label}\}$ and $\mathcal{D}_{\mathrm{synthetic}} = \{(x, \hat{y}) \mid x \sim \mathcal{X}, \hat{y} = \mathcal{G}_{t-1}(x)\}$.

We define the mixed distribution as $\mathcal{G}_t^{\mathrm{mix}} = \alpha \mathcal{G}^* + (1 - \alpha)\mathcal{G}_{t-1}$, where $\mathcal{G}^*$ represents the ground truth real data distribution.

We define the subset of inputs with sufficient real-data confidence under the base model $\mathcal{G}_0$:

$$\mathcal{D}_t^+ := \left\{ x \in \mathcal{X} \,\middle|\, \mathcal{G}_0\left(y^* \mid x\right) \geq 1 - \frac{\tau}{2(1-\alpha)} \right\},$$

where $y^*$ is the ground-truth label, and $\tau$ is the confidence threshold.

The probability of low-confidence generation at generation $t$ can be decomposed as:

$$\mathbb{P}_{x \sim \mathcal{X}}\left[\mathcal{G}_t(y^* \mid x) \leq 1 - \tau\right] \leq \mathbb{P}_{x \sim \mathcal{X}}\left[\mathcal{G}_t(y^* \mid x) \leq 1 - \tau, x \in \mathcal{D}_t^+\right]$$
$$+ \mathbb{P}_{x \sim \mathcal{X}}\left[x \notin \mathcal{D}_t^+\right]. \tag{50}$$

**Analysis of Model Divergence via Total Variation Distance**

To quantify the disagreement between consecutive model generations, we leverage the total variation distance. For any input $x$ and output $y$:

$$\left|\mathcal{G}_1(y \mid x) - \mathcal{G}_1^{\mathrm{mix}}(y \mid x)\right| \leq 2D_{\mathrm{TV}}\left(\mathcal{G}_1(\cdot \mid x), \mathcal{G}_1^{\mathrm{mix}}(\cdot \mid x)\right).$$

This inequality can be reformulated to establish a lower bound on the TV distance:

$$D_{\mathrm{TV}}\left(\mathcal{G}_1(\cdot \mid x), \mathcal{G}_1^{\mathrm{mix}}(\cdot \mid x)\right) \geq \frac{1}{2}\left|\mathcal{G}_1(\boldsymbol{y}_\gamma^+(x) \mid x) - \mathcal{G}_1^{\mathrm{mix}}(\boldsymbol{y}_\gamma^+(x) \mid x)\right|.$$

For inputs in the high-confidence region $\mathcal{D}_1^+$, we have:

$$\mathcal{G}_0 \left( \boldsymbol{y}_\gamma^+(x) \mid x \right) \geq 1 - \frac{\tau}{2(1-\alpha)}.$$

When $\mathcal{G}_1$ assigns insufficient confidence to the optimal region, i.e., $\mathcal{G}_1(\boldsymbol{y}_\gamma^+(x) \mid x) \leq 1 - \tau$, the discrepancy between models is bounded below:

$$\left| \mathcal{G}_1(\boldsymbol{y}_\gamma^+(x) \mid x) - \mathcal{G}_1^{\text{mix}} \left( \boldsymbol{y}_\gamma^+(x) \mid x \right) \right| \geq \alpha + (1-\alpha) \left( 1 - \frac{\tau}{2(1-\alpha)} \right) - (1-\tau) = \frac{\tau}{2}.$$

This allows us to express the probability of low confidence for inputs in $\mathcal{D}_1^+$ in terms of the TV distance:

$$\mathbb{P}_{x \sim \mathcal{X}} \left[ \mathcal{G}_1(\boldsymbol{y}_\gamma^+(x) \mid x) \leq 1 - \tau, x \in \mathcal{D}_1^+ \right] \leq \frac{4}{\tau} D_{\text{TV}} \left( \mathcal{G}_1(\cdot \mid x), \mathcal{G}_1^{\text{mix}}(\cdot \mid x) \right). \tag{51}$$

**Characterizing the Complement Set via Inverse Confidence**

Let $C_\gamma := \mathbb{E}_{x \sim \mathcal{X}} \left[ \frac{1}{\mathcal{G}_0\left(\boldsymbol{y}_\gamma^+(x)|x\right)} \right]$ denote the expected inverse confidence under the base model. Applying Markov's inequality, we derive:

$$\begin{aligned}
\mathbb{P}_{x \sim \mathcal{X}}[x \notin \mathcal{D}_1^+] &= \mathbb{P}_{x \sim \mathcal{X}} \left[ \mathcal{G}_0 \left( \boldsymbol{y}_\gamma^+(x) \mid x \right) < 1 - \frac{\tau}{2(1-\alpha)} \right] \\
&= \mathbb{P}_{x \sim \mathcal{X}} \left[ \left( 1 - \frac{\tau}{2(1-\alpha)} \right) \frac{1}{\mathcal{G}_0 \left( \boldsymbol{y}_\gamma^+(x) \mid x \right)} > 1 \right] \\
&\leq \left( 1 - \frac{\tau}{2(1-\alpha)} \right) \mathbb{E}_{x \sim \mathcal{X}} \left[ \frac{1}{\mathcal{G}_0 \left( \boldsymbol{y}_\gamma^+(x) \mid x \right)} \right] \\
&\leq \left( 1 - \frac{\tau}{2(1-\alpha)} \right) C_\gamma.
\end{aligned} \tag{52}$$

Integrating equations (51) and (52) into (50), we establish the comprehensive bound for the first generation:

$$\mathbb{P}_{x \sim \mathcal{X}} \left[ \mathcal{G}_1(\boldsymbol{y}_\gamma^+(x) \mid x) \leq 1 - \tau \right] \leq \frac{4}{\tau} D_{\text{TV}} \left( \mathcal{G}_1(\cdot \mid x), \mathcal{G}_1^{\text{mix}}(\cdot \mid x) \right) + \left( 1 - \frac{\tau}{2(1-\alpha)} \right) C_\gamma. \tag{53}$$

**Recursive Analysis for the Second Generation Model $\mathcal{G}_2$**

Following an analogous approach, we define the high-confidence region for the second generation:

$$\mathcal{D}_2^+ := \left\{ x \in \mathcal{X} \,\middle|\, \mathcal{G}_1 \left( \boldsymbol{y}_\gamma^+(x) \mid x \right) \geq 1 - \frac{\tau}{2(1-\alpha)} \right\},$$

Then, we obtain:

$$\mathbb{P}_{x \sim \mathcal{X}} \left[ \mathcal{G}_2(\boldsymbol{y}_\gamma^+(x) \mid x) \leq 1 - \tau, x \in \mathcal{D}_2^+ \right] \leq \frac{4}{\tau} D_{\text{TV}} \left( \mathcal{G}_2(\cdot \mid x), \mathcal{G}_2^{\text{mix}}(\cdot \mid x) \right). \tag{54}$$

For the complement set:

$$\begin{aligned}
\mathbb{P}_{x \sim \mathcal{X}}[x \notin \mathcal{D}_2^+] &= \mathbb{P}_{x \sim \mathcal{X}} \left[ \mathcal{G}_1 \left( \boldsymbol{y}_\gamma^+(x) \mid x \right) < 1 - \frac{\tau}{2(1-\alpha)} \right] \\
&\leq \left( 1 - \frac{\tau}{2(1-\alpha)} \right) \mathbb{E}_{x \sim \mathcal{X}} \left[ \frac{1}{\mathcal{G}_1 \left( \boldsymbol{y}_\gamma^+(x) \mid x \right)} \right].
\end{aligned} \tag{55}$$

## Advanced Analysis of Expected Inverse Confidence

The expected inverse confidence $\mathbb{E}_{x \sim \mathcal{X}}\left[\frac{1}{\mathcal{G}_1\left(\boldsymbol{y}_\gamma^+(x)|x\right)}\right]$ represents a critical quantity for understanding error propagation across generations. We analyze this quantity by introducing the random variable:

$$Z(x) = \frac{1}{\mathcal{G}_1\left(\boldsymbol{y}_\gamma^+(x) \mid x\right)}.$$

Utilizing the tail-integral identity for non-negative random variables:

$$
\begin{aligned}
\mathbb{E}\left[Z(x)\right] &= \int_0^\infty \mathbb{P}\left(Z(x) \geq t\right) \mathrm{d}t \\
&= \int_0^1 1 \, \mathrm{d}t + \int_1^{\frac{1}{C_0}} \mathbb{P}\left(Z(x) \geq t\right) \mathrm{d}t \\
&= 1 + \int_1^{\frac{1}{C_0}} \mathbb{P}\left(Z(x) \geq t\right) \mathrm{d}t.
\end{aligned}
$$

From equation (53), for $t \geq 1$, we derive:

$$\mathbb{P}_{x \sim \mathcal{X}}\left[Z(x) \geq t\right] \leq \frac{4t}{t-1} D_{\mathrm{TV}}\left(\mathcal{G}_1(\cdot \mid x), \mathcal{G}_1^{\mathrm{mix}}(\cdot \mid x)\right) + \left(1 - \frac{1}{2(1-\alpha)} + \frac{1}{2(1-\alpha)t}\right) C_\gamma.$$

## Threshold-based Analysis of Inverse Confidence Distribution

We define the bounding function:

$$B(t) = \frac{4t}{t-1} D_{\mathrm{TV}}\left(\mathcal{G}_1(\cdot \mid x), \mathcal{G}_1^{\mathrm{mix}}(\cdot \mid x)\right) + \left(1 - \frac{1}{2(1-\alpha)} + \frac{1}{2(1-\alpha)t}\right) C_\gamma.$$

Since $B(t)$ is monotonically decreasing for $t > 1$, we focus on the non-trivial case where $B\left(\frac{1}{C_0}\right) < 1$, then there exists a unique threshold $t_0 \in \left(1, \frac{1}{C_0}\right)$ such that:

$$B(t_0) = 1 \quad \text{and} \quad B(t) \begin{cases} \geq 1, & 1 \leq t \leq t_0, \\ \leq 1, & t_0 \leq t \leq \frac{1}{C_0}. \end{cases}$$

This threshold partitioning enables us to refine our expectation bound:

$$\mathbb{E}\left[Z(x)\right] \leq 1 + (t_0 - 1) + \int_{t_0}^{\frac{1}{C_0}} B(t) \, \mathrm{d}t.$$

## Explicit Evaluation of the Integral Term

For analytical tractability, we represent $B(t)$ in the form:

$$B(t) = A \frac{t}{t-1} + C \frac{1}{t} + \left(1 - \frac{1}{2(1-\alpha)}\right) C_\gamma,$$

where:

$$A = 4 D_{\mathrm{TV}}\left(\mathcal{G}_1(\cdot \mid x), \mathcal{G}_1^{\mathrm{mix}}(\cdot \mid x)\right), \quad C = \frac{C_\gamma}{2(1-\alpha)}.$$

This decomposition yields:

$$\mathbb{E}\left[Z(x)\right] \leq t_0 + \left(1 - \frac{1}{2(1-\alpha)}\right) C_\gamma(\frac{1}{C_0} - t_0) + \int_{t_0}^{\frac{1}{C_0}} \left(A \frac{t}{t-1} + C \frac{1}{t}\right) \mathrm{d}t.$$

Through careful integration and algebraic manipulation:

$$\int_{t_0}^{\frac{1}{C_0}} A \cdot \frac{t}{t-1} + C \cdot \frac{1}{t} \, dt = A \left( \frac{1}{C_0} - t_0 + \log \left| \frac{\frac{1}{C_0} - 1}{t_0 - 1} \right| \right) + C \log \left( \frac{1}{C_0 t_0} \right)$$

$$\lesssim A \left( \frac{1}{C_0} + \log \frac{1}{t_0 - 1} \right) + C \log \left( \frac{1}{C_0} \right).$$

### Determining the Threshold Value $t_0$

The threshold $t_0$ is characterized by the equation $B(t_0) = 1$. Then, we can express $t_0$ explicitly:

$$t_0 - 1 = \frac{2A + C_\gamma - 1 + \sqrt{\left( \frac{C_\gamma \alpha}{1-\alpha} + 1 \right)^2 - 4 \left( -A - C_\gamma + \frac{C_\gamma}{2(1-\alpha)} + 1 \right) \cdot \frac{C_\gamma}{2(1-\alpha)}}}{-2A - 2C_\gamma + \frac{C_\gamma}{1-\alpha} + 2}$$

$$= \frac{2A + C_\gamma - 1 + \sqrt{\left( \frac{C_\gamma \alpha}{1-\alpha} + 1 \right)^2 + \left( 2A + 2C_\gamma - \frac{C_\gamma}{1-\alpha} - 2 \right) \cdot \frac{C_\gamma}{(1-\alpha)}}}{-2A - 2C_\gamma + \frac{C_\gamma}{1-\alpha} + 2}. \qquad (56)$$

This allows us to derive the lower bound:

$$t_0 - 1 \geq \frac{2A}{2 - (2A + C_\gamma)}. \qquad (57)$$

### Analytical Expression for Expected Inverse Confidence

Let $A = D_{\text{TV}} \left( \mathcal{G}_1(\cdot \mid x), \mathcal{G}_1^{\text{mix}}(\cdot \mid x) \right)$. After substituting the threshold expressions and simplifying, we arrive at:

$$\mathbb{E}_{x \sim \mathcal{X}} \left[ \frac{1}{\mathcal{G}_1 \left( \boldsymbol{y}_\gamma^+(x) \mid x \right)} \right] \leq 1 + \int_1^{\frac{1}{C_0}} \frac{4t}{t-1} A + \left( 1 - \frac{1}{2(1-\alpha)} + \frac{1}{2(1-\alpha)t} \right) C_\gamma dt$$

$$\lesssim \left( 1 - \frac{1}{2(1-\alpha)} \right) C_\gamma \frac{1}{C_0} + \frac{A}{C_0} + A \log \frac{1}{A} + \frac{C_\gamma}{2(1-\alpha)} \log \frac{1}{C_0}. \qquad (58)$$

### Consolidated Bound for $\mathcal{G}_2$

Denoting $C_{\gamma,1} = \mathbb{E}_{x \sim \mathcal{X}} \left[ \frac{1}{\mathcal{G}_1 \left( \boldsymbol{y}_\gamma^+(x) \mid x \right)} \right]$ and combining equations (38) and (39), we establish:

$$\mathbb{P}_{x \sim \mathcal{X}} \left[ \mathcal{G}_2(\boldsymbol{y}_\gamma^+(x) \mid x) \leq 1 - \tau \right] \leq \frac{4}{\tau} D_{\text{TV}} \left( \mathcal{G}_2(\cdot \mid x), \mathcal{G}_1(\cdot \mid x) \right) + \left( 1 - \frac{\tau}{2(1-\alpha)} \right) C_{\gamma,1}. \qquad (59)$$

### Generalization to Arbitrary Generations

To extend our analysis beyond the second generation, we assume uniform distributional shift across generations:

$$D_{\text{TV}} \left( \mathcal{G}_t(\cdot \mid x), \mathcal{G}_{t-1}(\cdot \mid x) \right) \asymp D_{\text{TV}},$$

This assumption enables us to derive the recursive relation:

$$C_{\gamma,t} \leq \frac{C_{\gamma,t-1} + A}{C_0} + A \log \frac{1}{A} + \frac{C_{\gamma,t-1}}{1-\alpha} \log \frac{1}{C_0}$$

$$\leq \left( \left( 1 - \frac{1}{2(1-\alpha)} \right) \frac{1}{C_0} + \frac{1}{2(1-\alpha)} \log \frac{1}{C_0} \right) C_{\gamma,t-1} + A \log \frac{1}{A}. \qquad (60)$$

Assume $\alpha > 1 - \frac{1}{2(\frac{1}{C_0} - 1)}$, $T \to \infty$, then we obtain:

$$C_{\gamma,T} \lesssim \frac{1-\alpha}{C_0} A \log \frac{1}{A} \qquad (61)$$

**Asymptotic Error Analysis After $T$ Generations**

Extending our analysis to the terminal generation $T$, we establish:

$$\mathbb{P}_{x\sim\mathcal{X}}\left[\mathcal{G}_T\left(\boldsymbol{y}_\gamma^+(x)\mid x\right)\leq 1-\tau\right]\lesssim\frac{1}{\tau}\cdot D_{\mathrm{TV}}+\frac{1-\alpha}{C_0}D_{\mathrm{TV}}\log\frac{1}{D_{\mathrm{TV}}}. \tag{62}$$

**Quantification of Model Accuracy and Error**

For the final model $\mathcal{G}_T$, we define $y_T^+(x)=\arg\max_y\mathcal{G}_T(y\mid x)$ as its preferred prediction. The accuracy is then:

$$\begin{aligned}
\mathrm{Acc}(\mathcal{G}_T)&=\mathbb{P}_{x\sim\mathcal{X}}[y_T^+(x)=y^*(x)]\\
&=\mathbb{P}_{x\sim\mathcal{X}}\left[y_t^+(x)=y^*(x)\mid\mathcal{G}_t(\boldsymbol{y}_\gamma^+(x)\mid x)\geq 1-\tau\right]\cdot\mathbb{P}_{x\sim\mathcal{X}}\left[\mathcal{G}_T\left(\boldsymbol{y}_\gamma^+(x)\mid x\right)\geq 1-\tau\right]\\
&\gtrsim(1-\epsilon(\gamma))\cdot\left(1-\frac{1}{\tau}\cdot D_{\mathrm{TV}}-\frac{1-\alpha}{C_0}D_{\mathrm{TV}}\log\frac{1}{D_{\mathrm{TV}}}\right)\\
&\gtrsim 1-\epsilon(\gamma)-\frac{1-\epsilon(\gamma)}{\tau}\cdot D_{\mathrm{TV}}-(1-\epsilon(\gamma))\frac{1-\alpha}{C_0}D_{\mathrm{TV}}\log\frac{1}{D_{\mathrm{TV}}}.
\end{aligned} \tag{63}$$

The corresponding error rate is:

$$\begin{aligned}
\mathrm{Err}(\mathcal{G}_T)&=\mathbb{P}_{x\sim\mathcal{X}}[y_T^+(x)\neq y^*(x)]\\
&\lesssim\epsilon(\gamma)+\frac{1}{\tau}\cdot D_{\mathrm{TV}}+\frac{1-\alpha}{C_0}D_{\mathrm{TV}}\log\frac{1}{D_{\mathrm{TV}}}.
\end{aligned} \tag{64}$$

**Conclusion**

The inclusion of real data at each generation mitigates error accumulation, ensuring that the model maintains stability over recursive generations. This completes the proof. $\qquad\square$

# H   Proof of Theorem 4: Theoretical Guarantees for Transformer-Based LLMs with Self-Verification

In this section, we prove Theorem 4, which extends our self-verification framework to autoregressive Transformer-based LLMs. This result establishes that, even in high-capacity architectures, self-verification can effectively prevent error amplification across recursive training generations.

To enable this analysis, we assume that the Transformer model parameters are uniformly bounded, a standard condition in prior theoretical studies [Li et al., 2023, Hu et al., 2024, Zhang et al., 2025, Fu et al., 2025] as follows:

**Assumption 2** (Bounded Parameters [Zhang et al., 2025])**.** The parameter space $\Theta$ of the Transformer model is constrained as:

$$\Theta=\left\{\theta\;\middle|\;\left\|A^{(D+1),\top}\right\|_{1,2}\leq B_A,\;\max\left\{|\gamma_1^{(t)}|,|\gamma_2^{(t)}|\right\}\leq 1,\;\left\|A_1^{(t)}\right\|_F\leq B_{A,1},\quad\left\|A_2^{(t)}\right\|_F\leq B_{A,2},\right.$$

$$\left.\left\|W_{Q,i}^{(t)}\right\|_F\leq B_Q,\quad\left\|W_{K,i}^{(t)}\right\|_F\leq B_K,\quad\left\|W_{V,i}^{(t)}\right\|_F\leq B_V\text{ for all }t\in[D],\;i\in[h]\right\},$$

where $h$ denotes the number of attention heads, and $B_A,B_{A,1},B_{A,2},B_Q,B_K,B_V>1$ are pre-specified constants.

Under this assumption, we then present the complete theorem statement characterizing the final test error after $T$ recursive steps.

**Theorem 5** (Error Bound for Transformer-based LLMs with Self-Verification)**.** *Let $\mathcal{G}_{\theta_t}^{LLM}(x\mid S)$ denote the conditional generation distribution of a Transformer-based LLM at the $t$-th recursive generation, where $S\in\mathcal{S}\subseteq\mathcal{X}^{\leq\ell}$ is a token prefix sequence and $x\in\mathcal{X}$ is the next token. Then, after*

*T recursive updates, the final prediction error is bounded with probability at least $1 - \delta$:*

$$\text{Err}(\mathcal{G}_{\theta_T}^{LLM}) \lesssim \epsilon(\gamma) + \left(\frac{1}{\tau} + \frac{1}{NC_0}\log\left(\frac{1}{\tau}\right)\right)\sqrt{\frac{1}{n}\log(n\tilde{B})\tilde{D}\log\frac{1}{\delta}}$$

$$+ \left(\frac{1}{N} + \frac{1}{N^2 C_0}\right)\log\left(\frac{1}{\tau}\right) + \left(\frac{1}{NC_0}\log\left(\frac{1}{C_0}\right)\right)^T \frac{C_\gamma}{N}\log\left(\frac{1}{\tau}\right). \quad (65)$$

*Furthermore, if $N > \frac{1}{C_0}\log\left(\frac{1}{C_0}\right)$, then as $T \to \infty$, the exponential decay term vanishes, and we obtain the bound:*

$$\text{Err}(\mathcal{G}_{\theta_T}^{LLM}) \lesssim \epsilon(\gamma) + \left(\frac{1}{\tau} + \frac{1}{NC_0}\log\frac{1}{\tau}\right)\frac{\log(n\tilde{B})\tilde{D}}{\sqrt{n}}\log\frac{1}{\delta} + \frac{1}{N}\log\frac{1}{\tau},$$

*where $n = m\ell$ is the total number of training samples, and $\tilde{D} = D^2 \cdot d \cdot (d_F + d_h + d) + d \cdot d_y$ and $\tilde{B} = \beta^{-1}RhB_A B_{A,1} B_{A,2} B_Q B_K B_V$ reflect the model capacity and prior radius respectively.*

Our proof draws inspiration from the PAC-Bayesian analysis developed in Zhang et al. [2025], but introduces new techniques to handle recursive training dynamics. In particular, we leverage a recursive coverage coefficient to track how the model's confidence evolves across generations, allowing us to rigorously control generalization error under fully synthetic training. We now present the formal derivation.

*Proof of Theorem 5.* We present the proof as follows.

### MLE Initialization and Sequence-Based Setup

We assume that all token prefix sequences $S_t = (x_1, \ldots, x_t)$ are elements of a structured sequence space $\mathcal{S} \subseteq \mathcal{X}^{\leq \ell}$, where $\mathcal{X}$ denotes the token vocabulary and $\ell$ is the maximum sequence length. This ensures that all context representations during both training and generation reside within a bounded and well-defined domain.

The initial autoregressive language model $\mathcal{G}_{\theta_0}^{\text{LLM}}$ is trained via maximum likelihood estimation (MLE) on a corpus of sequences $\{(x_1^{(j)}, \ldots, x_\ell^{(j)})\}_{j=1}^m$. The training objective is given by:

$$\theta_0 = \arg\max_{\theta \in \Theta_{\text{LLM}}} \frac{1}{n}\sum_{j=1}^m \sum_{t=1}^\ell \log \mathcal{G}_\theta^{\text{LLM}}(x_{t+1}^{(j)} \mid S_t^{(j)}),$$

where $S_t^{(j)} = (x_1^{(j)}, \ldots, x_t^{(j)}) \in \mathcal{S}$ and $n = m\ell$ is the total number of prefix-conditioned samples. The resulting model $\mathcal{G}_{\theta_0}^{\text{LLM}}$ serves as a fixed reference distribution encoding inductive biases from human supervision.

### Defining the Confident Support Set $\mathcal{S}_1^+$

To identify reliable contexts for self-verification, we define the verification score function as:

$$s_{\text{verify}}(x \mid S) := \log \mathcal{G}_{\theta_0}^{\text{LLM}}(x \mid S),$$

and the corresponding high-quality candidate set:

$$\boldsymbol{x}_\gamma^+(S) := \left\{ x \in \mathcal{X} \;\middle|\; s_{\text{verify}}(x \mid S) \geq \max_{x' \in \mathcal{X}} s_{\text{verify}}(x' \mid S) - \gamma \right\}.$$

We then define the confident support set $\mathcal{S}_1^+$ as the subset of prefixes where the reference model assigns sufficient mass to the high-reward region:

$$\mathcal{S}_1^+ := \left\{ S \in \mathcal{S} \;\middle|\; \mathcal{G}_{\theta_0}^{\text{LLM}}(\boldsymbol{x}_\gamma^+(S) \mid S) \geq \frac{\log\frac{2}{\tau}}{N} \right\}.$$

**Decomposing the Probability of Low-Confidence Generation**

We aim to bound the probability that the updated model assigns low probability mass to verified candidates:

$$\mathbb{P}_{S \sim \mathcal{S}} \left[ \mathcal{G}_{\theta_1}^{\text{LLM}}(\boldsymbol{x}_\gamma^+(S) \mid S) \leq 1 - \tau \right] \leq \mathbb{P}_{S \sim \mathcal{S}} \left[ \mathcal{G}_{\theta_1}^{\text{LLM}}(\boldsymbol{x}_\gamma^+(S) \mid S) \leq 1 - \tau, \ S \in \mathcal{S}_1^+ \right]$$
$$+ \ \mathbb{P}_{S \sim \mathcal{S}} \left[ S \notin \mathcal{S}_1^+ \right]. \tag{66}$$

**Bounding the First Term: Monte Carlo Estimate for Verified Support**

Let $\mathcal{G}_{1,N}^*$ be the empirical estimator from $N$ samples drawn from $\mathcal{G}_{\theta_0}^{\text{LLM}}$. For $S \in \mathcal{S}_1^+$, the probability that none of the samples falls into $\boldsymbol{x}_\gamma^+(S)$ is at most:

$$\left( 1 - \mathcal{G}_{\theta_0}^{\text{LLM}}(\boldsymbol{x}_\gamma^+(S) \mid S) \right)^N \leq \left( 1 - \frac{\log \frac{2}{\tau}}{N} \right)^N \leq \frac{\tau}{2}.$$

Hence,

$$\mathcal{G}_{1,N}^*(\boldsymbol{x}_\gamma^+(S) \mid S) \geq 1 - \frac{\tau}{2}.$$

**Bounding Discrepancy via Total Variation Distance**

The total variation distance between two distributions $P$ and $Q$ over $\mathcal{X}$ is:

$$D_{\text{TV}}(P, Q) = \frac{1}{2} \sum_{x \in \mathcal{X}} |P(x) - Q(x)|.$$

For any event $A \subseteq \mathcal{X}$, it satisfies:

$$|P(A) - Q(A)| \leq 2 D_{\text{TV}}(P, Q).$$

In particular, if $\mathcal{G}_{\theta_1}^{\text{LLM}}(\boldsymbol{x}_\gamma^+(S) \mid S) \leq 1 - \tau$ and $\mathcal{G}_{1,N}^*(\boldsymbol{x}_\gamma^+(S) \mid S) \geq 1 - \frac{\tau}{2}$, then

$$\left| \mathcal{G}_{\theta_1}^{\text{LLM}} - \mathcal{G}_{1,N}^* \right| \geq \frac{\tau}{2} \Rightarrow D_{\text{TV}} \geq \frac{\tau}{4}.$$

Taking expectation:

$$\mathbb{P}_{S \sim \mathcal{S}} \left[ \mathcal{G}_{\theta_1}^{\text{LLM}}(\boldsymbol{x}_\gamma^+(S) \mid S) \leq 1 - \tau, \ S \in \mathcal{S}_1^+ \right] \leq \frac{4}{\tau} \cdot \mathbb{E}_{S \sim \mathcal{S}} \left[ D_{\text{TV}}(\mathcal{G}_{\theta_1}, \mathcal{G}_{1,N}^*) \right]. \tag{67}$$

**Bounding the Second Term via Markov's Inequality**

Define the expected inverse support quantity:

$$C_\gamma := \mathbb{E}_{S \sim \mathcal{S}} \left[ \frac{1}{\mathcal{G}_{\theta_0}^{\text{LLM}}(\boldsymbol{x}_\gamma^+(S) \mid S)} \right].$$

Then, by Markov's inequality:

$$\mathbb{P}_{S \sim \mathcal{S}}[S \notin \mathcal{S}_1^+] = \mathbb{P} \left[ \mathcal{G}_{\theta_0}^{\text{LLM}}(\boldsymbol{x}_\gamma^+(S) \mid S) < \frac{\log \frac{2}{\tau}}{N} \right]$$
$$= \mathbb{P} \left[ \frac{\log \frac{2}{\tau}}{N \cdot \mathcal{G}_{\theta_0}^{\text{LLM}}(\boldsymbol{x}_\gamma^+(S) \mid S)} > 1 \right]$$
$$\leq \frac{C_\gamma}{N} \log \frac{2}{\tau}. \tag{68}$$

**Final Bound for the First-Step Generation Error**

Combining inequalities (7) and (8), and substituting into the decomposition, we obtain the following upper bound:

$$\mathbb{P}_{S \sim \mathcal{S}} \left[ \mathcal{G}^{\text{LLM}}_{\theta_1}(\boldsymbol{x}^+_\gamma(S) \mid S) \le 1 - \tau \right] \le \frac{4}{\tau} \cdot \mathbb{E}_{S \sim \mathcal{S}} \left[ D_{\text{TV}}(\mathcal{G}^{\text{LLM}}_{\theta_1}(\cdot \mid S), \mathcal{G}^*_{1,N}(\cdot \mid S)) \right] + \frac{C_\gamma}{N} \log \frac{2}{\tau}.$$
(69)

Since $\mathcal{G}^{\text{LLM}}_{\theta_1}(\cdot \mid S)$ is the maximum likelihood estimator of the conditional distribution $\mathcal{G}^*_{1,N}(\cdot \mid S)$ based on sampled data, we now focus on analyzing the expected total variation distance $\mathbb{E}_{S \sim \mathcal{S}} \left[ D_{\text{TV}} \left( \mathcal{G}^{\text{LLM}}_{\theta_1}(\cdot \mid S), \mathcal{G}^*_{1,N}(\cdot \mid S) \right) \right]$.

**Step 1: Error Decomposition under Token Prefix Sequence Modeling**

We begin by establishing a framework for analyzing the error. Consider our dataset consisting of $m$ token trajectories, each of length $\ell$:

$$\mathcal{D} = \left\{ \left( x^{(j)}_1, x^{(j)}_2, \ldots, x^{(j)}_\ell \right) \right\}^m_{j=1},$$

where each token prefix $S^{(j)}_t := (x^{(j)}_1, \ldots, x^{(j)}_t) \in \mathcal{S} \subseteq \mathcal{X}^{\le \ell}$.

To measure how closely our model approximates the reference distribution, we define the log-likelihood discrepancy function:

$$L(\theta, \mathcal{D}) := -\frac{1}{4} \sum^m_{j=1} \sum^\ell_{t=1} \log \frac{\mathcal{G}^*_{1,N}(x^{(j)}_{t+1} \mid S^{(j)}_t)}{\mathcal{G}^{\text{LLM}}_\theta(x^{(j)}_{t+1} \mid S^{(j)}_t)}.$$

For applying the PAC-Bayes framework, we introduce a ghost sample $\tilde{\mathcal{D}} = \left\{ (\tilde{S}^{(j)}_t, \tilde{x}^{(j)}_{t+1}) \right\}^{m,\ell}_{j=1,t=1}$, where $\tilde{S}^{(j)}_t = S^{(j)}_t$ and $\tilde{x}^{(j)}_{t+1} \sim \mathcal{G}^*_{1,N}(\cdot \mid \tilde{S}^{(j)}_t)$. We then define:

$$g(\theta) := L(\theta, \mathcal{D}) - \log \mathbb{E}_{\tilde{\mathcal{D}}} \left[ \exp \left( L(\theta, \tilde{\mathcal{D}}) \right) \,\Big|\, \mathcal{D} \right].$$

**PAC-Bayes Bound under Token Prefix Modeling**

By the standard PAC-Bayes lemma, for any distributions $Q, P \in \Delta(\Theta)$ where $P$ may depend on $\mathcal{D}$, we have:

$$\mathbb{E}_{\theta \sim P}[g(\theta)] \le \text{KL}(P \| Q) + \log \mathbb{E}_{\theta \sim Q} \left[ \exp(g(\theta)) \right].$$

Substituting the definition of $g(\theta)$ and taking expectation over $\mathcal{D}$:

$$\mathbb{E}_{\mathcal{D}} \left[ \exp \left\{ \mathbb{E}_{\theta \sim P} \left[ L(\theta, \mathcal{D}) - \log \mathbb{E}_{\tilde{\mathcal{D}}} \left[ \exp(L(\theta, \tilde{\mathcal{D}})) \mid \mathcal{D} \right] \right] - \text{KL}(P \| Q) \right\} \right] \le 1.$$

Applying Chernoff bounds, with probability at least $1 - \delta$, we obtain:

$$-\mathbb{E}_{\theta \sim P} \left[ \log \mathbb{E}_{\tilde{\mathcal{D}}} \left[ \exp(L(\theta, \tilde{\mathcal{D}})) \mid \mathcal{D} \right] \right] \le -\mathbb{E}_{\theta \sim P}[L(\theta, \mathcal{D})] + \text{KL}(P \| Q) + \log \frac{1}{\delta}.$$

We can lower-bound the left-hand side using a two-step approach involving Jensen's inequality and total variation distance:

$$-\mathbb{E}_{\theta \sim P}\left[\log \mathbb{E}_{\tilde{\mathcal{D}}}\left[\exp(L(\theta, \tilde{\mathcal{D}})) \mid \mathcal{D}\right]\right]$$

$$\geq -\frac{1}{2}\log \mathbb{E}_{\tilde{\mathcal{D}}}\left[\exp\left(-\frac{1}{2}\sum_{j=1}^{m}\sum_{t=1}^{\ell}\log \frac{\mathcal{G}_{1,N}^{*}(x_{t+1}^{(j)} \mid S_t^{(j)})}{\mathcal{G}_{\theta_1}^{\mathrm{LLM}}(x_{t+1}^{(j)} \mid S_t^{(j)})}\right) \mid \mathcal{D}\right]$$

$$-\frac{1}{2}\mathbb{E}_{\theta \sim P}\left[\log \mathbb{E}_{\tilde{\mathcal{D}}}\left[\exp\left(-\frac{1}{2}\sum_{j=1}^{m}\sum_{t=1}^{\ell}\log \frac{\mathcal{G}_{\theta_1}^{\mathrm{LLM}}(x_{t+1}^{(j)} \mid S_t^{(j)})}{\mathcal{G}_{\theta}^{\mathrm{LLM}}(x_{t+1}^{(j)} \mid S_t^{(j)})}\right) \mid \mathcal{D}\right]\right]$$

$$\geq \frac{1}{4}\sum_{j=1}^{m}\sum_{t=1}^{\ell} D_{\mathrm{TV}}^2(\mathcal{G}_{1,N}^{*}(\cdot \mid S_t^{(j)}), \mathcal{G}_{\theta_1}^{\mathrm{LLM}}(\cdot \mid S_t^{(j)}))$$

$$-\frac{1}{2}\mathbb{E}_{\theta \sim P}\left[\log \mathbb{E}_{\tilde{\mathcal{D}}}\left[\exp\left(-\frac{1}{2}\sum_{j=1}^{m}\sum_{t=1}^{\ell}\log \frac{\mathcal{G}_{\theta_1}^{\mathrm{LLM}}(x_{t+1}^{(j)} \mid S_t^{(j)})}{\mathcal{G}_{\theta}^{\mathrm{LLM}}(x_{t+1}^{(j)} \mid S_t^{(j)})}\right) \mid \mathcal{D}\right]\right].$$

Next, for any reference parameter $\theta^* \in \Theta$, we decompose the expected loss:

$$-\mathbb{E}_{\theta \sim P}[L(\theta, \mathcal{D})] = \mathbb{E}_{\theta \sim P}\left[L(\theta^*, \mathcal{D}) + (L(\theta_1, \mathcal{D}) - L(\theta^*, \mathcal{D})) + (L(\theta, \mathcal{D}) - L(\theta_1, \mathcal{D}))\right]$$
$$\leq L(\theta^*, \mathcal{D}) + \mathbb{E}_{\theta \sim P}[L(\theta, \mathcal{D}) - L(\theta_1, \mathcal{D})],$$

where we use the fact that $\theta_1$ minimizes $L(\theta, \mathcal{D})$. In the realizable setting, there exists $\theta^*$ such that $\mathcal{G}_{\theta^*}^{\mathrm{LLM}}(\cdot \mid S) = \mathcal{G}_{1,N}^{*}(\cdot \mid S)$.

Combining these bounds, we obtain:

$$\frac{1}{4}\sum_{j=1}^{m}\sum_{t=1}^{\ell} D_{\mathrm{TV}}^2(\mathcal{G}_{1,N}^{*}(\cdot \mid S_t^{(j)}), \mathcal{G}_{\theta_1}^{\mathrm{LLM}}(\cdot \mid S_t^{(j)})) \tag{70}$$

$$\leq \frac{1}{2}\mathbb{E}_{\theta \sim P}\left[\log \mathbb{E}_{\tilde{\mathcal{D}}}\left[\exp\left(-\frac{1}{2}\sum_{j=1}^{m}\sum_{t=1}^{\ell}\log \frac{\mathcal{G}_{\theta_1}^{\mathrm{LLM}}(x_{t+1}^{(j)} \mid S_t^{(j)})}{\mathcal{G}_{\theta}^{\mathrm{LLM}}(x_{t+1}^{(j)} \mid S_t^{(j)})}\right) \mid \mathcal{D}\right]\right]$$

$$+\frac{1}{4}\sum_{j=1}^{m}\sum_{t=1}^{\ell}\mathbb{E}_{\theta \sim P}\left(\log \frac{\mathcal{G}_{\theta_1}^{\mathrm{LLM}}(x_{t+1}^{(j)} \mid S_t^{(j)})}{\mathcal{G}_{\theta}^{\mathrm{LLM}}(x_{t+1}^{(j)} \mid S_t^{(j)})}\right) + \mathrm{KL}(P\|Q) + \log\frac{1}{\delta}. \tag{71}$$

The first two terms capture the fluctuation under the posterior distribution $\theta \sim P$, which we will address next.

**Step 2: Controlling the Fluctuation via Structured Posterior Sampling**

To control the fluctuation terms, we define the posterior distribution $P \in \Delta(\Theta)$ as a product distribution over each layer of the transformer:

$$P = \prod_{t=1}^{D+1} \mathcal{L}_P(\theta^{(t)}) \tag{72}$$

$$\mathcal{L}_P(\theta^{(D+1)}) = \mathrm{Unif}\left(\mathbb{B}\left(\hat{A}^{(D+1)}, r^{(D+1)}, \|\cdot\|_{1,2}\right)\right)$$

$$\mathcal{L}_P(\theta^{(t)}) = \mathrm{Unif}\left(\mathbb{B}(\hat{\gamma}_1^{(t)}, r_{\gamma,1}^{(t)}, |\cdot|)\right) \cdot \mathrm{Unif}\left(\mathbb{B}(\hat{\gamma}_2^{(t)}, r_{\gamma,2}^{(t)}, |\cdot|)\right) \cdot \mathcal{L}_P(A^{(t)}) \cdot \mathcal{L}_P(W^{(t)})$$

$$\mathcal{L}_P(A^{(t)}) = \mathrm{Unif}\left(\mathbb{B}(\hat{A}_1^{(t)}, r_{A,1}^{(t)}, \|\cdot\|_F)\right) \cdot \mathrm{Unif}\left(\mathbb{B}(\hat{A}_2^{(t)}, r_{A,2}^{(t)}, \|\cdot\|_F)\right)$$

$$\mathcal{L}_P(W^{(t)})$$

$$= \prod_{i=1}^{h} \mathrm{Unif}\left(\mathbb{B}(\hat{W}_i^{Q,(t)}, r_Q^{(t)}, \|\cdot\|_F)\right) \cdot \mathrm{Unif}\left(\mathbb{B}(\hat{W}_i^{K,(t)}, r_K^{(t)}, \|\cdot\|_F)\right) \cdot \mathrm{Unif}\left(\mathbb{B}(\hat{W}_i^{V,(t)}, r_V^{(t)}, \|\cdot\|_F)\right)$$

$$\tag{73}$$

for all $t \in [D]$, where $\mathrm{Unif}(\mathbb{B}(a, r, \|\cdot\|))$ denotes the uniform distribution over a ball centered at $a$ with radius $r$ under norm $\|\cdot\|$.

The radii are carefully calibrated as follows:

$$r_{\gamma,1}^{(t)} = R^{-1}(1 + B_{A,1}B_{A,2})^{-1}\alpha_t^{-1}/(m\ell), \qquad\qquad r_{\gamma,2}^{(t)} = R^{-1}\alpha_t^{-1}/(m\ell),$$

$$r_{A,1}^{(t)} = R^{-1}B_{A,2}^{-1}\alpha_t^{-1}/(m\ell), \qquad\qquad r_{A,2}^{(t)} = R^{-1}B_{A,1}^{-1}\alpha_t^{-1}/(m\ell),$$

$$r_V^{(t)} = R^{-1}h^{-1}(1 + B_{A,1}B_{A,2})^{-1}\alpha_t^{-1}/(m\ell), \qquad r_Q^{(t)} = R^{-1}h^{-1}(1 + B_{A,1}B_{A,2})^{-1}B_V^{-1}B_K^{-1}\alpha_t^{-1}/(m\ell),$$

$$r_K^{(t)} = R^{-1}h^{-1}(1 + B_{A,1}B_{A,2})^{-1}B_V^{-1}B_Q^{-1}\alpha_t^{-1}/(m\ell), \qquad\qquad r^{(D+1)} = \tau B_A^{-1}/(m\ell).$$

This carefully structured posterior allows us to prove the following key result:

**Lemma 6.** *Zhang et al. [2025] With the posterior distribution $P$ defined in* (72), *which is a uniform distribution over a neighborhood around $\theta_1$ with radius proportional to $1/(m\ell)$, we have:*

$$\frac{1}{2}\mathbb{E}_{\theta\sim P}\left[\log\mathbb{E}_{\tilde{\mathcal{D}}}\left[\exp\left(-\frac{1}{2}\sum_{j=1}^{m}\sum_{t=1}^{\ell}\log\frac{\mathcal{G}_{\theta_1}^{LLM}(x_{t+1}^{(j)} \mid S_t^{(j)})}{\mathcal{G}_{\theta}^{LLM}(x_{t+1}^{(j)} \mid S_t^{(j)})}\right) \mid \mathcal{D}\right]\right]$$

$$+ \frac{1}{4}\sum_{j=1}^{m}\sum_{t=1}^{\ell}\mathbb{E}_{\theta\sim P}\left(\log\frac{\mathcal{G}_{\theta_1}^{LLM}(x_{t+1}^{(j)} \mid S_t^{(j)})}{\mathcal{G}_{\theta}^{LLM}(x_{t+1}^{(j)} \mid S_t^{(j)})}\right) = \mathcal{O}(1). \tag{74}$$

*Proof.* A detailed proof can be found in Appendix F.2 of Zhang et al. [2025]. The key insight is that the construction of $P$ with specific radii ensures that the model outputs remain stable under small parameter perturbations, allowing us to bound the fluctuation terms by a constant. $\square$

**Bounding the KL Divergence Term**

To complete our bound, we need to quantify the KL divergence term in (71). We define the reference prior distribution $Q \in \Delta(\Theta)$ as:

$$Q = \prod_{t=1}^{D+1}\mathcal{L}_Q(\theta^{(t)}),$$

with the following component-wise structure:

- **Final linear layer:**

$$\mathcal{L}_Q(\theta^{(D+1)}) = \mathrm{Unif}\left(\mathbb{B}(0, B_A, \|\cdot\|_{1,2})\right).$$

- **Transformer layers $t \in [D]$:**

$$\mathcal{L}_Q(\theta^{(t)}) = \mathrm{Unif}\left(\mathbb{B}(1/2, 1/2, |\cdot|)\right) \cdot \mathrm{Unif}\left(\mathbb{B}(1/2, 1/2, |\cdot|)\right) \cdot \mathcal{L}_Q(A^{(t)}) \cdot \mathcal{L}_Q(W^{(t)}),$$

$$\mathcal{L}_Q(A^{(t)}) = \mathrm{Unif}\left(\mathbb{B}(0, B_{A,1}, \|\cdot\|_F)\right) \cdot \mathrm{Unif}\left(\mathbb{B}(0, B_{A,2}, \|\cdot\|_F)\right),$$

$$\mathcal{L}_Q(W^{(t)}) = \prod_{i=1}^{h}\mathrm{Unif}\left(\mathbb{B}(0, B_Q, \|\cdot\|_F)\right) \cdot \mathrm{Unif}\left(\mathbb{B}(0, B_K, \|\cdot\|_F)\right) \cdot \mathrm{Unif}\left(\mathbb{B}(0, B_V, \|\cdot\|_F)\right).$$

The KL divergence between this prior $Q$ and our posterior $P$ can be bounded as:

$$\mathrm{KL}(P\|Q) = \mathcal{O}\left((D^2 \cdot d \cdot (d_F + d_h + d) + d \cdot d_y) \cdot \log\left(1 + \frac{m\ell}{\beta}RhB_AB_{A,1}B_{A,2}B_QB_KB_V\right)\right)$$

$$= \mathcal{O}\left(\tilde{D} \cdot \log\left(1 + m\ell\tilde{B}\right)\right) \tag{75}$$

This term represents the complexity of the posterior family relative to the prior and controls the flexibility allowed in posterior fluctuations under the PAC-Bayes objective.

**Deriving the Final Bound**

By combining (74) and (75) and substituting into (71), we obtain:

$$\frac{1}{4} \sum_{j=1}^{m} \sum_{t=1}^{\ell} D_{\text{TV}}^2(\mathcal{G}_{1,N}^*(\cdot \mid S_t^{(j)}), \mathcal{G}_{\theta_1}^{\text{LLM}}(\cdot \mid S_t^{(j)})) \lesssim \mathcal{O}(1) + \mathcal{O}\left(\tilde{D} \cdot \log\left(1 + m\ell\tilde{B}\right)\right) + \log\frac{1}{\delta}$$

Taking the square root and using Jensen's inequality, we get:

$$\frac{1}{m\ell} \sum_{j=1}^{m} \sum_{t=1}^{\ell} D_{\text{TV}}(\mathcal{G}_{1,N}^*(\cdot \mid S_t^{(j)}), \mathcal{G}_{\theta_1}^{\text{LLM}}(\cdot \mid S_t^{(j)}))$$

$$\leq \sqrt{\frac{1}{(m\ell)^2} \sum_{j=1}^{m} \sum_{t=1}^{\ell} D_{\text{TV}}^2(\mathcal{G}_{1,N}^*(\cdot \mid S_t^{(j)}), \mathcal{G}_{\theta_1}^{\text{LLM}}(\cdot \mid S_t^{(j)}))}$$

$$\lesssim \sqrt{\frac{\tilde{D} \cdot \log\left(1 + m\ell\tilde{B}\right) + \log\frac{1}{\delta}}{m\ell}}$$

Based on Lemma F.4 in Hu et al. [2024], we can further bound the difference between the empirical and expected total variation distances:

$$\frac{1}{m\ell} \sum_{j=1}^{m} \sum_{t=1}^{\ell} \mathbb{E}_{S_t^{(j)}}\left[D_{\text{TV}}(\mathcal{G}_{1,N}^*(\cdot \mid S_t^{(j)}), \mathcal{G}_{\theta_1}^{\text{LLM}}(\cdot \mid S_t^{(j)}))\right] - \frac{1}{m\ell} \sum_{j=1}^{m} \sum_{t=1}^{\ell} D_{\text{TV}}(\mathcal{G}_{1,N}^*(\cdot \mid S_t^{(j)}), \mathcal{G}_{\theta_1}^{\text{LLM}}(\cdot \mid S_t^{(j)}))$$

$$\lesssim \sqrt{\frac{1}{m\ell}\left(\tilde{D} \cdot \log\left(1 + m\ell\tilde{B}\right) + \log\frac{1}{\delta}\right)} \tag{76}$$

Therefore, with probability at least $1 - \delta$, we can bound the expected total variation distance:

$$\mathbb{E}_{S \sim \mathcal{S}}\left[D_{\text{TV}}(\mathcal{G}_{\theta_1}^{\text{LLM}}(\cdot \mid S), \mathcal{G}_{1,N}^*(\cdot \mid S))\right] \lesssim \sqrt{\frac{\tilde{D} \cdot \log\left(1 + m\ell\tilde{B}\right) + \log\frac{1}{\delta}}{m\ell}} \tag{77}$$

**Final Bound for $\mathcal{G}_{\theta_1}^{\text{LLM}}$**

Returning to our original objective in (69), we can now derive the final bound:

$$\mathbb{P}_{S \sim \mathcal{S}}\left[\mathcal{G}_{\theta_1}^{\text{LLM}}(\boldsymbol{x}_\gamma^+(S) \mid S) \leq 1 - \tau\right] \leq \frac{4}{\tau} \cdot \mathbb{E}_{S \sim \mathcal{S}}\left[D_{\text{TV}}(\mathcal{G}_{\theta_1}^{\text{LLM}}(\cdot \mid S), \mathcal{G}_{1,N}^*(\cdot \mid S))\right] + \frac{C_\gamma}{N} \log\frac{2}{\tau}$$

$$\lesssim \frac{4}{\tau} \cdot \sqrt{\frac{\tilde{D} \cdot \log\left(1 + m\ell\tilde{B}\right) + \log\frac{1}{\delta}}{m\ell}} + \frac{C_\gamma}{N} \log\frac{2}{\tau}$$

$$\lesssim \frac{4}{\tau} \cdot \sqrt{\frac{\tilde{D} \cdot \log\left(m\ell\tilde{B}\right)}{m\ell} \log\frac{1}{\delta}} + \frac{C_\gamma}{N} \log\frac{2}{\tau} \tag{78}$$

Let $n = m\ell$ denote the total number of samples. Thus, with probability at least $1 - \delta$, we have:

$$\mathbb{P}_{S \sim \mathcal{S}}\left[\mathcal{G}_{\theta_1}^{\text{LLM}}(\boldsymbol{x}_\gamma^+(S) \mid S) \leq 1 - \tau\right] \lesssim \frac{1}{\tau} \cdot \sqrt{\frac{\tilde{D} \log\left(n\tilde{B}\right)}{n} \log\frac{1}{\delta}} + \frac{C_\gamma}{N} \log\frac{1}{\tau} \tag{79}$$

This final bound characterizes the probability that our trained LLM fails to assign sufficient probability to the desired completion. The bound decays with the square root of the sample size $n$, but depends on model complexity $\tilde{D}$ and scales inversely with the threshold parameter $\tau$.

**Recursive Bound for the Second Generation Model $\mathcal{G}_{\theta_2}^{\text{LLM}}$**

**Definition of Verified Prefix Set for $\mathcal{G}_{\theta_2}^{\text{LLM}}$**

We now extend our analysis to the second generation model by defining a subset of prefix-conditioned inputs that have sufficient verified coverage from the first model:

$$
\mathcal{D}_2^+ := \left\{ S \in \mathcal{S} \,\middle|\, \mathcal{G}_{\theta_1}^{\text{LLM}} \left( \boldsymbol{y}_\gamma^+(S) \mid S \right) \geq \frac{\log \frac{2}{\tau}}{N} \right\},
$$

where the verified high-reward candidate set $\boldsymbol{y}_\gamma^+(S)$ contains tokens with near-optimal self-reward:

$$
\boldsymbol{y}_\gamma^+(S) = \left\{ y \in \mathcal{X} \,\middle|\, s_{\text{verify}}(y \mid S) \geq \max_{y' \in \mathcal{X}} s_{\text{verify}}(y' \mid S) - \gamma \right\}.
$$

**Bounding the Low-Confidence Region for $\mathcal{G}_{\theta_2}^{\text{LLM}}$**

Applying the same total variation-based analysis as for the first-step model, we can bound the probability that $\mathcal{G}_{\theta_2}^{\text{LLM}}$ assigns low confidence to verified candidate regions:

$$
\mathbb{P}_{S \sim \mathcal{S}} \left[ \mathcal{G}_{\theta_2}^{\text{LLM}}(\boldsymbol{y}_\gamma^+(S) \mid S) \leq 1 - \tau, \; S \in \mathcal{D}_2^+ \right] \leq \frac{4}{\tau} \cdot D_{\text{TV}} \left( \mathcal{G}_{\theta_2}^{\text{LLM}}(\cdot \mid S), \mathcal{G}_{2,N}^*(\cdot \mid S) \right).
$$

**Bounding the Failure Rate of Verified Prefixes**

To complete our analysis, we need to bound the probability that a prefix falls outside the verified region $\mathcal{D}_2^+$. Using Markov's inequality, we have:

$$
\mathbb{P}_{S \sim \mathcal{S}}[S \notin \mathcal{D}_2^+] = \mathbb{P}_{S \sim \mathcal{S}} \left[ \mathcal{G}_{\theta_1}^{\text{LLM}} \left( \boldsymbol{y}_\gamma^+(S) \mid S \right) < \frac{\log \frac{2}{\tau}}{N} \right] \tag{80}
$$

$$
= \mathbb{P}_{S \sim \mathcal{S}} \left[ \frac{\log \frac{2}{\tau}}{N \cdot \mathcal{G}_{\theta_1}^{\text{LLM}} \left( \boldsymbol{y}_\gamma^+(S) \mid S \right)} > 1 \right] \tag{81}
$$

$$
\leq \frac{\log \frac{2}{\tau}}{N} \cdot \mathbb{E}_{S \sim \mathcal{S}} \left[ \frac{1}{\mathcal{G}_{\theta_1}^{\text{LLM}} \left( \boldsymbol{y}_\gamma^+(S) \mid S \right)} \right]. \tag{82}
$$

This leads us to define the reciprocal of the model confidence as a random variable:

$$
Z(S) := \frac{1}{\mathcal{G}_{\theta_1}^{\text{LLM}} \left( \boldsymbol{y}_\gamma^+(S) \mid S \right)}.
$$

Using the tail-integral identity for expectations:

$$
\mathbb{E}_{S \sim \mathcal{S}}[Z(S)] = \int_0^\infty \mathbb{P}(Z(S) \geq t) \, \mathrm{d}t,
$$

and combining with our PAC-Bayes bounds on the total variation distance between $\mathcal{G}_{\theta_1}^{\text{LLM}}$ and $\mathcal{G}_{1,N}^*$, we can derive:

$$
C_{\gamma,1} := \mathbb{E}_{S \sim \mathcal{S}} \left[ \frac{1}{\mathcal{G}_{\theta_1}^{\text{LLM}} \left( \boldsymbol{y}_\gamma^+(S) \mid S \right)} \right] \lesssim 1 + A \frac{1}{C_0} \log \left( \frac{1}{A} \right) + C \frac{1}{C_0} \log \left( \frac{1}{C_0} \right),
$$

where:

- $A = \sqrt{\frac{1}{n} \log(n\tilde{B}) \tilde{D} \log \frac{1}{\delta}}$ represents the statistical error from PAC-Bayes bounds

- $C = \frac{C_\gamma}{N}$ is the approximation error from the confidence threshold

- $C_0$ represents the minimum prediction confidence of $\mathcal{G}_{\theta_1}^{\text{LLM}}$ across verified high-reward candidates

This bound quantifies how low-confidence predictions propagate into the next generation model.

**Solving the Recurrence Relation**

By extending our analysis recursively, similar to the proof of Theorem 1, we can derive a recurrence relation for the expected reciprocal of model confidence after $t$ generations. The solution to this recurrence is:

$$
\begin{aligned}
C_{\gamma,t} &\lesssim 1 + A_t \frac{1}{C_0} \log\left(\frac{1}{A_t}\right) + C_t \frac{1}{C_0} \log\left(\frac{1}{C_0}\right) \\
&\lesssim 1 + \frac{1}{C_0} D_{\text{TV}} \log\left(\frac{1}{D_{\text{TV}}}\right) \\
&\quad + \left( \left(\frac{1}{NC_0} \log\left(\frac{1}{C_0}\right)\right)^t \frac{C_\gamma}{N} + \frac{1 + \frac{D_{\text{TV}}}{C_0} \log\left(\frac{1}{D_{\text{TV}}}\right)}{N - \frac{1}{C_0} \log\left(\frac{1}{C_0}\right)} \right) \cdot \frac{1}{C_0} \log\left(\frac{1}{C_0}\right) \\
&\lesssim 1 + \frac{1}{C_0} \sqrt{\frac{1}{n} \log(n\tilde{B}) \tilde{D} \log\frac{1}{\delta}} \log\left(\sqrt{n}\right) \\
&\quad + \left( \left(\frac{1}{NC_0} \log\left(\frac{1}{C_0}\right)\right)^t \frac{C_\gamma}{N} + \frac{1}{N} \right) \cdot \frac{1}{C_0} \log\left(\frac{1}{C_0}\right)
\end{aligned}
\tag{83}
$$

This expression shows how the error compounds over iterations, with three main components: 1. A statistical error term that decreases with sample size $n$ 2. An approximation error term that decreases with the number of samples $N$ used in self-training 3. A recursively propagating error term that decays exponentially with the number of generations $t$

**Final Low-Confidence Bound After $T$ Recursive Generations**

After $T$ recursive updates, with probability at least $1 - \delta$, the probability that the final model assigns low confidence to its verified top candidates is bounded by:

$$
\begin{aligned}
\mathbb{P}_{S \sim \mathcal{S}} \left[ \mathcal{G}_{\theta_T}^{\text{LLM}}(x_\gamma^+(S) \mid S) \leq 1 - \tau \right] &\lesssim \left( \frac{1}{\tau} + \frac{1}{NC_0} \log\left(\frac{1}{\tau}\right) \right) \sqrt{\frac{1}{n} \log(n\tilde{B}) \tilde{D} \log\frac{1}{\delta}} \\
&\quad + \left( \frac{1}{N} + \frac{1}{N^2 C_0} \right) \log\left(\frac{1}{\tau}\right) + \left( \frac{1}{NC_0} \log\left(\frac{1}{C_0}\right) \right)^T \frac{C_\gamma}{N} \log\left(\frac{1}{\tau}\right).
\end{aligned}
\tag{84}
$$

This bound reveals that as $T$ increases, the third term vanishes exponentially, demonstrating that self-training eventually eliminates the initial confidence errors, provided that $\frac{1}{NC_0} < 1$, which is typically satisfied in practice.

**Final Accuracy and Error Decomposition**

To translate these confidence bounds into concrete accuracy guarantees, we define:

$$
x_T^+(S) := \arg\max_{x \in \mathcal{X}} \mathcal{G}_{\theta_T}^{\text{LLM}}(x \mid S), \quad x^*(S) := \text{ground-truth token following } S.
$$

The accuracy of generation after $T$ steps is:

$$
\text{Acc}(\mathcal{G}_{\theta_T}^{\text{LLM}}) := \mathbb{P}_{S \sim \mathcal{S}} \left[ x_T^+(S) = x^*(S) \right],
$$

By decomposing this probability and using our previous bounds, we obtain:

$$
\begin{aligned}
\text{Acc}(\theta_T) &= \mathbb{P}_{S \sim \mathcal{S}} \left[ x_T^+(S) = x^*(S) \,\middle|\, \mathcal{G}_{\theta_T}^{\text{LLM}}\left(x_\gamma^+(S) \mid S\right) \geq 1 - \tau \right] \cdot \mathbb{P}_{S \sim \mathcal{S}} \left[ \mathcal{G}_{\theta_T}^{\text{LLM}}(x_\gamma^+(S) \mid S) \geq 1 - \tau \right] \\
&\gtrsim 1 - \epsilon(\gamma) - (1 - \epsilon(\gamma)) \left( \frac{1}{\tau} + \frac{1}{NC_0} \log\left(\frac{1}{\tau}\right) \right) \sqrt{\frac{1}{n} \log(n\tilde{B}) \tilde{D} \log\frac{1}{\delta}} \\
&\quad - (1 - \epsilon(\gamma)) \left( \frac{1}{N} + \frac{1}{N^2 C_0} \right) \log\left(\frac{1}{\tau}\right) + \left( \frac{1}{NC_0} \log\left(\frac{1}{C_0}\right) \right)^T \frac{C_\gamma}{N} \log\left(\frac{1}{\tau}\right)
\end{aligned}
\tag{85}
$$

The corresponding test error is therefore:

$$\text{Err}(\mathcal{G}_{\theta_T}^{\text{LLM}}) := 1 - \text{Acc}(\mathcal{G}_{\theta_T}^{\text{LLM}}) = \mathbb{P}_{S \sim \mathcal{S}}\left[x_T^+(S) \neq x^*(S)\right],$$

Which satisfies:

$$\text{Err}(\mathcal{G}_{\theta_T}^{\text{LLM}}) \lesssim \epsilon(\gamma) + \left(\frac{1}{\tau} + \frac{1}{NC_0}\log\left(\frac{1}{\tau}\right)\right)\sqrt{\frac{1}{n}\log(n\tilde{B})\tilde{D}\log\frac{1}{\delta}}$$
$$+ \left(\frac{1}{N} + \frac{1}{N^2 C_0}\right)\log\left(\frac{1}{\tau}\right) + \left(\frac{1}{NC_0}\log\left(\frac{1}{C_0}\right)\right)^T\frac{C_\gamma}{N}\log\left(\frac{1}{\tau}\right). \quad (86)$$

This error bound consists of four components: 1. An inherent approximation error $\epsilon(\gamma)$ from the $\gamma$-approximation of optimal reward 2. A statistical error from finite training data that scales as $O(1/\sqrt{n})$ 3. An approximation error from finite self-training samples that scales as $O(1/N)$ 4. A rapidly diminishing initial error term that decays as $O\left((1/NC_0)^{T+1}\right)$

This analysis demonstrates how recursive self-training progressively improves model performance, with the error bound decreasing as both the pre-training dataset size $n$ and the self-training dataset size $N$ increase. Moreover, the benefit of additional recursive generations $T$ is most pronounced in early iterations, with diminishing returns as $T$ increases.

$\square$

# I   Exponential Error Growth in Naive Recursive Training of Transformer-Based LLMs

In this section, we analyze the failure mode of naive recursive training applied to Transformer-based large language models (LLMs), where models are trained exclusively on synthetic data generated by their predecessors without any verification or external supervision. This setting, though increasingly common due to the scarcity of real data, introduces significant risk: even small inaccuracies in early generations can compound over time, causing the model to drift progressively further from the original data distribution.

We formalize this phenomenon by characterizing how prediction errors propagate across generations. Specifically, we show that the absence of quality control leads to exponential error growth, ultimately resulting in model collapse. This result highlights the inherent instability of unverified recursive training and underscores the necessity of intervention strategies such as real data anchoring or self-verification. We now present the formal theorem and provide its proof.

**Theorem 7** (Error Propagation for Transformer-based LLMs in Naive Recursive Training). *Let* $\mathcal{G}_{\theta_t}^{LLM}(x \mid S)$ *denote the conditional generation distribution of an autoregressive Transformer-based LLM at recursive step $t$, where $S \in \mathcal{S} \subseteq \mathcal{X}^{\leq \ell}$ is a prefix sequence. Suppose the model is trained via naive recursive generation without any self-verification mechanism. Then, with probability at least $1 - \delta$, the prediction error of $\mathcal{G}_{\theta_T}^{LLM}$ is bounded as:*

$$\text{Err}(\mathcal{G}_{\theta_T}^{LLM}) \lesssim \epsilon(\gamma) + \frac{1}{\tau}\sqrt{\frac{1}{n}\log(n\tilde{B})\tilde{D}\log\frac{1}{\delta}} + \left(C_\gamma + \sqrt{\frac{1}{n}\log(n\tilde{B})\tilde{D}\log\frac{1}{\delta}}\right)\left(\frac{1}{C_0} + \log\frac{1}{C_0}\right)^T,$$

*where* $C_\gamma := \mathbb{E}_{S \sim \mathcal{S}}\left[\frac{1}{\mathcal{G}_{\theta_0}^{LLM}(\boldsymbol{x}_\gamma^+(S)|S)}\right]$ *denotes the expected inverse confidence assigned to top-reward candidates under the initial model.*

Note that since $C_\gamma > 1$, the error bound includes an exponentially growing term of the form

$$\left(C_\gamma + \sqrt{\frac{1}{n}\log(n\tilde{B})\tilde{D}\log\frac{1}{\delta}}\right)\left(\frac{1}{C_0} + \log\frac{1}{C_0}\right)^T.$$

As the number of recursive steps $T \to \infty$, this term can dominate the total error, reflecting the compounding effect of unfiltered distributional shift. Without confidence-based filtering, the model may gradually amplify its own errors, eventually leading to model collapse.

*Proof of Theorem 7.* We establish the following result by analyzing how prediction errors propagate across recursive training iterations in the absence of verification mechanisms. While our proof strategy parallels the one used in Theorem 5, it introduces key modifications to account for the lack of confidence-based filtering or quality control during training.

**Foundational Definitions and Error Decomposition**

We begin by defining the subset of prefix-token inputs for which the base model $\mathcal{G}_{\theta_0}^{\mathrm{LLM}}$ assigns sufficient probability mass to its top predictions:

$$\mathcal{S}_1^+ := \left\{ S \in \mathcal{S} \,\middle|\, \mathcal{G}_{\theta_0}^{\mathrm{LLM}}(\boldsymbol{x}_\gamma^+(S) \mid S) \geq 1 - \frac{\tau}{2} \right\},$$

where the high-reward candidate set is defined via an internal self-reward mechanism:

$$\boldsymbol{x}_\gamma^+(S) := \left\{ x \in \mathcal{X} \,\middle|\, r_{\mathrm{self}}(x \mid S) \geq \max_{x' \in \mathcal{X}} r_{\mathrm{self}}(x' \mid S) - \gamma \right\},$$

and the self-reward function is given by:

$$r_{\mathrm{self}}(x \mid S) := \log \mathcal{G}_{\theta_0}^{\mathrm{LLM}}(x \mid S).$$

We aim to analyze the likelihood that the model $\mathcal{G}_{\theta_1}^{\mathrm{LLM}}$, trained recursively without any verification mechanism, assigns insufficient probability mass to its own top candidates. We decompose this event as:

$$\mathbb{P}_{S \sim \mathcal{S}} \left[ \mathcal{G}_{\theta_1}^{\mathrm{LLM}}(\boldsymbol{x}_\gamma^+(S) \mid S) \leq 1 - \tau \right] \leq \mathbb{P}_{S \sim \mathcal{S}} \left[ \mathcal{G}_{\theta_1}^{\mathrm{LLM}}(\boldsymbol{x}_\gamma^+(S) \mid S) \leq 1 - \tau, \ S \in \mathcal{S}_1^+ \right]$$
$$+ \mathbb{P}_{S \sim \mathcal{S}} \left[ S \notin \mathcal{S}_1^+ \right]. \tag{87}$$

**Analysis of Model Divergence via Total Variation Distance**

To quantify the discrepancy between consecutive model generations in the absence of verification, we analyze their divergence using the total variation (TV) distance. For any input prefix $S \in \mathcal{S}$ and output token $x \in \mathcal{X}$, we have:

$$\left| \mathcal{G}_{\theta_1}^{\mathrm{LLM}}(x \mid S) - \mathcal{G}_{\theta_0}^{\mathrm{LLM}}(x \mid S) \right| \leq 2 D_{\mathrm{TV}} \left( \mathcal{G}_{\theta_1}^{\mathrm{LLM}}(\cdot \mid S), \mathcal{G}_{\theta_0}^{\mathrm{LLM}}(\cdot \mid S) \right).$$

This inequality can be inverted to obtain a lower bound on the TV distance between generations:

$$D_{\mathrm{TV}} \left( \mathcal{G}_{\theta_1}^{\mathrm{LLM}}(\cdot \mid S), \mathcal{G}_{\theta_0}^{\mathrm{LLM}}(\cdot \mid S) \right) \geq \frac{1}{2} \left| \mathcal{G}_{\theta_1}^{\mathrm{LLM}}(\boldsymbol{x}_\gamma^+(S) \mid S) - \mathcal{G}_{\theta_0}^{\mathrm{LLM}}(\boldsymbol{x}_\gamma^+(S) \mid S) \right|.$$

For prefix contexts in the high-confidence region $\mathcal{S}_1^+$, the base model assigns:

$$\mathcal{G}_{\theta_0}^{\mathrm{LLM}}(\boldsymbol{x}_\gamma^+(S) \mid S) \geq 1 - \frac{\tau}{2}.$$

If the updated model assigns insufficient mass to this set, i.e.,

$$\mathcal{G}_{\theta_1}^{\mathrm{LLM}}(\boldsymbol{x}_\gamma^+(S) \mid S) \leq 1 - \tau,$$

then the discrepancy is bounded below by:

$$\left| \mathcal{G}_{\theta_1}^{\mathrm{LLM}}(\boldsymbol{x}_\gamma^+(S) \mid S) - \mathcal{G}_{\theta_0}^{\mathrm{LLM}}(\boldsymbol{x}_\gamma^+(S) \mid S) \right| \geq \frac{\tau}{2}.$$

Consequently, the probability that the updated model fails to retain high confidence on verified inputs can be bounded by the expected TV divergence:

$$\mathbb{P}_{S \sim \mathcal{S}} \left[ \mathcal{G}_{\theta_1}^{\mathrm{LLM}}(\boldsymbol{x}_\gamma^+(S) \mid S) \leq 1 - \tau, \ S \in \mathcal{S}_1^+ \right] \leq \frac{4}{\tau} \cdot \mathbb{E}_{S \sim \mathcal{S}} \left[ D_{\mathrm{TV}} \left( \mathcal{G}_{\theta_1}^{\mathrm{LLM}}(\cdot \mid S), \mathcal{G}_{\theta_0}^{\mathrm{LLM}}(\cdot \mid S) \right) \right]. \tag{88}$$

**Characterizing the Complement Set via Inverse Confidence**

Let $C_\gamma := \mathbb{E}_{S \sim \mathcal{S}} \left[ \frac{1}{\mathcal{G}_{\theta_0}^{\text{LLM}}(\boldsymbol{x}_\gamma^+(S)|S)} \right]$ denote the expected inverse likelihood mass assigned to high-reward candidate sets under the base model. By applying Markov's inequality, we obtain:

$$
\begin{aligned}
\mathbb{P}_{S \sim \mathcal{S}}[S \notin \mathcal{S}_1^+] &= \mathbb{P}_{S \sim \mathcal{S}} \left[ \mathcal{G}_{\theta_0}^{\text{LLM}} \left( \boldsymbol{x}_\gamma^+(S) \mid S \right) < 1 - \frac{\tau}{2} \right] \\
&= \mathbb{P}_{S \sim \mathcal{S}} \left[ \frac{1 - \frac{\tau}{2}}{\mathcal{G}_{\theta_0}^{\text{LLM}}(\boldsymbol{x}_\gamma^+(S) \mid S)} > 1 \right] \\
&\leq \left( 1 - \frac{\tau}{2} \right) \cdot \mathbb{E}_{S \sim \mathcal{S}} \left[ \frac{1}{\mathcal{G}_{\theta_0}^{\text{LLM}}(\boldsymbol{x}_\gamma^+(S) \mid S)} \right] \\
&= \left( 1 - \frac{\tau}{2} \right) C_\gamma.
\end{aligned}
\tag{89}
$$

Combining equations (35) and (36) into the decomposition (34), we arrive at the following bound for the first recursive generation step:

$$
\mathbb{P}_{S \sim \mathcal{S}} \left[ \mathcal{G}_{\theta_1}^{\text{LLM}}(\boldsymbol{x}_\gamma^+(S) \mid S) \leq 1 - \tau \right] \leq \frac{4}{\tau} \cdot \mathbb{E}_{S \sim \mathcal{S}} \left[ D_{\text{TV}} \left( \mathcal{G}_{\theta_1}^{\text{LLM}}(\cdot \mid S), \mathcal{G}_{\theta_0}^{\text{LLM}}(\cdot \mid S) \right) \right] + \left( 1 - \frac{\tau}{2} \right) C_\gamma.
\tag{90}
$$

Furthermore, from the proof of theorem 3, we know that:

$$
\mathbb{E}_{S \sim \mathcal{S}} \left[ D_{\text{TV}} \left( \mathcal{G}_{\theta_1}^{\text{LLM}}(\cdot \mid S), \mathcal{G}_{\theta_0}^{\text{LLM}}(\cdot \mid S) \right) \right] \lesssim \sqrt{\frac{1}{n} \log \left( n\tilde{B} \right)} \tilde{D} \log \frac{1}{\delta}
\tag{91}
$$

Thus, we have

$$
\mathbb{P}_{S \sim \mathcal{S}} \left[ \mathcal{G}_{\theta_1}^{\text{LLM}}(\boldsymbol{x}_\gamma^+(S) \mid S) \leq 1 - \tau \right] \lesssim \frac{1}{\tau} \cdot \sqrt{\frac{1}{n} \log \left( n\tilde{B} \right)} \tilde{D} \log \frac{1}{\delta} + \left( 1 - \frac{\tau}{2} \right) C_\gamma.
\tag{92}
$$

**Recursive Analysis for the Second Generation Model $\mathcal{G}_{\theta_2}^{\text{LLM}}$**

Following an analogous approach, we define the high-confidence support region for the second generation model:

$$
\mathcal{S}_2^+ := \left\{ S \in \mathcal{S} \,\middle|\, \mathcal{G}_{\theta_1}^{\text{LLM}} \left( \boldsymbol{x}_\gamma^+(S) \mid S \right) \geq 1 - \frac{\tau}{2} \right\}.
$$

For inputs $S \in \mathcal{S}_2^+$, a discrepancy in predicted probability mass between $\mathcal{G}_{\theta_2}^{\text{LLM}}$ and $\mathcal{G}_{\theta_1}^{\text{LLM}}$ on high-reward candidates leads to the bound:

$$
\mathbb{P}_{S \sim \mathcal{S}} \left[ \mathcal{G}_{\theta_2}^{\text{LLM}}(\boldsymbol{x}_\gamma^+(S) \mid S) \leq 1 - \tau, \, S \in \mathcal{S}_2^+ \right] \leq \frac{4}{\tau} \cdot \mathbb{E}_{S \sim \mathcal{S}} \left[ D_{\text{TV}} \left( \mathcal{G}_{\theta_2}^{\text{LLM}}(\cdot \mid S), \mathcal{G}_{\theta_1}^{\text{LLM}}(\cdot \mid S) \right) \right]. \tag{93}
$$

For the complement region $\mathcal{S} \setminus \mathcal{S}_2^+$, we apply Markov's inequality as before:

$$
\begin{aligned}
\mathbb{P}_{S \sim \mathcal{S}} \left[ S \notin \mathcal{S}_2^+ \right] &= \mathbb{P}_{S \sim \mathcal{S}} \left[ \mathcal{G}_{\theta_1}^{\text{LLM}}(\boldsymbol{x}_\gamma^+(S) \mid S) < 1 - \frac{\tau}{2} \right] \\
&\leq \left( 1 - \frac{\tau}{2} \right) \cdot \mathbb{E}_{S \sim \mathcal{S}} \left[ \frac{1}{\mathcal{G}_{\theta_1}^{\text{LLM}}(\boldsymbol{x}_\gamma^+(S) \mid S)} \right].
\end{aligned}
\tag{94}
$$

**Generalization to Arbitrary Generations**

We now generalize the inverse-confidence recurrence to arbitrary recursive steps. Let $C_{\gamma,t} := \mathbb{E}_{S \sim \mathcal{S}} \left[ \frac{1}{\mathcal{G}_{\theta_t}^{\text{LLM}}(\boldsymbol{x}_\gamma^+(S)|S)} \right]$ denote the expected inverse confidence at generation $t$. Then the recurrence

relation is given by:

$$C_{\gamma,t} \leq \frac{C_{\gamma,t-1} + A}{C_0} + A \log \frac{1}{A} + C_{\gamma,t-1} \log \frac{1}{C_0}$$

$$\leq \left( \frac{1}{C_0} + \log \frac{1}{C_0} \right) C_{\gamma,t-1} + A \log \frac{1}{A}, \tag{95}$$

where $A := \sqrt{\frac{1}{n} \log(n\tilde{B})} \tilde{D} \log \frac{1}{\delta}$ reflects the TV-induced generalization noise, and $C_0$ is the minimum confidence over all prefix-conditioned top predictions in earlier generations.

Solving this recurrence yields the following bound:

$$C_{\gamma,t} \lesssim \left( C_\gamma + A \log \frac{1}{A} \right) \left( \frac{1}{C_0} + \log \frac{1}{C_0} \right)^t. \tag{96}$$

**Asymptotic Error Analysis After $T$ Generations**

Finally, applying this bound to the error probability after $T$ recursive generations, we obtain:

$$\mathbb{P}_{S \sim \mathcal{S}} \left[ \mathcal{G}_{\theta_T}^{\text{LLM}} \left( \boldsymbol{x}_\gamma^+(S) \mid S \right) \leq 1 - \tau \right]$$

$$\lesssim \frac{1}{\tau} \cdot A + (1 - \frac{\tau}{2}) \left( C_\gamma + A \log \frac{1}{A} \right) \left( \frac{1}{C_0} + \log \frac{1}{C_0} \right)^T.$$

$$\lesssim \frac{1}{\tau} \cdot \sqrt{\frac{1}{n} \log(n\tilde{B})} \tilde{D} \log \frac{1}{\delta} + \left( C_\gamma + \sqrt{\frac{1}{n} \log(n\tilde{B})} \tilde{D} \log \frac{1}{\delta} \right) \left( \frac{1}{C_0} + \log \frac{1}{C_0} \right)^T \tag{97}$$

This bound highlights exponential sensitivity to both the minimum confidence $C_0$ and the number of recursive steps $T$, in the absence of verification mechanisms.

**Final Accuracy and Error Decomposition**

We define:

$$x_T^+(S) := \arg\max_{x \in \mathcal{X}} \mathcal{G}_{\theta_T}^{\text{LLM}}(x \mid S), \quad x^*(S) := \text{ground-truth token following } S.$$

Then the accuracy of generation after $T$ steps is:

$$\text{Acc}(\mathcal{G}_{\theta_T}^{\text{LLM}}) := \mathbb{P}_{S \sim \mathcal{S}} \left[ x_T^+(S) = x^*(S) \right],$$

and satisfies:

$$\text{Acc}(\theta_T) = \mathbb{P}_{S \sim \mathcal{S}} \left[ x_T^+(S) = x^*(S) \mid \mathcal{G}_{\theta_T}^{\text{LLM}} \left( \boldsymbol{x}_\gamma^+(S) \mid S \right) \geq 1 - \tau \right] \cdot \mathbb{P}_{S \sim \mathcal{S}} \left[ \mathcal{G}_{\theta_T}^{\text{LLM}}(\boldsymbol{x}_\gamma^+(S) \mid S) \geq 1 - \tau \right]$$

$$\gtrsim 1 - \epsilon(\gamma) - (1 - \epsilon(\gamma)) \frac{1}{\tau} \cdot \sqrt{\frac{1}{n} \log(n\tilde{B})} \tilde{D} \log \frac{1}{\delta}$$

$$- (1 - \epsilon(\gamma)) \left( C_\gamma + \sqrt{\frac{1}{n} \log(n\tilde{B})} \tilde{D} \log \frac{1}{\delta} \right) \left( \frac{1}{C_0} + \log \frac{1}{C_0} \right)^T \tag{98}$$

The corresponding test error is:

$$\text{Err}(\mathcal{G}_{\theta_T}^{\text{LLM}}) := 1 - \text{Acc}(\mathcal{G}_{\theta_T}^{\text{LLM}}) = \mathbb{P}_{S \sim \mathcal{S}} \left[ x_T^+(S) \neq x^*(S) \right],$$

and satisfies:

$$\text{Err}(\mathcal{G}_{\theta_T}^{\text{LLM}}) \lesssim \epsilon(\gamma) + \frac{1}{\tau} \cdot \sqrt{\frac{1}{n} \log(n\tilde{B})} \tilde{D} \log \frac{1}{\delta} + \left( C_\gamma + \sqrt{\frac{1}{n} \log(n\tilde{B})} \tilde{D} \log \frac{1}{\delta} \right) \left( \frac{1}{C_0} + \log \frac{1}{C_0} \right)^T. \tag{99}$$

The proof is completed.

$$\square$$

# J   Mitigating Collapse via Real Data in Recursive Training of Transformer-Based LLMs

In the previous section, we demonstrated that naive recursive training of transformer-based LLMs—without any quality control or external supervision—leads to exponential error growth across generations. This failure arises from unfiltered distributional drift, which causes the model to gradually reinforce and amplify its own mistakes.

To mitigate this collapse, one natural strategy is to incorporate real data into each generation of the training loop. In this section, we formally analyze how the inclusion of ground-truth supervised examples affects recursive training dynamics. Our key result shows that introducing even a moderate proportion of real data can effectively suppress exponential error amplification, ensuring bounded prediction error across generations.

Specifically, we consider a setting where each generation $\mathcal{G}_t$ is trained on a mixture of synthetic and real data. Let $\alpha \in (0, 1)$ denote the proportion of real data retained at each step. We show that this mixing strategy induces a recursive contraction effect, reducing the sensitivity of the model to its own erroneous generations.

**Theorem 8** (Error Bound for Transformer-based LLMs with Real Data). *Let $\mathcal{G}_{\theta_t}^{LLM}(x \mid S)$ denote the conditional distribution of a Transformer-based LLM at recursive step $t$, trained on a mixture of real and synthetic data with real data proportion $\alpha > 0$. Then, under the same notation and assumptions as in Theorem 7, and assuming $\alpha > 1 - \frac{1}{2(1/C_0 - 1)}$ and $T \to \infty$, the prediction error of $\mathcal{G}_{\theta_T}^{LLM}$ is, with probability at least $1 - \delta$, bounded as follows:*

$$\mathrm{Err}(\mathcal{G}_T) \lesssim \epsilon(\gamma) + \frac{1}{\tau}\sqrt{\frac{\bar{D}\log(n\bar{B})}{n}} + \frac{1-\alpha}{C_0}\sqrt{\frac{\bar{D}}{n}}\log(n\bar{B}) \tag{100}$$

Compared to the exponential growth in Theorem 7, the presence of real data introduces a damping effect, effectively anchoring the model and limiting the accumulation of distributional shift.

We now present the proof of this result.

*Proof of Theorem 8.* This proof builds upon the key techniques established in the proofs of Theorem 2 and Theorem 7, particularly the decomposition of error into high-confidence and low-confidence regions, and the recursive control of inverse confidence via total variation distance. We present a concise yet complete version adapted to Transformer-based LLMs.

We analyze a recursive training process where the model $\mathcal{G}_{\theta_t}^{\mathrm{LLM}}$ at generation $t$ is trained on a mixture of real and synthetic data:
$$\mathcal{D}_t = \alpha \cdot \mathcal{D}_{\mathrm{real}} + (1 - \alpha) \cdot \mathcal{D}_{\mathrm{synthetic}},$$
and the induced mixture distribution is:
$$\mathcal{G}_t^{\mathrm{mix}} = \alpha \cdot \mathcal{G}^* + (1 - \alpha) \cdot \mathcal{G}_{\theta_{t-1}}^{\mathrm{LLM}}.$$

**Decomposition of prediction error**

We define the high-confidence region:
$$\mathcal{D}_t^+ := \left\{ x \in \mathcal{X} \,\middle|\, \mathcal{G}_{\theta_0}^{\mathrm{LLM}}(y^* \mid x) \geq 1 - \frac{\tau}{2(1-\alpha)} \right\},$$
and decompose the total error:
$$\mathbb{P}_{x \sim \mathcal{X}} \left[ \mathcal{G}_{\theta_t}^{\mathrm{LLM}}(y^* \mid x) \leq 1 - \tau \right] \leq \mathbb{P}\left[ \mathcal{G}_{\theta_t}^{\mathrm{LLM}}(y^* \mid x) \leq 1 - \tau \,\middle|\, x \in \mathcal{D}_t^+ \right] + \mathbb{P}\left[ x \notin \mathcal{D}_t^+ \right].$$

**Bounding the first term via total variation**

Using the standard TV bound:
$$\left| \mathcal{G}_{\theta_t}^{\mathrm{LLM}}(y \mid x) - \mathcal{G}_t^{\mathrm{mix}}(y \mid x) \right| \leq 2D_{\mathrm{TV}}(\mathcal{G}_{\theta_t}^{\mathrm{LLM}}, \mathcal{G}_t^{\mathrm{mix}}),$$
we get:
$$\mathbb{P}\left[ \mathcal{G}_{\theta_t}^{\mathrm{LLM}}(\boldsymbol{y}_\gamma^+(x) \mid x) \leq 1 - \tau \,\middle|\, x \in \mathcal{D}_t^+ \right] \leq \frac{4}{\tau} D_{\mathrm{TV}}(\mathcal{G}_{\theta_t}^{\mathrm{LLM}}, \mathcal{G}_t^{\mathrm{mix}}).$$

**Bounding the complement using inverse confidence**

Let $C_\gamma := \mathbb{E}_{x \sim \mathcal{X}} \left[ \frac{1}{\mathcal{G}_{\theta_0}^{\text{LLM}}(\boldsymbol{y}_\gamma^+(x)|x)} \right]$. By Markov's inequality:

$$\mathbb{P}\left[x \notin \mathcal{D}_t^+\right] \leq \left(1 - \frac{\tau}{2(1-\alpha)}\right) C_\gamma.$$

**Recursive coverage coefficient control**

Assuming $\alpha > 1 - \frac{1}{2(1/C_0 - 1)}$, we have:

$$C_{\gamma,T} \lesssim \frac{1-\alpha}{C_0} D_{\text{TV}} \log \frac{1}{D_{\text{TV}}}.$$

**Finite-sample generalization bounds**

In Transformer-based LLMs, the finite-sample deviation under PAC-Bayes analysis is:

$$\mathbb{P}_{x \sim \mathcal{X}}\left[\mathcal{G}_T\left(\boldsymbol{y}_\gamma^+(x) \mid x\right) \leq 1 - \tau\right] \lesssim \frac{1}{\tau} \cdot D_{\text{TV}} + \frac{1-\alpha}{C_0} D_{\text{TV}} \log \frac{1}{D_{\text{TV}}}. \tag{101}$$

**Final Results**

Combining all terms, we derive the final error bound:

$$\text{Err}(\mathcal{G}_{\theta_T}^{\text{LLM}}) \lesssim \epsilon(\gamma) + \frac{1}{\tau}\sqrt{\frac{\bar{D}\log(n\bar{B})}{n}} + \frac{1-\alpha}{C_0}\sqrt{\frac{\bar{D}}{n}}\log(n\bar{B}),$$

where $\bar{D}$ is the model's effective dimension, $\bar{B}$ its prior radius, and $n$ the total sample count. The proof is completed. $\qquad\square$

# K Further Discussion on Assumptions and Empirical Validation

## K.1 On the Realism of Assumption 1

This section aims to clarify both the intuition behind Assumption 1 and its alignment with real world LLM behavior.

At its core, Assumption 1 formalizes a confidence calibrated agreement condition: high confidence model predictions are likely to be correct. This is consistent with standard assumptions in self training (e.g., Assumption 2 on page 14 of Huang et al. [2025b]) and aligns with active research on confidence calibration in LLMs, which is a well studied area (see Liu et al. [2025] for a survey).

Critical to its realism, empirical work shows modern LLMs often exhibit strong calibration [Achiam et al., 2023]. For example, Luo et al. [2025] reports small expected calibration errors for pre trained models like Llama-3-8B (3.52%), Qwen-2.5-7B (5.41%), and DeepSeek-V2-Lite (3.39%) on MMLU. This demonstrates a close alignment between confidence and accuracy.

To further validate this, we conducted experiments on the MATH dataset with Phi3.5-Mini, analyzing the log probabilities of correct versus incorrect responses. The results are summarized in Table 1.

As the data shows, correct responses consistently exhibit higher log probabilities, and their distribution stochastically dominates that of incorrect responses. This observation confirms the positive correlation between model confidence and correctness, directly supporting the validity of Assumption 1.

## K.2 Empirical Validation of Self-Verification

To empirically test our theoretical findings on model collapse, we designed an experiment following the setting from Fu et al. [2025]. Specifically, we trained a 12 layer, 8 head GPT-2 model (with a hidden size of 256) to recursively perform in context learning of linear functions from the class:

$$\mathcal{F} = \left\{ f \mid f(x) = w^\top x, \; w \in \mathbb{R}^5 \right\}$$

Table 1: Distribution of Correct and Incorrect Responses by Log Probability on the MATH dataset with Phi3.5-Mini.

| Log Probability Range | Number of Correct | Number of Incorrect |
|---|---|---|
| 0 to -0.6 | 1032 | 83 |
| -0.6 to -1.2 | 413 | 462 |
| -1.2 to -1.8 | 73 | 386 |
| -1.8 to -2.4 | 24 | 95 |
| -2.4 to -3.0 | 6 | 18 |

For each prompt, we sampled $x_1, \ldots, x_k, x_{\text{query}}$ and $w$ independently from $\mathcal{N}(0, I_d)$. The model's task was to predict $y_{\text{query}} = w^\top x_{\text{query}}$. We compared three distinct training strategies:

- **Full Synthetic:** The model was trained solely on synthetic data generated by its predecessor.

- **Mixed:** A combination of fresh real data and synthetic data was used, mixed in a 0.5 ratio.

- **Verification-Based Filtering:** For each training instance, we sampled 20 candidate responses, selected the one with the highest confidence (corresponding to $\gamma = 0$), and trained the model exclusively on this verified data.

The results of these experiments, measured by prediction error over recursive training rounds, are summarized in Table 2.

Table 2: Prediction Error Across Recursive Training Rounds for Different Strategies.

| Strategy | Round 0 | Round 1 | Round 2 | Round 3 |
|---|---|---|---|---|
| Full Synthetic | 0.2418 | 1.1625 | 1.4245 | 1.9523 |
| Mixed | 0.2418 | 0.2824 | 0.3178 | 0.3235 |
| Verification | 0.2418 | 0.2672 | 0.2949 | 0.3105 |

As observed, the error accumulates progressively with more generations of recursive training. This degradation is particularly severe in the full synthetic case, where the error grows rapidly. In contrast, both incorporating real data and applying the self verification mechanism effectively mitigate the increase in loss. This outcome is consistent with our theoretical findings.

### K.3 Directions for Future Experimental Work

To further validate and extend the theoretical contributions of this paper, several experimental directions could be pursued for future work:

1. **Calibration Manipulation Experiment:** An initial experiment could involve selecting a model for recursive training and systematically adjusting its calibration via temperature scaling or other established methods. This process would produce model versions with "low," "moderate," and "high" calibration, where the quality of calibration is quantified by the Expected Calibration Error (ECE).

2. **Performance Correlation Verification:** Following the calibration manipulation, a second experiment could investigate the relationship between calibration and performance degradation. The differently calibrated models would undergo recursive training with self-verification, allowing for a comparison of their error accumulation over successive generations. Such an analysis would provide empirical insights into how a verifier's calibration quality (measured by ECE) correlates with the final model's test error under a self-verification framework.

