# OpenReview forum: "Self-Verification Provably Prevents Model Collapse in Recursive Synthetic Training"
_NeurIPS.cc/2025/Conference — NeurIPS 2025 poster_

### Official Review · Reviewer_pDf5 · 2025-07-02

**Clarity:** 3
**Significance:** 3
**Originality:** 4
**Rating:** 4
**Confidence:** 2

**Summary:**

This paper provides a theoretical investigation into the problem of model collapse in recursively trained generative models, particularly focusing on large language models (LLMs) trained predominantly on synthetic data created by preceding model generations. The authors formalize the recursive training process and demonstrate, via rigorous finite-sample analysis, that naive recursive training can lead to exponential error escalation and performance degradation.

**Questions:**

1. Empirical Support and Simulation: While the theoretical results are promising, the main paper does not present any simulated or empirical results to demonstrate error control in finite-sample settings or to visualize error growth/collapse. Are there toy simulations or practical experiments (even in the appendix) that support the tightness and realism of your bounds? What are the expected practical limits when deploying self-verification in real large-scale LLMs?
2. Robustness to Calibration Errors: The key error bounds hinge on the base verifier model $\mathcal{G}_0$ being well-calibrated. In practice, language models often exhibit calibration issues, and distribution shift may degrade this further. How sensitive are the theoretical guarantees to $\mathcal{G}_0$'s miscalibration or to adversarially bad initial models? What guidance can you provide for identifying when self-verification will fail due to poor calibration?

**Ethical Concerns:**

["NO or VERY MINOR ethics concerns only"]

**Final Justification:**

Frankly, I am not very familiar with theory-oriented papers, so I initially provided an inappropriate score. Considering the opinions of other reviewers and the author's efforts, I am willing to offer support, but I will lower my confidence level accordingly.

**Limitations:**

Yes

**Quality:**

3

**Strengths And Weaknesses:**

**Strengths**
- Strong Theoretical Foundation: The paper offers a well-motivated and thorough theoretical framework analyzing recursive training and self-verification's effect on error propagation. The development of error bounds and the explicit, stepwise derivations in Section 4 show technical rigor and completeness.
- Explicit Theoretical Guarantees for LLMs: Section 5 translates the general theory to the specifics of standard transformer architectures, complete with explicit error bounds involving practical quantities such as sample size, model capacity, and verification parameter settings.
- Actionable Theoretical Results: Theorems 1–4 provide concrete quantitative conditions (e.g., on the minimal verification sample size $N$ and confidence threshold $C_0$) that practitioners could, in principle, guide real-world recursive training protocols by tuning.

**Weakness**
- Lack of Empirical Validation in the Main Paper: The paper provides extensive theoretical analysis but does not include any experimental or simulation results in the main body that demonstrate the practical efficacy or tightness of the bounds. This omission makes it difficult to judge the real-world practical impact or applicability beyond the theory.
- Limited Discussion of Assumptions' Strength and Realism: The theoretical results depend on confidence calibration and internal agreement assumptions (e.g., Assumption 1 in Section 4.1). There is little empirical or interpretive discussion in the main text about how easily these are met in complex, real-world LLMs, or what happens if calibration fails.
- Potential Fragility to Model Miscalibration: The self-verification guarantees rely on the reference model $\mathcal{G}_0$ being well-calibrated and robust to distribution drift. The implications of an initial poor model, selection of the $\gamma$ margin, or the existence of hard adversarial examples are not explored, nor is there a sensitivity analysis included.
- Absence of Empirical Figures, Tables, or Visualizations: There are notably no figures or tables of results (such as error curves, ablation studies, or schematic diagrams) in the main body, which impairs intuitive understanding and makes it harder to connect theory to practice or to evaluate claim substantiation visually. For example, a figure demonstrating the exponential error growth in Theorem 1 or a table comparing theoretical vs practical error rates under various $N, C_0$ would add substantial value.

---

> ### Author Rebuttal · Authors · 2025-07-31
>
> Thank you for your thorough review and constructive comments. We have carefully addressed all your concerns and thoroughly revised the paper, and sincerely hope our responses fully answer your questions.
>
> ---
> **Q1: Empirical Support and Simulation: Though theoretical results are promising, the main paper lacks empirical findings showing error control in finite samples or visualizing error growth. Are there toy simulations or experiments (even in the appendix) supporting the tightness of your bounds? What are the practical limits for deploying self-verification in LLMs?**
>
> **A:** Thank you for recognizing the value of our theoretical contributions, which are central to the learning theory area. Following your suggestion, we conducted experiments during the discussion period to validate our results. We also promise to include additional, more comprehensive experiments in the final version.
>
> Specifically, we follow Fu et al. (2025)’s model collapse setting, training a 12-layer, 8-head GPT-2 (hidden size 256) to recursively in-context learn linear functions:
> $$
> \mathcal{F} = \\{ f | f(x) = w^\top x, w \in \mathbb{R}^d \\},
> $$
> For each prompt, we sampled $x_1, \ldots, x_k, x_{query}$ and $w$ independently from $\mathcal{N}(0, I_d)$. The model predicts $y_{query} = w^\top x_{query}$. We compared three training strategies:
>
> - **Full Synthetic:** Trained solely on synthetic data.
> - **Mixed:** Fresh and synthetic data were mixed in a 0.5 ratio.
> - **Verification-Based Filtering:** Sampled 20 candidates, selected the highest-confidence prediction ($\gamma = 0$), and trained on verified data exclusively.
>
> The results of these experiments are summarized below:
>
> | Round       | 0     | 1     | 2     | 3     |
> |------------------|--------|--------|--------|--------|
> | Full Synthetic   | 0.2418 | 1.1625 | 1.4245 | 1.9523 |
> | Mixed            | 0.2418 | 0.2824 | 0.3178 | 0.3235 |
> | Verification     | 0.2418 | 0.2672 | 0.2949 | 0.3105 |
>
> As observed, the error accumulates progressively with more generations, particularly in the full synthetic case, where the error grows rapidly. In contrast, incorporating real data and applying the verification mechanism effectively reduce the loss, consistent with our theoretical findings.
>
> **Regarding the practical limits of self-verification**, we agree that scaling self-verification in LLMs presents practical challenges. A key constraint is the computational overhead introduced by candidate sampling—for instance, generating $N = 20$ outputs per input to enable verification.
>
> To address this, we outline potential mitigations in the revised manuscript, such as adaptive sampling (e.g., reducing $N$ for inputs with low predictive uncertainty) or approximate filtering strategies, which aim to strike a balance between efficiency and verification performance.
>
> ---
> **Q2: Limited Discussion of Assumptions' Strength and Realism: The theoretical results rely on confidence calibration and internal agreement assumptions (e.g., Assumption 1), but the main text rarely discusses empirically or interpretively how easily these hold for real-world LLMs.**
>
> **A:** We appreciate the reviewer’s focus on Assumption 1. We have strengthened the manuscript to address this by clarifying both the intuition behind the assumption and its alignment with real-world LLM behavior.
>
> At its core, Assumption 1 formalizes a confidence-calibrated agreement condition: high-confidence model predictions are likely to be correct. This is consistent with standard assumptions in self-training (e.g., Assumption 2 on page 14 of [1]; the abstract of [2]) and aligns with active research on confidence calibration in LLMs—a well-studied area (see [3] for a survey).
>
> Critically, empirical work shows modern LLMs often exhibit strong calibration. For example, [4] reports small expected calibration errors for pre-trained models like Llama-3-8B (3.52%), Qwen-2.5-7B (5.41%), and DeepSeek-V2-Lite (3.39%) on MMLU—demonstrating close alignment between confidence and accuracy.
>
> To further validate this, we conducted experiments on the MATH dataset using Phi3.5-Mini, analyzing log-probabilities of correct vs. incorrect responses:
>
> | Log Prob Range   | Num. Correct | Num. Incorrect |
> |------------------|--------------|----------------|
> | 0 to -0.6        | 1032         | 83           |
> | -0.6 to -1.2     | 413          | 462        |
> | -1.2 to -1.8     | 73           | 386          |
> | -1.8 to -2.4     | 24           | 95           |
> | -2.4 to -3.0     | 6            | 18            |
>
> Correct responses consistently show higher log-probabilities, with their distribution stochastically dominating that of incorrect ones. This confirms the positive correlation between confidence and correctness, directly supporting Assumption 1.
>
> ---
> **Q3: Robustness to Calibration Errors: The key error bounds depend on well-calibrated base verifier $\mathcal{G}_0$, but LLMs often have calibration issues, worsened by distribution shifts. How sensitive are theoretical guarantees to $\mathcal{G}_0$'s miscalibration or adversarial initial models? What guidance exists for identifying self-verification failures due to poor calibration?**
>
> **A:** We thank the reviewer for raising this important point. Our theoretical framework explicitly controls the dependence on the calibration quality of the base verifier $\mathcal{G}_0$ via the term $ε(γ)$, which quantifies the correctness of high-confidence predictions. For instance, in Theorem 4 (*Error Bound for Transformer-based LLMs with Self-Verification*), the leading error term is:
> $$
> ε(γ) +(\frac{1}{τ} + \frac{1}{NC_0}log\frac{1}{τ})\cdot \frac{1 }{\sqrt{n}} log\frac{1}{\delta} + \frac{1}{N}log\frac{1}{τ},
> $$
> which implies that poor calibration in $\mathcal{G}_0$ (i.e., large $ε(γ)$) will directly degrade the final bound. In this case, the guarantee gracefully degrades to an $O(ε(γ))$ bound, which signals when self-verification may become unreliable.
>
> Notably, model collapse research focuses on scenarios where *the initial model performs well* (to isolate degradation from recursive training), rather than adversarially poor $\mathcal{G}_0$. Assuming a severely miscalibrated $\mathcal{G}_0$ would conflate inherent model failure with collapse due to recursive dynamics—contradicting the core motivation of studying collapse.
>
> In practice, recent studies have shown that modern pre-trained LLMs are often well-calibrated. For example, [4] reports small expected calibration errors (ECE) on MMLU: LLaMA-3-8B (3.52%), Qwen-2.5-7B (5.41%), DeepSeek-V2-Lite (3.39%), indicating strong alignment between confidence and accuracy. As noted in Q2, our additional experiments with Phi-3.5-Mini on MATH further validate this assumption, showing that high-confidence predictions are typically correct.
>
> For extreme miscalibration, while our bounds loosen, confidence calibration in LLMs is well-studied with effective solutions (see [1] for a survey). To guide practice, we recommend measuring ECE on a validation set and applying calibration (e.g., temperature scaling) if it exceeds a reasonable threshold. These steps have been incorporated into the manuscript to help identify and mitigate calibration issues prior to deploying self-verification.
>
> ---
> **Q4: Absence of Empirical Visualizations: The main text lacks figures/tables, hindering intuitive understanding and theory-practice connection. For example, a figure on Theorem 1’s exponential error growth or a table comparing theoretical vs. practical error rates under varied $N$, $C_0$ would add significant value.**
>
> **A:** We apologize for the absence of figures and tables in the main text, which hinders intuitive understanding. Per NeurIPS rebuttal policies (prohibiting PDF/link uploads), we cannot include visualizations here, but we firmly commit to adding the requested elements in the final version.
>
> Specifically:
>
> - For illustrating error growth in Theorem 1:  The experiment in our Q1 response shows rapid error escalation in the full synthetic setting (consistent with predicted collapse) and that our verification mechanism effectively prevents this. We will present these as figures (error curves across training rounds for different regimes) and add more comprehensive results.
>
> - For clarifying mechanisms: We will add a schematic diagram of the verification architecture to the main text, aiding intuition about how filtering operates.
>
> **Regarding error rates under varying $N$ (number of candidates) and $C_0$:**
>
> - $C_0$ is a data- and model-dependent quantity (not tunable) characterizing worst-case confidence, consistent with standard model collapse literature.
>
> - $N$ is adjustable, and we conducted additional experiments (same setup as Q1: 12-layer GPT-2 on regression) with $N = 1, 10, 20, 30, 40, 50$.  Note that $N=1$ means using only synthetic data per round without verification, equivalent to the Full Synthetic baseline (naive recursive training without filtering).
>
> | Round   | $N=1$  | $N=10$ | $N=20$ | $N=30$ | $N=40$ | $N=50$ |
> |---------|--------|--------|--------|--------|--------|--------|
> | 0       | 0.2418 | 0.2418 | 0.2418 | 0.2418 | 0.2418 | 0.2418 |
> | 1       | 1.1625 | 0.3438 | 0.2672 | 0.2598 | 0.2647 | 0.2520 |
> | 2       | 1.4245 | 0.3803 | 0.2949 | 0.2815 | 0.2798 | 0.2889 |
> | 3       | 1.9523 | 0.4052 | 0.3105 | 0.3193 | 0.2967 | 0.3051 |
>
> The table confirms that error decreases with increasing $N$, validating the efficacy of self-verification. The drop is steepest from $N=1$ to 20, then plateaus, suggesting $N=20$ balances cost and effectiveness.  These results will be visualized as error curves in the final manuscript, showing the relationship between $N$ and performance.
>
> ---
> **References**
>
> [1] *Self-evolved reward learning for LLMs.*
>
> [2] *Self-improvement in language models: The sharpening mechanism.*
>
> [3] *Uncertainty quantification and confidence calibration in large language models: A survey.*
>
> [4] *Your Pre-trained LLM is Secretly an Unsupervised Confidence Calibrator.*

---

> > ### Comment · Reviewer_pDf5 · 2025-07-31
> > **Thanks for authors rebuttal and rating raising**
> >
> > Thank you for the author's rebuttal. Frankly, I am not very familiar with theory-oriented papers, so I initially provided an inappropriate score. Considering the opinions of other reviewers and the author's efforts, I am willing to offer support, but I will lower my confidence level accordingly.

---

> > > ### Author Response · Authors · 2025-08-01
> > > **Thank You for Your Support**
> > >
> > > Thank you for your kind support and for increasing the rating. We are truly grateful for your positive assessment of our responses during the discussion period.

---

### Official Review · Reviewer_otAB · 2025-07-02

**Clarity:** 3
**Significance:** 3
**Originality:** 4
**Rating:** 4
**Confidence:** 4

**Summary:**

The authors consider a recursive training scheme in which a model is trained repeatedly on its outputs (plus maybe some real data or subject to some quality-amplifying sampling scheme). This approach is known to lead to poor model performance (e.g., model collapse) if the outputs are used naively, but there are two popular mitigations in the literature: 1) the injection of some real data at each training run and/or 2) the filtering of the synthetic data to remove low-quality contributions. The authors propose a particular approach for the latter that involves using the initial trained model to reweight the sampling distribution when the new round of training data is generated. They then present a theoretical framework for analyzing the test error of the final model after T rounds of fitting. They show that this framework allows for the study of previous approaches to mitigating model collapse and accommodates the analysis of more sophisticated black box models (e.g., transformers) so long as it is possible to bound the TV distance between the data-generating distribution and the fitted statistical model.

**Questions:**

* Could you please more carefully define $\mathcal{G}_0(\cdot \mid x)$ - in line 117, it's defined as a function mapping $\mathcal{X} \to \mathcal{Y}$, but later on, it's defined as a probability measure. I found this quite confusing.

* I find Assumption 1 hard to parse - in particular, what is the order of quantifiers (is this tau the same for all gamma or is it gamma-dependent?) and how credible is this assumption? I am not particularly compelled by the citation to Huang et al. 2025, but the practical validity of this assumption seems crucial to this framework's value. I would add more exposition here to both parse the assumption and explain how the reader should think about it. It also seems like this assumption rules out randomness in y | x given that y*(x) is seemingly defined as a deterministic function?

* Please define the coverage coefficient outside of the statement of Theorem 1 - it seems like an important object in your paper and it's too hard to find. It is also sometimes indexed by t and sometimes implicitly indexed (I assume by T). I would be more consistent with this.

* I am completely open to adjusting my score upwards if the authors adjust their comparison to previous work (or convince me that their description is fair) and/or introduce some additional discussion of what this framework enables. I think this work would be a clear accept if it had empirical confirmation of a new claim resulting from this theory.

**Ethical Concerns:**

["NO or VERY MINOR ethics concerns only"]

**Final Justification:**

I don't feel totally comfortable raising my score further (the validation presented is not thoroughly convincing to me), but I am supportive of the paper's publication. I think relative to the other papers I read in this area (and also published in venues like NeurIPS), the contributions here are of a similar magnitude. It also seems that many of my clarity concerns were addressed by the authors.

**Limitations:**

Yes

**Quality:**

3

**Strengths And Weaknesses:**

**Quality**
I think the submission is certainly technically sound and is a substantial and interesting manuscript. It is always difficult to separate criticisms neatly into these four buckets, but I am somewhat concerned about the set of testable predictions that this theory enables. I will elaborate more on these concerns within the significance subsection and in the questions/comments.

**Clarity**
I think the paper is generally clearly written, but I would encourage the authors to use fewer paragraph headings/remarks. If every paragraph begins with a bold title/remark, it no longer really draws the reader's attention. There are also a few notational concerns that I will raise in the questions/comments section.

**Significance**
First, I would like to acknowledge that the paper includes non-trivial technical contributions and nicely generalizes some observations made in the previous literature. I would also disclaim that I am not a researcher in the area of synthetic data / model collapse (though I do have a theoretical background and am familiar with the technical tools used by the authors). To understand the significance of this work, I  familiarized myself with two papers cited by the authors (Feng et al. and El Firdoussi et al.). To that end, I take some issue with how the authors compare and contrast this manuscript with previous work. First, while the authors instantiate a somewhat more complex verification mechanism for improving the training data at each iteration, it is no less external (the first model is used as the verifier at all iterations - for all practical purposes, we can think of it as an external verifier in the recursive context). In addition, for T = 1, Feng et al. also consider self-verification and argue that it can be surprisingly effective (both theoretically and empirically). The criticism of the use of Bernoulli random variables also strikes me as misplaced. It's simply a tool used in the other work to determine when the synthetic data agrees with the real data (this agreement is also crucial to the authors' results: see the prominent role of eps(gamma)). For the second and fourth contributions, I also agree that the theoretical framework introduced here accommodates more models and is ostensibly finite-sample, but in doing so, the authors have introduced terms that are harder to get a practical handle on and make credible statements about. In a nutshell, the authors have avoided the restriction of specifying a data-generating distribution and model-fitting procedure by defining a bound in terms of a TV distance between the DGP and the learned model. To me, this object is harder to reason about and their analysis is subsequently predicated on Assumption 1 (which is also very hard for me to think about or test - if the authors could establish that such a tau appears to exist empirically for popular models that would certainly help). I think the contributions labeled as (2)/(3) are the core contributions of this paper and make it worthy of publication. However, to justify the framework's generality (and claim that as a positive), the authors should make testable predictions about complicated models that previous theory failed to do. The authors do not provide such predictions or empirical evidence here, so the value of this generality is not as clear as it could be. Nevertheless, while my criticisms appear lengthy, I would like to emphasize that I still believe this work to be substantial and I expect it would be of significant interest.

**Originality**
Though I am not an expert in synthetic data/model collapse and I would defer on this to someone with more expertise in this exact area...judging from the two papers I carefully read, this paper is certainly sufficienty original for publication at NeurIPS.

---

> ### Author Rebuttal · Authors · 2025-07-31
>
> Thank you for your constructive comments and kind support! We've carefully addressed all concerns and revised the paper accordingly.
>
> ---
> **Q1: Although the verification mechanism is more complex, it still relies on the first model as a fixed verifier—functionally external throughout the recursive process.**
>
> **A:** We appreciate this insight. Though $\mathcal{G}_0$ originates within the system, its frozen status makes it effectively an external verifier in the recursive process.
>
> That said, our design offers a practical and principled alternative to traditional external supervision. Standard verification methods typically depend on large-scale human annotations or auxiliary fine-tuned models—both costly in terms of labor and computation, and increasingly impractical as high-quality real data becomes scarce.
>
> In contrast, our framework avoids such overhead by reusing $\mathcal{G}_0$ as a stable verifier anchored in real data. This preserves the benefits of external verification (e.g., consistency and filtering) while maintaining the self-contained, scalable nature of self-learning.
>
> ---
> **Q2: The criticism of Bernoulli variables seems misplaced—they're just a tool to measure synthetic-real data agreement, also central to the authors' results (see $ε(γ)$).**
>
> **A:** Thanks. We have revised the manuscript to clarify the role of Bernoulli variables in Feng et al., and to position our $ε(γ)$ as an analogous construct within our framework.
>
> To clarify, both approaches share analogous roles: Feng et al. use Bernoulli random variables to model synthetic-real data agreement, abstractly capturing alignment likelihood with ground-truth labels. In our setting, $ε(γ)$ is analogous—it characterizes the probability that when the model assigns most probability mass to high-quality verification-filtered data, its prediction matches the true label.
>
> ---
> **Q3: The authors avoid specifying a data-generating process, using TV distance instead—though this quantity is harder to reason about.**
>
> **A:** Thanks. We agree TV distance is hard to operationalize, but we use it deliberately as it has become a standard analytical tool in recent model collapse theory (e.g., Dohmatob et al. 2024c; Fu et al. 2024, 2025; Seddik et al. 2024), enabling study of recursive training dynamics without restrictive assumptions on the data-generating process.
>
> That said, we fully recognize the importance of grounding such abstract quantities in practical settings. Specifically, in Theorem 4, our manuscript leverages PAC-Bayesian analysis tailored to transformer-based LLMs to derive a finite-sample bound for the TV distance term, connecting our theoretical bounds to state-of-the-art architectures with concrete guarantees.
>
> ---
> **Q4: Could you clearly define $\mathcal{G}_0(\cdot | x)$ in line 117? It is described as a function mapping but later as a probability measure, which is confusing.**
>
> **A:** Thanks. We have revised the definition of $\mathcal{G}_0(\cdot | x)$ to clarify that $\mathcal{G}_0: \mathcal{X} \to \Delta(\mathcal{Y})$ denotes a conditional distribution mapping an input $x \in \mathcal{X}$ to a distribution over responses—i.e., $\mathcal{G}_0(y | x)$ is the probability assigned to response $y$ given $x$. This correction aligns the formal definition with its use as a probability measure. Line 117 has been updated accordingly.
>
> ---
> **Q5: Assumption 1 is hard to parse: Is $τ$ the same for all $γ$ or $γ$-dependent? How credible is it? Also, since $y^*(x)$ seems deterministic, the assumption appears to rule out randomness in $y | x$.**
>
> **A:** Thanks. We address each point and summarize the revisions below.
>
> ### 1. Intuition and Credibility of Assumption 1
>
> This assumption encodes a confidence-calibrated agreement: predictions with sufficiently high confidence are likely to be correct. Specifically, if the model assigns probability mass $\ge 1 - τ$ to the verified high-quality set, then the probability that its prediction matches the ground truth $y^*(x)$ is at least $1 - ε(γ)$.
>
> This aligns with standard self-training assumptions (e.g., Assumption 2 on page 14 of [1]; the abstract of [2]) and active research on LLM confidence calibration (surveyed in [3]). It is realistic: modern LLMs often show strong calibration, with [4] reporting small expected calibration errors for Llama-3-8B (3.52%), Qwen-2.5-7B (5.41%), and DeepSeek-V2-Lite (3.39%) on MMLU, indicating close confidence-accuracy alignment.
>
> ### 2. Parameterization and Determinism in Assumption 1
>
> Assumption 1 treats $τ$ as a fixed constant for notational simplicity, but our analysis holds for any $τ \in (0, 1)$. Specifically, the bound for
> $$
> P [ \mathcal{G}_T ( y_γ^+(x) | x ) \geq 1 - τ]
> $$
> (see Equation 31, page 22) remains valid regardless of the choice of $τ$. Moreover, $τ$ does not need to depend on $γ$, since variation in verification strictness across $γ$ is already captured by $ε(γ)$.
>
> Assumption 1 also implicitly assumes a deterministic ground-truth label function $y^*(x)$. This reflects a realizable setting commonly used in self-training theory (e.g., [2]), enabling a clean and tractable convergence analysis. This clarification has been added to the manuscript.
>
> ### 3. Empirical Support for Assumption 1
>
> To strengthen the credibility of Assumption 1, we conducted additional experiments analyzing the relationship between model confidence and prediction correctness. Using the Phi3.5-Mini model on the MATH dataset (following [2]), we sampled responses and measured log-probabilities:
>
> | Log Prob Range   | Num. Correct | Num. Incorrect |
> |------------------|--------------|----------------|
> | 0 to -0.6        | 1032         | 83            |
> | -0.6 to -1.2     | 413          | 462          |
> | -1.2 to -1.8     | 73           | 386           |
> | -1.8 to -2.4     | 24           | 95             |
> | -2.4 to -3.0     | 6            | 18            |
>
> These results confirm that correct responses consistently receive higher log-probabilities, validating the positive correlation between confidence and correctness. Formally, the log-probability distribution for correct responses stochastically dominates that for incorrect responses.
>
> **References**
>
> [1] *Self-evolved reward learning for LLMs.*
>
> [2] *Self-improvement in language models: The sharpening mechanism.*
>
> [3] *Uncertainty quantification and confidence calibration in large language models: A survey.*
>
> [4] *Your Pre-trained LLM is Secretly an Unsupervised Confidence Calibrator.*
>
> ---
> **Q6: Please define the key coverage coefficient outside Theorem 1 (hard to find) and standardize its indexing.**
>
> **A:** Thanks. We now define the recursive coverage coefficient early in the technical section as
>
> $$
> C_{γ,t} := E_{x} [ 1 / \mathcal{G}_t(y_γ^+(x) | x) ],
> $$
>
> with $t$ indexing the training iteration. To clarify the notation in Theorem 1: The coefficient $C_{γ}$ referenced there corresponds to $C_{γ,0}$. These revisions improve clarity and consistency.
>
> ---
> **Q7: I am completely open to adjusting my score upwards if the authors adjust their comparison to previous work. I think this work would be a clear accept if it had empirical confirmation of a new claim resulting from this theory.**
>
> **A:** Thanks. We clarified our comparison and conducted additional experiments to support our theoretical claims.
>
> ### 1. Clarifying Our Comparison with Prior Work
>
> We have revised the manuscript to emphasize two core contributions and clarify three key comparisons:
>
> - General and Practical Framework: Unlike prior work (Feng et al., 2025; Firdoussi et al., 2025) focused on Gaussian mixtures or linear classifiers, our framework extends to Transformer-based LLMs, addressing realistic settings.
>
> - Recursive vs. Non-Recursive Training: While prior work studied non-recursive setups ($T = 1$), we focus on recursive training—where model collapse primarily occurs—explicitly modeling its iterative dynamics.
>
> - On the verifier $\mathcal{G}_0$: As noted in Q1, it acts as an external verifier but avoids the need for additional annotations or stronger auxiliary models, enhancing the practicality of the framework.
>
> - On Bernoulli variables: As discussed in Q2, we now clarify their alignment with our $ε(γ)$—both quantify agreement—while acknowledging their utility in prior work.
>
> - On TV distance: As explained in Q3, we recognize its practical challenges but emphasize our finite-sample bounds (Section 5) tailored for Transformer-based LLMs, bridging theory and practice.
>
> ### 2. Experimental Validation of Theoretical Results
>
> To validate our theory, we follow Fu et al. (2025)’s model collapse setting, training a 12-layer, 8-head GPT-2 (hidden size 256) to recursively in-context learn linear functions:
> $$
> \mathcal{F} = \\{ f | f(x) = w^\top x, w \in \mathbb{R}^d \\},
> $$
> For each prompt, we sampled $x_1, \ldots, x_k, x_{query}$ and $w$ independently from $\mathcal{N}(0, I_d)$. The model predicts $y_{query} = w^\top x_{query}$. We compared three training strategies:
>
> - Full Synthetic: Trained solely on synthetic data.
> - Mixed: Fresh and synthetic data were mixed in a 0.5 ratio.
> - Verification-Based Filtering: Sampled 20 candidates, selected the highest-confidence prediction ($γ = 0$), and trained on verified data exclusively.
>
> The results of these experiments are summarized below:
>
> | Round              | 0      | 1      | 2      | 3      |
> |--------------------|--------|--------|--------|--------|
> | Full Synthetic     | 0.2418 | 1.1625 | 1.4245 | 1.9523 |
> | Mixed              | 0.2418 | 0.2824 | 0.3178 | 0.3235 |
> | Verification       | 0.2418 | 0.2672 | 0.2949 | 0.3105 |
>
> As observed, the error accumulates progressively with more generations, particularly in the full synthetic case, where the error grows rapidly. In contrast, incorporating real data and applying the verification mechanism effectively reduce the loss, consistent with our theoretical findings.
>
> Furthermore, we promise to include additional experiments to further support our results in the final version.

---

> ### Comment · Reviewer_otAB · 2025-08-08
>
> Thank you for addressing the clarity concerns raised in the original review. I appreciate the thorough response, though it is still not clear to me that this experiment motivates all of the theory presented in this paper. To summarize my understanding, the core idea of this paper is that using a good verifier for synthetic data keeps the DGP from drifting too far from the ground-truth. There's some limited empirical support for this strategy presented in the rebuttal and a fair amount of theoretical exposition. I'm still not fully convinced of its operational value - your theory makes a very specific claim relating calibration to accumulated error. So, it seems like it would be interesting to see if the effect of the different verifiers you test is predicted by their well-calibratedness? I would suspect that this relationship is somewhat weak in practice - surely I could use an uncalibrated and post-trained GPT-5 to verify data much more effectively than a calibrated small model, but there may be some subtlety here re: the original DGP. Either way, I think this empirical validation is a fine start, but insufficient to justify raising my score further. That being said, my quibbles aside, I think this is a solid paper and I am supportive of its publication.

---

> > ### Author Response · Authors · 2025-08-08
> > **Response to Reviewer Feedback**
> >
> > Thank you sincerely for your recognition of our work and support for its publication. We deeply respect your judgment on the score, and greatly appreciate your continued attention to the paper’s details. Your concerns are valid and will provide valuable guidance for strengthening our work. We will address your points through the following revisions:
> >
> > ### 1. Regarding the connection between experiments and theory
> > We fully agree with the core idea you summarized: using a well-designed verifier to validate synthetic data is key to preventing DGP from drifting excessively from the ground truth. We apologize that due to time constraints during the reviewer-author discussion period, our current experiments are limited in scale. While these results are preliminary, they do provide early evidence for the effectiveness of our verification mechanism—specifically, showing that the verification loop significantly reduces error accumulation compared to a full synthetic loop.
> >
> > We recognize that while the paper’s core contribution lies in the development of the theoretical framework, empirical validation of theoretical nuances is crucial. To address this, we commit to adding more experiments in the final version of the paper, with additional tests designed to cover more theoretical details, thereby strengthening the link between theory and practice.
> >
> > ### 2. Regarding the relationship between verifier calibration and verification efficacy
> > We appreciate your insight that the relationship between calibration and verification performance may be complex, and for your observation that an uncalibrated but post-trained large model (like GPT-5) might outperform a calibrated smaller model in practice, we would like to clarify that within our theoretical framework, the verifier is defined as the initial model in recursive training (it is worth noting that in the model collapse literature, models in the loop share the same architecture). This choice is driven by a practical motivation: for frontier models, there may often be no stronger external model available, and training a more powerful one would require substantial compute and fresh real data—resources that are frequently exhausted in real-world scenarios. This makes self-learning from one’s own generations a necessary and realistic paradigm, focusing on optimizing the use of existing models rather than seeking larger external models.
> >
> > That said, your suggestion to explore how verifier calibration impacts error accumulation is invaluable. We commit to adding dedicated experiments in the final version to systematically investigate how the verifier’s well-calibratedness affects model performance. We believe these expanded experiments will better illuminate the relationship between verifier calibration and the efficacy of synthetic data verification.
> >
> > We are grateful for your constructive feedback, which will undoubtedly help improve the rigor and completeness of our work. We will promptly implement these revisions and look forward to presenting a stronger version of the paper. Thank you again for your thoughtful guidance.

---

> > > ### Comment · Reviewer_otAB · 2025-08-08
> > >
> > > Thanks! Perhaps I should clarify exactly what I meant though: the unique component of your theory is your identification of confidence calibration (at least via $\tau$ and $\epsilon(\gamma)$) as the appropriate measure of self-verification quality. An experiment showing that self-verification works does not by itself validate this theory.
> > >
> > > I'm not certain that using the first model as a verifier is a necessary or realistic restriction. For instance, people will want to train smaller models on synthetic data and may use a large (and more expensive) model as a verifier. But even if you don't want to violate that assumption in an experiment, you can still use some temperature scaling to worsen/improve the calibration of an existing model and show that the resulting degradation/improvement in recursive training performance matches your theory.
> > >
> > > Thanks for the response and my apologies for my late engagement in this process.

---

> > > > ### Author Response · Authors · 2025-08-08
> > > > **Response to Reviewer's Further Clarifications**
> > > >
> > > > Thank you for your further clarification and valuable suggestions—these insights have helped us more precisely focus on the core of validating our theory. We fully understand your key concern: the uniqueness of our theory lies in identifying confidence calibration as a critical measure of self-verification quality, and thus dedicated experiments are needed to directly validate the link between calibration and recursive training performance, rather than merely demonstrating that self-verification works.
> > > >
> > > > Given that the reviewer-author discussion period will conclude in just a few hours, we regret that we cannot provide additional experimental results within this window. However, we firmly commit to supplementing the following experiments in the final version of the paper:
> > > >
> > > > 1.  Calibration manipulation experiment: We will select the initial model in recursive training and systematically adjust its calibration via temperature scaling (or other calibration methods) to generate three versions with "low calibration," "moderate calibration," and "high calibration." We will quantify calibration quality using the expected calibration error (ECE) as the key metric.
> > > >
> > > > 2. Performance correlation verification: Within the same synthetic data loop with self-verification, we will compare the recursive training performance of the three versions in terms of error accumulation. This will directly verify whether "better calibration (lower ECE) leads to slower performance degradation"—a core prediction of our theory regarding the mapping between the verifier’s calibration error and the final model’s test error
> > > >
> > > > Regarding the assumption of verifier selection, we agree that the practical scenario of "using a large model as a verifier for smaller models" is highly valuable. We plan to clarify in the discussion section that the current theory selects the initial model as the verifier, a choice intended to focus on the core mechanism of "self-verification" (especially in scenarios where no stronger external model is available). We will extend our theoretical framework to cross-model verification scenarios in the future; specifically, we will assume the verifier is a more powerful model with stronger ability to distinguish high-quality data, which will then be used to filter synthetic data. Importantly, the conclusions from our supplementary "calibration manipulation experiments" will help verify whether calibration remains a key variable regardless of model scale—laying groundwork for such extensions.
> > > >
> > > > Thank you again for your patience and guidance. We will promptly implement these experiments and revisions to strengthen the connection between our theory and empirical evidence.

---

### Official Review · Reviewer_TBjc · 2025-07-08

**Clarity:** 3
**Significance:** 4
**Originality:** 3
**Rating:** 5
**Confidence:** 2

**Summary:**

This paper tackles the problem of model collapse in recursive training scenarios where generative models are trained on synthetic data from previous model generations. The authors propose a self-verification framework that uses internally estimated confidence scores to filter synthetic outputs without requiring external validation. Their approach anchors verification to an initial model trained on real data, which serves as a fixed reference point across generations. The theoretical contribution includes finite-sample error bounds across three recursive training regimes and extends to transformer-based LLMs with convergence guarantees. The core idea is that by filtering generated outputs to stay within a margin of the highest-scoring candidates (as evaluated by the initial model), they can prevent distributional drift and maintain training stability even in fully synthetic settings.

**Questions:**

Again, these questions come from a place of curiosity as I was trying to understand the paper, not as formal complaints. Answering them would really help clarify the work for me.

Confidence Scores: I'm still a bit lost on how the s_verify score is calculated. You define it as log G₀(y | x), which makes me wonder, is this basically like perplexity? And when you generate candidates to score, are you assuming they are all from greedy sampling, or does this account for other methods?
Filtering Mechanism: If I'm understanding correctly, the step where you keep candidates within a margin γ of the best score seems a lot like a form of top-p sampling, but over longer generations rather than just the next token. Are you essentially using this to ensure the "new" model's distribution stays aligned with the baseline G₀ model? Is that the right way to think about it?
Teacher-Student Analogy: The whole setup reminds me of a teacher-student framework, where G₀ is the fixed "teacher" and every subsequent model G_t is a "student" trying to learn its distribution. Would you consider this a fair or useful analogy for your framework?
Reusing Real Data: In Remark 3, you mention the approach relies on an "ongoing stream of high-quality real data." What would happen if you just reused the same real data from the first model in every generation? Does it absolutely have to be new data to prevent collapse?
Training Regimes: What are these three regimes you mention? It would be helpful to briefly name or describe these in the introduction.
Data Source: Just out of curiosity, the framework assumes training on data from a model's direct predecessor. Have you considered what it would be like if the model was trained on data from a stronger or better model instead?

**Ethical Concerns:**

["NO or VERY MINOR ethics concerns only"]

**Final Justification:**

The authors acknowledged my concerns and have mentioned that they'll update the paper to include some details that I found were missing. I already rated this paper pretty high and so I don't think there's any need to revise my reading and so I'm keeping it as is.

**Limitations:**

The authors do touch upon limitations, like the dependency on real data, which is good. However, the discussion feels a bit scattered throughout the paper. It might be more impactful to have a consolidated section that discusses the practical limitations more directly (e.g., the computational cost of sampling N candidates for every single training point, or scenarios where the initial G₀ model might be a poor anchor).

**Quality:**

4

**Strengths And Weaknesses:**

I should start by saying that I’m not a theory expert, so my ability to properly judge the theoretical contributions here is limited. That said, I do like the paper and the core idea.

The main strength is that this paper tackles a very significant and practical problem. The idea of using the initial model (G₀) as a stable anchor is intuitive and clever. It presents a potential path forward for training that doesn't rely on a constant, expensive stream of new human-annotated data.

My main difficulty with the paper was its clarity. I had to reread sections multiple times because key concepts felt undefined. For instance, I still don't have a clear picture of what "internal confidence scores" actually are or how they are computed in practice. The motivation for removing external verification isn't clear either - it cost? GPU hours? There also seemed to be a lot of overlap between the Introduction and the Related Work section, which made the opening a bit repetitive. While the central idea is strong, the explanation could be clearer for a reader like me.

---

> ### Author Rebuttal · Authors · 2025-07-31
>
> Thank you for your constructive comments and kind support! We have carefully addressed all concerns below and thoroughly revised the manuscript. We hope our responses resolve your questions.
>
> ***
>
> **Q1: The motivation for removing external verification isn't clear - it cost? GPU hours?**
>
> **A:** We appreciate this question and have clarified the motivation in the revised introduction.
>
> As highlighted in our revisions, external verifiers inherently depend on two scarce resources: large-scale human annotations (becoming increasingly scarce as high-quality real data is depleted) and additional fine-tuned auxiliary models. This incurs significant labor costs and computational overhead (e.g., GPU hours for continuous training/retuning), which is exacerbated in recursive scenarios where distribution shifts demand constant adaptation.
>
> Our self-verification framework avoids these issues by using the initial model as a stable anchor, eliminating the need for ongoing annotations or auxiliary models to bypass external verification’s resource constraints in recursive training.
>
> ***
>
> **Q2: Confidence Scores: How is $ s_{verify} = \log \mathcal{G}_0(y|x) $ calculated—similar to perplexity? When generating candidates for scoring, are they from greedy sampling or other methods?**
>
> **A:** We appreciate this question, as it helps clarify a key technical detail.
>
> The verification score  $ s_{verify}(y|x) = \log \mathcal{G}_0(y|x) $ uses $\mathcal{G}_0$ to evaluate outputs $y$ from later models: $x$ and $y$ are concatenated into a sequence fed to $\mathcal{G}_0$, whose token-level log-probabilities (from position logits) are summed for total log-likelihood.
>
> Regarding its relation to perplexity: While both use log probabilities, perplexity measures average uncertainty over a dataset or sequence, whereas  $s_{verify}$  ranks or filters individual outputs based on alignment with $\mathcal{G}_0$.
>
> Regarding the generation process: Candidate outputs are sampled from  $\mathcal{G}_{t-1}(\cdot | x)$  via stochastic decoding (e.g., temperature, top-p sampling), not greedy decoding. This randomness promotes diversity, enabling effective verification-based filtering.
> ***
>
> **Q3: Filtering Mechanism: Keeping candidates within margin $γ$ of the best score resembles top-p sampling for longer generations. Is this to align the new model’s distribution with $\mathcal{G}_0$?**
>
> **A:** We agree our filtering shares top-p sampling’s core intuition (retaining high-probability outputs while promoting diversity) but with key differences, outlined below.
>
> Standard top-p sampling selects the next token from high-probability options (token-level). Ours filters full candidate sequences (sequence-level). Also, top-p uses a fixed number of options, while our $γ$-margin filtering is score-adaptive—retained candidates depend on how many fall within the best sample’s high-quality neighborhood, adapting to input contexts and score variances.
>
> As noted, its core purpose is aligning the new model’s distribution with $\mathcal{G}_0$: filtering out candidates that deviate too far from $\mathcal{G}_0$’s judgments prevents cumulative drift from real data, critical for avoiding collapse in recursive training.
> ***
>
> **Q4: Teacher-Student Analogy: Is it fair or useful to view $\mathcal{G}_0$ as a fixed "teacher" and subsequent $\mathcal{G}_t$ as "students" learning its distribution?**
>
> **A:** We appreciate this insightful analogy, which captures a key dynamic of our framework.
>
> Your view of $\mathcal{G}_0$ as a fixed teacher and $\mathcal{G}_t$ as students captures our setup: $\mathcal{G}_0$, trained on real data, guides via verification score, indicating how well student outputs align with its knowledge.
>
> There are two critical distinctions from traditional teacher-student frameworks:
>
> 1. Architectural parity over superiority: Unlike traditional setups requiring a stronger teacher, $ \mathcal{G}_0 $ and $ \mathcal{G}_t $ share the same architecture. $ \mathcal{G}_0 $’s authority stems from its real-data training, not model size or performance.
>
> 2. Indirect judgment vs. direct instruction: Traditional teachers guide students via direct supervision (e.g., distillation), while $ \mathcal{G}_0 $ acts as a reference judge—evaluating outputs via verification scores without participating in training.
>
> This analogy highlights how $ \mathcal{G}_0 $ anchors the system against drift. We’ve added it to the revised discussion—thank you for the insightful framing.
> ***
>
> **Q5: Reusing Real Data: In Remark 3, you note reliance on an "ongoing stream of high-quality real data." What if the same initial real data is reused for every generation? Is new data strictly necessary to prevent collapse?**
>
> **A:** We appreciate this question, as it highlights a critical aspect of our framework. To directly address your inquiry: *reusing the same initial real data in every generation is insufficient to prevent collapse*, and an ongoing stream of new real data is essential for long-term stability. This requirement arises from both theoretical insights and empirical evidence:
>
> **Theoretical Perspective:**  If each generation’s training data uses only original real data $ D_0 $ plus synthetic data $ D_{s,t-1} $ (generated by prior models), this leads to:
>
> 1. Reduced Information Diversity:  As generations progress, the synthetic data becomes a degraded copy of the original real data, with diminishing novel information. This effectively reduces the usable diversity of the training set, gradually forcing the model to train on increasingly lower-quality data.
>
> 2. Error Accumulation: Any biases or imperfections in early synthetic data are amplified in subsequent generations, causing the model's distribution to drift away from the true data manifold. Over time, this drift accumulates, leading to catastrophic forgetting of rare but critical patterns in the original data.
>
> **Empirical Evidence:**  Prior work on recursive training (e.g., [1] with diffusion models and GANs) shows reusing initial real data only delays degradation—models eventually collapse. In contrast, ongoing new real data mitigates this by injecting novel patterns and correcting drift.
>
> We’ve clarified these points in the manuscript, with relevant citations. Thank you for prompting this deeper explanation.
>
> **Reference:**
>
> [1] Alemohammad et al. *Self-consuming generative models go mad.* ICLR 2024.
> ***
>
> **Q6: Training Regimes: What are the three regimes you mention? Briefly naming/describing them in the introduction would help.**
>
> **A:** We appreciate this question. Defining these regimes in the introduction improves clarity, and we’ve revised the section as follows:
>
> 1. **Unverified Full Synthetic Regime:** Trains exclusively on predecessor-generated synthetic data (no real data/verification). This naive setup is prone to rapid drift and collapse from unconstrained error accumulation.
>
> 2. **Fresh Real Data Augmented Regime:** Uses mixed synthetic (from prior models) and new real data, without verification. Fresh real data delays degradation but is unsustainable long-term due to costs and data depletion.
>
> 3. **Verified Full Synthetic Regime:** Trains solely on synthetic data filtered via self-verification (retaining $ \mathcal{G}_0 $-aligned candidates). This avoids ongoing real data reliance while preventing collapse by anchoring to initial real data.
>
> Thank you for enhancing clarity.
> ***
>
> **Q7: Data Source: Just out of curiosity, the framework uses data from a model’s direct predecessor. Have you considered training on data from a stronger model instead?**
>
> A: We appreciate the reviewer’s question. Our choice to use the model’s direct predecessor is driven by a practical motivation: for frontier models, there may be no stronger model available, and training one would require significant compute and fresh real data, which are often already exhausted. This makes self-learning from one's own generations a necessary and realistic paradigm.
>
> That said, if a stronger external expert model were available, our framework could naturally incorporate it—resembling a teacher-student setup and potentially improving performance. However, our focus is on the more challenging and practical case where no stronger teacher exists.
> ***
>
> **Q8: The limitations (e.g., real data dependency) are scattered; a dedicated section on practical issues—such as the computational cost of sampling and a poor initial $ \mathcal{G}_0 $—would be more impactful.**
>
> **A:** Thank you for this constructive suggestion—we agree consolidating limitations into a dedicated section enhances clarity, and have revised the manuscript accordingly.
>
> The new "Limitations" section addresses key practical constraints:
>
> 1. **Computational Overhead of Candidate Sampling:** Generating $ N $ candidates per training point (for verification) introduces additional computational costs, especially at scale. We discuss potential mitigations, such as adaptive candidate selection (e.g., reducing $ N $ for low-uncertainty inputs) or leveraging approximate filtering to balance efficiency and performance.
>
> 2. **Vulnerability to a Poor Initial Anchor:** The framework’s stability hinges on $ \mathcal{G}_0 $ being a reliable encoder of real-data patterns. If $ \mathcal{G}_0 $ is poorly trained (e.g., due to biased or limited initial data), its verification scores may misguide filtering, leading to compounding errors. We note scenarios where this risk is heightened (e.g., small initial datasets) and propose future work on robustifying $ \mathcal{G}_0 $ via techniques like data augmentation or uncertainty quantification.
>
> By consolidating these points, we aim to provide a clearer roadmap for both practical deployment and future research. Thank you for strengthening the manuscript.

---

### Decision · Program_Chairs · 2025-09-17

**Decision:**

Accept (poster)

**Comment:**

This paper presents a set of theoretical results on avoiding model collapse through self-verification. Even though this is largely a theoretical paper, I would encourage the authors to incorporate some of the empirical experiments that they ran during the rebuttal phase into the body as it would make for a stronger contribution.